# Matrix Compression via Randomized Low Rank and Low Precision Factorization

**Rajarshi Saha, Varun Srivastava, Mert Pilanci**

Department of Electrical Engineering
Stanford University
Stanford, CA 94305, USA
`{rajsaha,vsriva,pilanci}@stanford.edu`

## Abstract

Matrices are exceptionally useful in various fields of study as they provide a convenient framework to organize and manipulate data in a structured manner. However, modern matrices can involve billions of elements, making their storage and processing quite demanding in terms of computational resources and memory usage. Although prohibitively large, such matrices are often approximately low rank. We propose an algorithm that exploits this structure to obtain a low rank decomposition of any matrix $\mathbf{A}$ as $\mathbf{A} \approx \mathbf{LR}$, where $\mathbf{L}$ and $\mathbf{R}$ are the low rank factors. The total number of elements in $\mathbf{L}$ and $\mathbf{R}$ can be significantly less than that in $\mathbf{A}$. Furthermore, the entries of $\mathbf{L}$ and $\mathbf{R}$ are quantized to low precision formats – compressing $\mathbf{A}$ by giving us a low rank and low precision factorization. Our algorithm first computes an approximate basis of the range space of $\mathbf{A}$ by randomly sketching its columns, followed by a quantization of the vectors constituting this basis. It then computes approximate projections of the columns of $\mathbf{A}$ onto this quantized basis. We derive upper bounds on the approximation error of our algorithm, and analyze the impact of target rank and quantization bit-budget. The tradeoff between compression ratio and approximation accuracy allows for flexibility in choosing these parameters based on specific application requirements. We empirically demonstrate the efficacy of our algorithm in image compression, nearest neighbor classification of image and text embeddings, and compressing the layers of LlaMa-7b. Our results illustrate that we can achieve compression ratios as aggressive as one bit per matrix coordinate, all while surpassing or maintaining the performance of traditional compression techniques.

## 1 Introduction

Low-rank structures for matrices have proven to be incredibly valuable and ubiquitous across numerous fields of study. Several real-world matrices approximately exhibit low-rank structure due to inherent redundancy or patterns, allowing them to be approximated using low-rank factors. Udell and Townsend [69] provide a potential justification by considering a generative model with latent variables for real-world matrices. Applications where low-rank structures in matrices are exploited include, but are not limited to, imaging (Lingala et al. [34]), fine-tuning large language models (Aghajanyan et al. [3], Hu et al. [25], Karimi Mahabadi et al. [30], Valipour et al. [70], Wang et al. [76]), compressing neural networks (Ben Noach and Goldberg [8], Idelbayev and Carreira-Perpinan [26], Mao et al. [39], Phan et al. [47], Swaminathan et al. [63], Tahaei et al. [64], Wang et al. [77], Winata et al. [78], Yu et al. [84]), obtaining efficient NN architectures (Jaderberg et al. [27], Tai et al. [65]), etc.

Given a matrix $\mathbf{A} \in \mathbb{R}^{n \times d}$, a low-rank approximation is given by $\mathbf{A} \approx \mathbf{LR}$, where $\mathbf{L} \in \mathbb{R}^{n \times m}$, $\mathbf{R} \in \mathbb{R}^{m \times d}$ denote the *left* and *right* low-rank factors with $m \ll \min\{n, d\}$. Since $m(n + d) \ll nd$,

37th Conference on Neural Information Processing Systems (NeurIPS 2023).

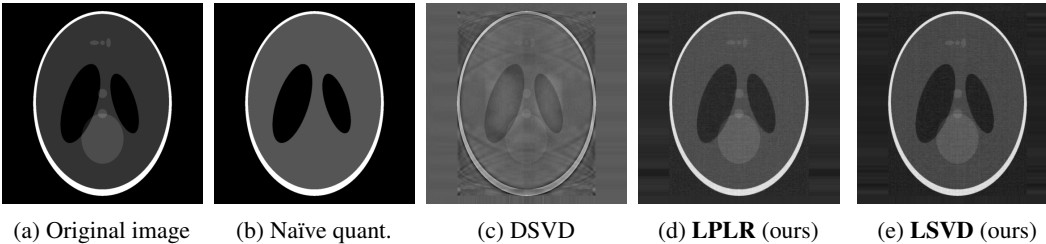

|     |     |     |     |     |
| :-: | :-: | :-: | :-: | :-: |
| (a) Original image | (b) Naïve quant. | (c) DSVD | (d) **LPLR** (ours) | (e) **LSVD** (ours) |

Figure 1: Compression of Shepp-Logan phantom (a standard test image for medical image reconstruction). Naive quant. was done with 2-bits per pixel of this $10^3 \times 10^3$ image. Quantizing the SVD factors "directly" (i.e., DSVD) and (our) LPLR/LSVD algorithms, factorize the image into a product of tall & wide matrices which reduces the total number of elements, allowing each entry to be represented using upto 8-bits of precision per pixel. Despite the increase in precision per pixel, the total number of bits remains the same at $2 \cdot 10^6$.

the total number of entries in **LR** can be significantly smaller than **A**, enabling matrix compression. The need for compression is driven by the overwhelming storage demands and computational complexity linked to large matrices. In a comparable but distinct line of work, low-precision (LP) representations have also been studied extensively as a means to reduce the memory footprint. Quantization of a continuous valued variable using a small number of bits reduces storage requirement while trading off accuracy. In addition, they also facilitate low latency for real-time inference and low energy consumption (see Gholami et al. [21]). Works of Alimisis et al. [4], Safaryan et al. [55] have studied various matrix compression operators for distributed optimization.

In this work, we introduce **LPLR**: *Low-Precision Low-Rank* factorization – a general matrix compression algorithm that simultaneously exploits the low-rank structure of a matrix and quantizes it, to obtain a low-rank factorization, such that the elements of the factors are represented using a small number of bits. Although LPLR can generically refer to a class of algorithms whose main goal is to obtain low-precision representations and exploiting low-rank structures in matrices, in the rest of the paper, we reserve this acronym for our proposed algorithm in Alg. 1. Fig. 1 shows the effectiveness of LPLR in preserving the semantic features in image compression. Other algorithms designed towards achieving the same objective have been named and described appropriately throughout the paper.

## 1.1 Related Works

**Low-rank approximation**: The optimal rank-$k$ approximation of a matrix $\mathbf{A} \in \mathbb{R}^{n \times d}$, denoted as $\mathbf{A}_k$, can be obtained by performing the singular value decomposition (SVD) of $\mathbf{A}$, which gives $\mathbf{A} = \mathbf{U} \mathbf{\Sigma} \mathbf{V}^\top$. To obtain $\mathbf{A}_k$, we keep only the top $k$ singular values along with their corresponding left and right singular vectors, setting the remaining singular values to zero. In other words, $\mathbf{A}_k = \mathbf{U}_k \mathbf{\Sigma}_k \mathbf{V}_k^\top = \sum_{i=1}^{k} \sigma_i \mathbf{u}_i \mathbf{v}_i^\top$, where $\sigma_i$, $\mathbf{u}_i$, and $\mathbf{v}_i$ represent the $i$-th singular value, left singular vector, and right singular vector, respectively. The matrix $\mathbf{A}_k$ minimizes the Frobenius norm $\|\mathbf{A} - \widehat{\mathbf{A}}\|_{\mathrm{F}}$ under the constraint that the rank of $\widehat{\mathbf{A}}$ is at most $k$. This result is known as the Eckart-Young-Mirsky theorem [18], which states that $\|\mathbf{A} - \mathbf{A}_k\|_{\mathrm{F}}^2 = \sum_{i>k} \sigma_i^2$. However, computing the SVD has a high computational complexity of $\mathrm{O}(nd^2)$ ($n \geq d$ without loss of generality), which can be impractical for large matrices. Therefore, alternative methods such as in Jin et al. [28], Ye and Du [82], Zhang et al. [85] have been proposed which solve variants of the minimization problem. Chi et al. [11] provide a survey of optimization-based approaches for obtaining low-rank approximation.

Along a parallel line of work, randomized low-rank factorization algorithms have demonstrated their effectiveness in handling massive datasets efficiently while maintaining competitive accuracy levels. Several works, including those by Derezinski et al. [13], Drineas and Mahoney [16], Drineas et al. [17], Halko et al. [23], Ma and Solomonik [37], Mahoney [38], Martinsson et al. [43], Tropp et al. [67], Witten and Candès [79] have focused on reducing the complexity of computing SVD to approximately $\mathrm{O}(nm^2)$, where $m \ll d$. These algorithms aim to approximate the range space of matrix $\mathbf{A}$ by utilizing a sketched version $\mathbf{AS}$, where $\mathbf{S} \in \mathbb{R}^{d \times m}$ is a random matrix with $m \ll d$. The columns of $\mathbf{A}$ are then projected onto this approximate range space. By choosing $m = k + p$, where $p \geq 2$ is a small integer, the resulting low-rank approximation $\mathbf{A} \approx \mathbf{LR}$ provided by these algorithms can be proven to be within a small multiplicative factor of the optimal rank-$k$ approximation, stated as $\mathbb{E} \|\mathbf{LR} - \mathbf{A}\|_{\mathrm{F}}^2 \leq (1 + \delta) \|\mathbf{A}_k - \mathbf{A}\|_{\mathrm{F}}^2$. The sketching matrix $\mathbf{S}$ is typically selected from the

widely used class of Johnson-Lindenstrauss (JL) embeddings. A popular choice when $\mathbf{A}$ is dense is to sample a Gaussian matrix where each entry $S_{ij} \sim \mathcal{N}(0, \frac{1}{m})$.

**Randomized quantization**: JL embeddings also have practical uses in vector quantization. In this context, when quantizing a vector $\mathbf{x} \in \mathbb{R}^d$, instead of quantizing $\mathbf{x}$ directly, the encoder quantizes $\mathbf{Sx}$, and then the decoder obtains an approximation of $\mathbf{x}$ through $\mathbf{S}^\top \mathbf{Q}(\mathbf{Sx}) \approx \mathbf{x}$. This randomized transformation equalizes the coordinate values of $\mathbf{x}$, allowing for a smaller range of values and higher precision for the quantizer Q under a fixed bit-budget. As a result, it leads to a smaller $\ell_2$ error compared to independently quantizing each coordinate in a naive manner. Works such as Chen et al. [10], Lyubarskii and Vershynin [36], Mayekar and Tyagi [44], Safaryan et al. [54], Saha et al. [56, 57, 58], Studer et al. [60], Suresh et al. [61, 62], Vargaftik et al. [71, 72], Young et al. [83] explore different variations of this concept for different applications.

## 2 Proposed Algorithm

Prior to detailing our algorithm, we first review the characteristics of the *uniformly dithered quantizer* that we employ within our approach. Uniformly dithered quantization has been utilized in various prior works Alistarh et al. [5], Bernardo et al. [9], Gray and Stockham [22], Mayekar and Tyagi [44], Suresh et al. [61, 62], and is succinctly described in §(2.1) below.

### 2.1 Uniformly dithered quantizer

Let us consider quantizing a scalar $x$ with $|x| \leq \mathrm{R}$. Given a *bit-budget* of B bits, the scalar quantizer with *dynamic range* R is described by first specifying the $M = 2^{\mathrm{B}}$ quantization points as:
$$q_1 = -\mathrm{R}, q_2 = -\mathrm{R} + \Delta, q_3 = -\mathrm{R} + 2\Delta, \ldots, q_M = -\mathrm{R} + (M-1)\Delta.$$
Here, the resolution is given by $\Delta = \frac{2\mathrm{R}}{M-1}$, and the quantizer operation is defined as:

$$\mathrm{Q}_{\mathrm{R,B}}(x) = \begin{cases} q_{k+1} & \text{with probability } r, \\ q_k & \text{with probability } 1-r, \end{cases} \tag{1}$$

where $k = \arg\max_j\{q_j \leq x\}$, i.e., $x \in [q_k, q_{k+1})$, and $r = \frac{x-q_k}{\Delta}$. Such a quantizer satisfies

$$\mathbb{E}\left[\mathrm{Q}_{\mathrm{R,B}}(x)\right] = x \quad \text{and} \quad \mathbb{E}\left(\mathrm{Q}_{\mathrm{R,B}}(x) - x\right)^2 \leq \frac{\Delta^2}{4} = \frac{\mathrm{R}^2}{\left(2^{\mathrm{B}} - 1\right)^2}, \tag{2}$$

i.e., it is unbiased and the quantization error variance is dictated by R and B. Here, the $\mathbb{E}(\cdot)$ is over the randomness from dithering in (1) (ref. App. C). If the input $x$ to the quantizer falls outside this range, i.e., $x > \mathrm{R}$ or $x < -\mathrm{R}$, the quantizer is said to be *saturated*. Finally, to quantize any matrix $\mathbf{X}$, we obtain $\mathrm{Q}_{\mathrm{R,B}}(\mathbf{X})$ by quantizing each entry independently, i.e., $\left[\mathrm{Q}_{\mathrm{R,B}}(\mathbf{X})\right]_{ij} \triangleq \mathrm{Q}_{\mathrm{R,B}}(X_{ij})$.

### 2.2 Proposed algorithm: LPLR

In order to comprehend low precision and low rank representations, we first introduce a simple strategy, which we name as *direct-SVD quant*. This method involves two main steps: It first computes the optimal rank-$k$ decomposition $\mathbf{A}_k = \mathbf{U}_k \boldsymbol{\Sigma}_k \mathbf{V}_k^\top$, and then, it quantizes the low-rank factors independently, namely, $\mathbf{L} = \mathrm{Q}(\mathbf{U}_k \boldsymbol{\Sigma}_k)$ and $\mathbf{R} = \mathrm{Q}'(\mathbf{V}_k^\top)$. Here, Q and Q' are uniform scalar quantizers with respective bit-budgets B and B'. A detailed analysis of this direct-SVD quantization approach can be found in Appendix H, along with pseudocode provided in Algorithm 2. Despite its simplicity, this is not the optimal strategy due to two reasons. Firstly, it necessitates computing the SVD of $\mathbf{A}$, which is computationally expensive at $\mathrm{O}(nd^2)$. Secondly, let us consider improving the approximation to $\mathbf{A}$ when the first factor is fixed to $\mathrm{Q}(\mathbf{U}_k)$ by solving the optimization problem:

$$\mathbf{X}^* = \arg\min_{\mathbf{X} \in \mathbb{R}^{k \times d}} \|\mathrm{Q}(\mathbf{U}_k)\mathbf{X} - \mathbf{A}\|_{\mathrm{F}}^2 = \mathrm{Q}(\mathbf{U}_k)^\dagger \mathbf{A}. \tag{3}$$

This computes the projection of the columns of $\mathbf{A}$ onto the range space of $\mathrm{Q}(\mathbf{U}_k)$. Suppose the resulting projection coefficients are further quantized to get $\mathrm{Q}'(\mathrm{Q}(\mathbf{U}_k)^\dagger \mathbf{A})$. Since $\mathbf{X}^*$ is the solution of (3), for a sufficiently large value of B', it is evident that $\left\|\mathrm{Q}(\mathbf{U}_k)\mathrm{Q}'(\mathrm{Q}(\mathbf{U}_k)^\dagger \mathbf{A}) - \mathbf{A}\right\|_{\mathrm{F}}^2 \leq \left\|\mathrm{Q}(\mathbf{U}_k \boldsymbol{\Sigma}_k)\mathrm{Q}'(\mathbf{V}_k^\top) - \mathbf{A}\right\|_{\mathrm{F}}^2$, and hence, better than direct-SVD quant. However, this approach of projecting onto the range space of $\mathrm{Q}(\mathbf{U}_k)$ (which we refer to as LPLR-SVD and analyze in App. I), still requires the computation of $\mathbf{U}_k$, so we can replace $\mathbf{U}_k$ by $\mathbf{AS}$, i.e., an approximation of the

Algorithm 1: **LPLR**: Randomized Low-Precision Low-Rank factorization

---

**Input** : Matrix $\mathbf{A} \in \mathbb{R}^{n \times d}$, sketch size $m$, Quantizers Q, Q$'$ with dynamic ranges R$_Q$, R$_{Q'}$ and bit-budgets B, B$'$ respectively.

**Output** : Factorization: $\mathbf{LR}$ where $\mathbf{L} \in \mathbb{R}^{n \times m}, \mathbf{R} \in \mathbb{R}^{m \times d}$

1  Sample a Gaussian sketching matrix $\mathbf{S} \in \mathbb{R}^{d \times m}$ with entries $S_{ij} \sim \mathcal{N}\left(0, \frac{1}{m}\right)$.
2  Compute an approximate basis of column space of $\mathbf{A}$ by forming the sketch: $\mathbf{AS}$.
3  Quantize the approximate basis with Q to get Q($\mathbf{AS}$).
4  Find $\mathbf{W}^* = \arg\min_{\mathbf{W}} \|Q(\mathbf{AS})\mathbf{W} - \mathbf{A}\|_F^2$.
5  Quantize $\mathbf{W}^*$ using quantizer Q$'$ to get Q$'(\mathbf{W}^*)$.

6  **return** *Low-rank and low-precision approximation* $\mathbf{LR}$ *where* $\mathbf{L} = Q(\mathbf{AS})$, $\mathbf{R} = Q'(\mathbf{W}^*)$.

---

range space obtained through random linear combinations of the columns of $\mathbf{A}$ (also known as a randomized rangefinder [42]). This leads us to our **LPLR** algorithm, described in Alg. 1, which finds $\mathbf{W}^* = \arg\min_{\mathbf{W} \in \mathbb{R}^{k \times d}} \|Q(\mathbf{AS})\mathbf{W} - \mathbf{A}\|_F^2$ and forms the low-rank low-precision approximation $Q(\mathbf{AS})Q'(\mathbf{W}^*)$ where Q, Q$'$ are quantization operators. While the solution of this problem is available in closed form as $\mathbf{W}^* = Q(\mathbf{AS})^\dagger \mathbf{A}$, one can also use an approximation of $\mathbf{W}^*$ by solving this least-squares minimization using an iterative method such as conjugate gradient descent.

In addition to the above argument supporting the superiority of LPLR compared to other baselines (ref. to Tabs. 1 and 2), there exists another essential reason why LPLR outperforms them. This reason directly relates to the selection of $\mathbf{S}$ as a Gaussian matrix, which is an integral component of LPLR. Random Gaussian matrices are JL embeddings and possess an equalization property that enhances the precision of uniform quantizers. In particular, let us consider an arbitrary vector $\mathbf{x} \in \mathbb{R}^d$ and obtain an estimate as $\widehat{\mathbf{x}} = \mathbf{S}^\top Q(\mathbf{Sx})$ using a uniform quantizer Q. It can be shown that the vector quantization error remains constant and does not grow with the dimension $d$, expressed as $\mathbb{E}\|\widehat{\mathbf{x}} - \mathbf{x}\|_2^2 = O(1)$. This represents a substantial improvement compared to the naïve strategy of independently quantizing each coordinate of $\mathbf{x}$, which leads to a quantization error growth rate of $O(d)$. We provide a detailed explanation of this phenomenon in App. D. Furthermore, even when the quantization is 1-bit per coordinate, e.g., $Q(\mathbf{Sx}) = \text{Sign}(\mathbf{Sx})$, this embedding provides strong near-isometric embedding properties due to the properties of random hyperplane tessellations [50].

While the strong equalization property of Gaussian embeddings is a known result, certain works such as Saha et al. [57, 58], Suresh et al. [61, 62] opt for using randomized Hadamard embeddings instead of Gaussian ones. The reason behind this choice is twofold: (i) Gaussian matrices are dense, requiring $O(d^2)$ multiplications when computing $\mathbf{Sx}$, and (ii) the entries of $\mathbf{S}$ are floating point numbers that must be stored in full precision, contradicting the objective of quantizing $\mathbf{x}$ using fewer bits. However, these concerns are not problematic for **LPLR** because the effects of $\mathbf{S}$ in the first low-rank factor are neutralized by the second low-rank factor, and $\mathbf{S}$ does not need to be stored. In fact, we exploit both the equalization property and the subspace approximation property of Gaussian matrices to derive a superior upper bound for the approximation error, as discussed next in §3.

## 3  Approximation Error Analysis

We are now in a position to state the approximation error guarantee of **LPLR** Let us denote the $i^{\text{th}}$ row of our input matrix $\mathbf{A}$ as $\mathbf{a}^{(i)}$. For convenience of analysis, we make the following assumption.

**Assumption 3.1.** Rows of matrix $\mathbf{A}$ have bounded norm, i.e., $\|\mathbf{a}^{(i)}\| \leq R$ for some known $R > 0$.

The following result gives an informal upper bound on the expected Frobenius norm error of the factorization returned by Alg. 1.

**Theorem 3.2. LPLR approximation error (Informal)** *Suppose our input matrix* $\mathbf{A} \in \mathbb{R}^{n \times d}$ *with* $\|\mathbf{a}^{(i)}\| \leq R = O(1)$ *has singular values* $\sigma_1, \ldots, \sigma_r$ *with* $r = \text{rank}(\mathbf{A})$ *and target rank as* $k$. *Let* $\kappa(\mathbf{A}) = \sigma_1/\sigma_r$ *and* $\kappa(\mathbf{A}_k) = \sigma_1/\sigma_k$ *respectively be the condition numbers of* $\mathbf{A}$ *and the best rank-$k$ approximation of* $\mathbf{A}$, *and let us denote* $\kappa = \min\left\{\kappa(\mathbf{A}), \kappa(\mathbf{A}_k)\left(1 - c_4\sigma_{k+1}/\sigma_k\right)^{-1}\right\}$. *Furthermore, for a sufficiently small constant* $\epsilon > 0$, *suppose the dynamic range of* Q *is set to be* $c_1\sqrt{\log\left(n/\epsilon\right)/m}$,

*and that of* $Q'$ *is set to* $2\kappa\sqrt{m/d}$. *Then, the* **LPLR** *factorization returned by Alg. 1 satisfies*

$$\mathbb{E}\left\|\mathbf{LR} - \mathbf{A}\right\|_{\mathrm{F}}^2 \leq \left(1 + \frac{k}{m-k-1}\right)\left\|\mathbf{A}_k - \mathbf{A}\right\|_{\mathrm{F}}^2 + \epsilon,$$

*while utilizing a total budget of* $\log_2\left(\frac{c\kappa(\mathbf{A}_k)\kappa}{\epsilon}\frac{nm}{\sqrt{d}}\sqrt{\log\left(\frac{mn^2}{\epsilon}\right)}\right)$ *bits for* $n \approx d$. *Here,* $c_1$, $c_2$, $c_3$, *and* $c_4$ *are constants that depend on* R.

We provide a less formal version of our main result here, suitable for interpretation. The formal statement of this result, including specific constant values, can be found in Thm. G.2 of App. G. It does not necessitate the assumption $n \approx d$ and provides distinct thresholds for B and B'. Thm. G.2 asserts that, for a target rank-$k$, as long as $m \geq k + 2$, one can ensure an arbitrarily small approximation error of $\epsilon$ by selecting the number of bits to be at least above a certain threshold budget. The threshold depends on the error tolerance $\epsilon$, dimensions $n$ and $d$, the sketch size $m$, and the spectrum of $\mathbf{A}$. The value of $\kappa$ is determined by taking the smaller of two quantities. In the case of matrices with a sharp decline in singular values (e.g., matrices of exact rank-$k$), where the ratio $\sigma_k/\sigma_{k+1}$ approaches zero, $\kappa \approx \kappa(\mathbf{A}_k)$. For matrices with a smoother spectrum (e.g., all singular values are equal), $\kappa = \kappa(\mathbf{A})$, the condition number of the input matrix $\mathbf{A}$.

**Remark 1.** We consider two distinct scenarios based on Asm. 3.1. The first case assumes that the row norms are bounded by a constant, represented by $R = O(1)$. This assumption is reasonable when the rows of $\mathbf{A}$ correspond to different normalized features of a data point. The second case assumes that the individual entries of $\mathbf{A}$ are bounded by a constant, i.e., $A_{ij} = O(1)$. This implies that $R = O(\sqrt{d})$, which is a reasonable assumption for scenarios like images, where it is known that each pixel value is bounded. In Tab. 1, we compare the performance of the algorithms when $R = O(1)$, while in Tab. 2, we assume $R = O(\sqrt{d})$. Thm. 3.2 assumes that $R = O(1)$. The expressions in Tab. 2 can be obtained in a similar manner from the formal version in Thm G.2.

## 3.1 Analysis outline

The derivation of the upper bound on the approximation error of LPLR is presented in App. G. In this section we outline a brief proof sketch that highlights the main challenges of the proof. As mentioned already in §2.2, the analysis of LPLR utilizes the subspace embedding and equalization properties of random Gaussian matrices. A key component in Alg. 1 is the choice of dynamic range for quantizers Q and $Q'$. In our analysis, we assume that when either Q or $Q'$ gets saturated in lines 3 and 5 of Alg. 1, a trivial factorization of $\mathbf{LR} = \mathbf{0}$ is returned. We choose the dynamic ranges $R_Q$ and $R_{Q'}$ to be sufficiently high enough so that this happens with a very low probability. Formally, for quantizer Q, Lemma E.1 states the following:

$$\left\|\mathbf{AS}\right\|_{\max} \leq R\sqrt{\frac{2\log\left(\frac{16R^2n^2m}{\epsilon}\right)}{m}} \quad \text{with probability exceeding } 1 - \frac{\epsilon}{8nR^2}.$$

Here, $\left\|\mathbf{AS}\right\|_{\max}$ is max-norm of the matrix $\mathbf{AS}$, i.e., the coordinate with maximum magnitude. This concentration result is a consequence of the equalization property of Gaussian matrix $\mathbf{S}$.

On the other hand, the input to the second quantizer is $Q(\mathbf{AS})^{\dagger}\mathbf{A}$ and Lemma E.2 provides a concentration result for the max-norm of this matrix. Although in a general worst-case scenario, the coordinate values of the pseudo-inverse of a matrix with small entries can be large (Alon and Vu [6]), because we compute the pseudo-inverse of the matrix $\mathbf{AS}$, which is rectangular ($m \ll n$) and with random Gaussian entries, $\left\|(\mathbf{AS})^{\dagger}\right\|_2$ does not shoot up arbitrarily as shown by Rudelson and Vershynin [53]. We then show that $\left\|Q(\mathbf{AS})^{\dagger}\right\|_2$ is not too far from $\left\|(\mathbf{AS})^{\dagger}\right\|_2$, allowing us to derive an expression for $R_{Q'}$. We get for $\gamma = \frac{d}{m}$ and $t = \sqrt{\frac{2\log\left(\frac{32nR^2}{\epsilon}\right)}{m}}$,

$$\left\|Q(\mathbf{AS})^{\dagger}\mathbf{A}\right\|_{\max} \leq \frac{2\kappa}{\sqrt{\gamma} - 1 - t} \quad \text{with probability exceeding } 1 - \frac{\epsilon}{4nR^2}.$$

The second part of the analysis deals with upper bounding the approximation error when the second low-rank factor is unquantized, i.e., $\left\|Q(\mathbf{AS})Q(\mathbf{AS})^{\dagger}\mathbf{A} - \mathbf{A}\right\|_{\mathrm{F}}^2$ conditioned on the event that Q and $Q'$ are unsaturated. We reduce this problem to analyzing the solution of the following:

$$\widetilde{\mathbf{X}} = \arg\min_{\mathbf{X}}\left\|\mathbf{X}^{\top}\mathbf{A}_k\mathbf{S} - Q(\mathbf{AS})\right\|_{\mathrm{F}}^2. \tag{4}$$

Table 1: Comparison with baselines (row-norm bound is constant, i.e., $\|\mathbf{a}^{(i)}\| = \mathrm{O}(1)$). $k, m \ll \min\{d, n\}$. $n$: no. of rows, $d$: no. of columns, $m$: sketch size, $\epsilon$: error tolerance, $\delta = k/(m - k - 1)$. The expressions for bit-budget (per entry) ignores constant multiplicative factors inside the $\log_2(\cdot)$. We assume $n \geq d$.

| Algorithms | Approximation error | Bit-budget (per entry) | Computation |
|---|---|---|---|
| Naïve uniform | $\epsilon$ | $\frac{1}{2} \log_2 \left( \frac{nd}{\epsilon} \right)$ | $\mathrm{O}(nd)$ |
| Direct-SVD | $\|\mathbf{A}_k - \mathbf{A}\|_{\mathrm{F}}^2 + \epsilon$ | $\frac{1}{2} \log_2 \left( \frac{k\sigma_1^2}{\epsilon} \sqrt{nd} \right)$ | $\mathrm{O}(nd^2)$ |
| **LPLR** (*ours*) | $(1 + \delta) \|\mathbf{A}_k - \mathbf{A}\|_{\mathrm{F}}^2 + \epsilon$ | $\frac{1}{2} \log_2 \left( \frac{\kappa(\mathbf{A}_k)\kappa}{\epsilon} \frac{nm}{\sqrt{d}} \sqrt{\log\left(\frac{mn^2}{\epsilon}\right)} \right)$ | $\mathrm{O}(ndm)$ |

Table 2: Comparison with baselines (individual entries of $\mathbf{A}$ are bounded by a constant, i.e., $A_{ij} = \mathrm{O}(1)$). Dimension dependent terms are color highlighted for ease of comparison with Tab. 1.

| Algorithms | Approximation error | Bit-budget (per entry) | Computation |
|---|---|---|---|
| Naïve uniform | $\epsilon$ | $\frac{1}{2} \log_2 \left( \frac{nd}{\epsilon} \right)$ | $\mathrm{O}(nd)$ |
| Direct-SVD | $\|\mathbf{A}_k - \mathbf{A}\|_{\mathrm{F}}^2 + \epsilon$ | $\frac{1}{2} \log_2 \left( \frac{k\sigma_1^2}{\epsilon} \sqrt{nd} \right)$ | $\mathrm{O}(nd^2)$ |
| **LPLR** (*ours*) | $(1 + \delta) \|\mathbf{A}_k - \mathbf{A}\|_{\mathrm{F}}^2 + \epsilon$ | $\frac{1}{2} \log_2 \left( \frac{\kappa(\mathbf{A}_k)\kappa}{\epsilon} nm \sqrt{d \log\left(\frac{mn^2 d^2}{\epsilon}\right)} \right)$ | $\mathrm{O}(ndm)$ |

We refer to (4) as the *sketched least squares problem with quantized response* as it is a variant of the generalized least squares problem, $\mathbf{X}^* = \arg\min_{\mathbf{X}} \left\| \mathbf{X}^\top \mathbf{A}_k - \mathbf{A} \right\|_{\mathrm{F}}^2$. This is potentially a problem of independent interest, and we analyze the solution of (4) in detail in App. F. Exploiting the subspace embedding property of $\mathbf{S}$ we show that

$$\left\| \mathbf{X}^{*\top} \mathbf{A}_k - \mathbf{A} \right\|_{\mathrm{F}}^2 \leq \mathbb{E} \left\| \widetilde{\mathbf{X}}^\top \mathbf{A}_k - \mathbf{A} \right\|_{\mathrm{F}}^2 \leq \frac{m-1}{m-k-1} \left\| \mathbf{X}^{*\top} \mathbf{A}_k - \mathbf{A} \right\|_{\mathrm{F}}^2 + \text{quantization error term.}$$

This leads us to the proof of Lemma G.1 which gives the approximation error of LPLR when Q and Q' are unsaturated. Finally, taking into account the low-probability saturation events, for which the error is $\|\mathbf{A}\|_{\mathrm{F}}^2$, we derive our main result in Thm. G.2. Subsequently, we discuss the approximation made in App. G.2 and arrive at the informal result of Thm. 3.2.

### 3.2 Comparison with baselines

We are now in a position to compare the performance with baselines in Tabs. 1 and 2.

**Naive quantization**: The most straightforward baseline for matrix quantization is naïve quantization where each coordinate of the matrix is quantized independently, agnostic to any low-rank structure in the matrix $\mathbf{A}$. In this, we allocate B bits to each coordinate of $\mathbf{A}$ and since there are $nd$ entries in the matrix, from (1), the Frobenius norm error is upper bounded by $\frac{\mathrm{R}^2 nd}{(2^{\mathrm{B}} - 1)^2}$. Note that this holds true irrespective of whether $\mathrm{R} = \mathrm{O}(1)$ or $\mathrm{O}(\sqrt{d})$ because $\|\mathbf{a}^{(i)}\| \leq \mathrm{R}$ implies $A_{ij} \leq \mathrm{R}$. To ensure that the error is within a certain tolerance $\epsilon$, we then require $\frac{1}{2} \log_2 \left( \frac{nd}{\epsilon} \right)$ bits. In this, and also other expressions for bit-budget requirements of algorithms in Tabs. 1 and 2, we have ignored the multiplicative constant factors inside the $\log_2(\cdot)$ for simplicity of exposition. The exact expressions can be found in the corresponding appendices where we derive them.

One of the primary reasons why both direct-SVD and LPLR are expected to perform better than naïve is that the former strategies exploit the low-rank structure of the matrices to reduce the total number of parameters being quantized, i.e., $k(n + d)$ for direct-SVD and $m(n + d)$ for LPLR, vs. $nd$ for naïve. Given a total bit-budget for the entire matrix $\mathbf{A}$, since we now quantize fewer parameters than before, we can allocate a higher number of bits to each parameter, enabling higher precision. The price we pay for exploiting the low-rank structure of matrices is the additional $\|\mathbf{A}_k - \mathbf{A}\|_{\mathrm{F}}^2$ dependent term, which is usually very small for matrices that can be well approximated by a low-rank structure. For matrices that are exactly rank $k$, this term is 0. As we see in our numerical simulations in §4, several real-world matrices can be well-approximated by a low-rank structure.

**Direct-SVD quant.**: From Tab. 1, we see that to achieve an $\epsilon$-quantization error, direct-SVD requires $\frac{1}{2} \log_2(k\sqrt{nd})$ bits per entry, which is greater than $\frac{1}{2} \log_2 \left( \frac{nm}{\sqrt{d}} \right)$ required by LPLR (ignoring the

logarithmic terms). Evidently, LPLR demands fewer bits than direct-SVD because $k, m \ll \min\{n, d\}$ for inherently low-rank matrices. For the regime presented in Tab. 2, the bit requirement for direct-SVD remains unchanged. However, LPLR now requires $\frac{1}{2} \log_2 \left( nm\sqrt{d} \right)$, slightly more than direct-SVD, due to the additional $\sqrt{n}$ factor. Thus, it makes sense to expect that direct-SVD can perform better in this regime. This is supported by our numerical experiments in Tabs. 4 to 7, where direct-SVD indeed outperforms LPLR in certain scenarios. Nevertheless, it is crucial to emphasize that direct-SVD necessitates computing the SVD, which can be prohibitive for very large matrices due to the current memory limitations of available GPUs, making LPLR the only viable option.

**Computational complexity**: Unsurprisingly, naïve quant. requires the least computation, i.e., $\mathrm{O}(nd)$, as it just does a single pass over all the elements of $\mathbf{A}$. The $\mathrm{O}(nd^2)$ complexity of direct-SVD quant. stems from the requirement of computing SVD (assuming $n \geq d$). LPLR is the best of both worlds – for the same bit-budget, LPLR has a smaller approximation error than both direct-SVD and naïve, and a complexity of $\mathrm{O}(ndm)$, arising from the requirement to compute $\mathbf{AS}$, i.e., a product of two dense matrices of dimensions $n \times d$ and $d \times m$, which is better than direct-SVD, since $m \ll d$.

# 4  Numerical Simulations

## 4.1  Overview

We evaluate the robustness of LPLR on multiple tasks, namely, image compression, binary, and multi-class classification across disparate domains including vision, text and raw images, and neural network weight matrices. We consider a range of input configurations to showcase the performance and non linear effects of joint quantization and low rank approximation on a given dataset, especially at lower bit budgets. LPLR provides competitive results at bit budgets as low as a single bit, providing extreme model compression while maintaining non trivial performance for the task at hand.

**Baselines.** We employ naive quantization, which quantizes the input matrix by rounding to the nearest scalar in the underlying data type's quantization grid, as our primary benchmark. Naïve quant. and its variants – Dettmers et al. [14], Yao et al. [81], are the most popular method in use across domains, as their memory and computational run time requirements scale extremely well with model and dataset sizes. In addition, we also evaluate the performances of direct-SVD quantization and LPLR-SVD to disambiguate between the entangled effects of quantization and exploiting low rank structure.

**Metrics.** We evaluate LPLR performance using task specific goodness of fit metrics, as well as relative Frobenius norm error between the original and matrix reconstructed from its low-rank factors, i.e., $\|\mathbf{LR} - \mathbf{A}\|_{\mathrm{F}}^2$. In addition to this, we enforce parity between the number of bits used by all quantization schemes, so that the total space required (in bits) for storing the approximated matrix is *identical* across LPLR, LPLR-SVD, direct-SVD quant., and naïve quant.

**Notation.** In all our experiments, we denote the bit budget for the left low rank factor $\mathbf{L}$ as B, for the right low rank factor $\mathbf{R}$ as B$'$, and the corresponding bit budget for naive quantization as $\mathrm{B}_{\mathrm{nq}}$. For simplicity, we maintain equal bit budgets B = B$'$ for both quantizers Q and Q$'$. Wherever necessary, we also abbreviate direct-SVD quant. as DSVD, LPLR-SVD as LSVD, and naïve quant. as NQ.

The main algorithm is implemented in Pytorch (Paszke et al. [46]), and utilizes Hugging Face [80] implementations of all datasets and large language models. All experiments were performed on a single GPU NVIDIA TITAN RTX. Further simulations and experimental details can be found in App. J. Our code is available at `https://github.com/pilancilab/matrix-compressor`.

## 4.2  Image Compression

Image compression is a prototypical application of low rank matrix compression, as images are known to be significantly rank deficient in many practical scenarios (Zhang et al. [86], Zhou et al. [87]). In this task, we apply LPLR on $1000 \times 1000$ dimensional Shepp Logan phantom images from Gach et al. [20]. These are a set of synthetic 2D images designed to simulate the typical characteristics and structures found in computed tomography (CT) scans. They consist of geometric shapes, including circles and ellipses, representing different tissues or organs within the scanned object.

The main results are summarized in Tab. 3. To ensure a fair comparison, we adjust the sketch size/target rank so that bit budgets are identical between naïve quant. and LPLR. This allows

us to preserve the original datatype of the image (consequently a large dynamic range), while substantially reducing the pixels used for representing the image to as low as 1 bit per pixel (on average). Specifically, in Figure 1, we can observe the least visual distortion in the case of LPLR, which preserves *critical* semantic features of the images, such as the small ellipses. It is clear that LPLR outperforms both techniques at lower naive quantization bit budgets. We attribute the better visual and quantitative performance to the higher dynamic range available to LPLR as well as structure preserved in the low rank decomposition.

Table 3: Comparison of **LPLR** and **LPLR-SVD (LSVD)** Frobenius norm errors with baselines, for different input LPLR bit budgets. Each triplet $(B, B', B_{nq})$ of configurations has an **identical compression ratio**. Here, $B = B'$. The second column specifies the sketch size $m$ for LPLR, and target rank $k$ for DSVD or LSVD. We provide results for input bit budgets at a finer granularity to identify regimes where naïve quant. is outperformed.

| B | Target Rank $(k)$ / Sketch Size $(m)$ | $B_{nq}$ | LPLR | DSVD | LSVD | NQ | | B | Target Rank $(k)$ / Sketch Size $(m)$ | $B_{nq}$ | LPLR | DSVD | LSVD | NQ |
|---|---|---|---|---|---|---|---|---|---|---|---|---|---|---|
| 32 | 15 | 1 | 0.610 | 0.553 | **0.506** | 0.532 | | 32 | 31 | 2 | 0.447 | 0.523 | 0.392 | **0.312** |
| 28 | 17 | 1 | 0.557 | 0.546 | **0.490** | 0.532 | | 28 | 35 | 2 | 0.434 | 0.521 | 0.380 | **0.312** |
| 24 | 20 | 1 | 0.540 | 0.537 | **0.454** | 0.532 | | 24 | 41 | 2 | 0.401 | 0.517 | 0.358 | **0.312** |
| 20 | 25 | 1 | 0.485 | 0.529 | **0.426** | 0.532 | | 20 | 50 | 2 | 0.371 | 0.513 | 0.331 | **0.312** |
| 16 | 31 | 1 | 0.447 | 0.523 | **0.391** | 0.532 | | 16 | 62 | 2 | 0.341 | 0.509 | **0.308** | 0.312 |
| 12 | 41 | 1 | 0.402 | 0.518 | **0.360** | 0.532 | | 12 | 83 | 2 | 0.310 | 0.506 | **0.286** | 0.312 |
| 8 | 62 | 1 | 0.340 | 0.508 | **0.326** | 0.532 | | 8 | 125 | 2 | 0.267 | 0.499 | **0.284** | 0.312 |

## 4.3 Embeddings extracted from pre-trained models

The efficacy of pre-trained embeddings is well established in vision (Li et al. [33], Parisi et al. [45]), text (Qi et al. [51], Rezaeinia et al. [52]) for rapid feature computation as an input to a variety of downstream tasks. Embeddings also play a crucial role in a number of software applications, including but not limited to, open source vector search libraries (Liu [35], Marqo [41]), semantic search engines (Amazon AWS [7]), vector databases (Pinecone [49]). Since most applications rely on proximity in "embedded space", it is essential that common operations on embeddings be computationally efficient. Specifically, one would like to optimize nearest neighbor (NN) searches which solve the optimization problem $\arg \min_i \|\mathbf{x}_i - \mathbf{y}\|_2^2$ (reducible to $\arg \max_i \mathbf{X}\mathbf{y}$ where $\mathbf{X}$ is the matrix with training vectors $\{\mathbf{x}_i\}$ as its rows, and $\mathbf{y}$ is the query vector). The time complexity of NN search scales linearly with dimensions of $\mathbf{X} \in \mathbb{R}^{n \times d}$ and number of neighbors $(k)$. By embedding the data matrix in a dimension $m \ll d$, we directly speedup the run-time and reduce storage costs. Moreover, as datasets grow exponentially in size (especially document databases) and transfer learning becomes the dominant modality of training new models, embedding compression becomes a necessity for storing data without a corresponding exponential increase in hardware requirements.

### 4.3.1 Embedding Classification

In this experiment, we evaluate several embeddings of standard datasets, namely CIFAR-10, CIFAR-100, IMDB and Emotion datasets. CIFAR-10 consists of 60,000 color images divided into 10 classes, with each class containing 6,000 images. The dataset is split into 50,000 training images and 10,000 test images, with a resolution of 32x32 pixels. CIFAR-100 increases the number of classes to 100 categories for an identical training and test size as CIFAR-10. The IMDB (mte [2]) dataset consists of 25,000 train and test sentences containing annotated binary sentiment labels for movie reviews, plot summaries and other rating information. The Emotion (mte [1]) dataset is a sentiment analysis dataset, containing 16,000 train and 2000 test sentences, each exemplifying a singular emotion, which represents the sentiment label for that sentence.

For CIFAR-10 and CIFAR-100, we embed the entire dataset using MobileNet v3 (Howard et al. [24]) pretrained on ImageNet (Deng et al. [12]) producing an embedding matrix of dimension $60000 \times 1024$, which we compress using LPLR and compare with the baselines in §4.1. To evaluate the goodness of embeddings, we build a 3-NN, a KNN Classifier using $K = 3$ nearest neighbors under Euclidean distance). We report the performance of the model using standard classification metrics – classification accuracy and weight averaged F1 score. We utilize a uniform bit budget $B = B' = 8$ bits for the quantizers $Q, Q'$ across all cases. Tabs. 4 and 5 present our results under this setup. For each case we benchmark absolute performance using a 3-NN classifier on the training set.

Similarly, we embed text sentences from IMDB and Emotion databases using BeRT (Devlin et al. [15]) into 512 dimensional vectors, and construct a 3-NN classifier using Euclidean distance to perform binary and multi-class classification on the respective embeddings, and report classification metrics in Tabs. 6 and 7. We see that LPLR outperforms direct-SVD quant. and naïve quant. at lower bit budgets, and has performance parity as we increase $B_{nq}$ to 4 bits. We find that we match (and even exceed) the unquantized benchmark at single bit precision, which we attribute to the dominating low rank factorization, and its regularizing effect on data under extreme rank constraints. It is important to note that performance parity with direct-SVD quant. is also a successful outcome, since LPLR provides runtime improvements over taking an SVD to compress the data.

Table 4: **CIFAR10 embeddings generated by MobileNetV3 with an unquantized accuracy and F1 score** 91%:Results on LPLR and LPLR-SVD with $B = B' = 8$ bits [1]

| | Frobenius Norm Error | | | | Accuracy (%) | | | | Weighted F1 Score (%) | | | |
|---|---|---|---|---|---|---|---|---|---|---|---|---|
| $B_{nq}$ | **LPLR** | **LSVD** | **DSVD** | **NQ** | **LPLR** | **LSVD** | **DSVD** | **NQ** | **LPLR** | **LSVD** | **DSVD** | **NQ** |
| 1 | **1.05** | 1.08 | 1.09 | 7.17 | **92** | **92** | **92** | 11 | **92** | **92** | **92** | 4 |
| 2 | **1.08** | 1.1 | 1.1 | 2.29 | **92** | **92** | 91 | 30 | **92** | **92** | 91 | 23 |
| 4 | **1.1** | 1.11 | 1.11 | 1.15 | 91 | **92** | 91 | 91 | 91 | **92** | 91 | 91 |

Table 5: **CIFAR100 embeddings generated by MobileNetV3 with an unquantized accuracy and F1 score** 76%:Results on LPLR and LPLR-SVD with $B = B' = 8$ bits

| | Frobenius Norm Error | | | | Accuracy (%) | | | | Weighted F1 Score (%) | | | |
|---|---|---|---|---|---|---|---|---|---|---|---|---|
| $B_{nq}$ | **LPLR** | **LSVD** | **DSVD** | **NQ** | **LPLR** | **LSVD** | **DSVD** | **NQ** | **LPLR** | **LSVD** | **DSVD** | **NQ** |
| 1 | **1.04** | 1.08 | 1.09 | 6.75 | 79 | **82** | **82** | 1 | 79 | **82** | **82** | 0 |
| 2 | **1.08** | 1.1 | 1.12 | 2.18 | **80** | **80** | **80** | 1.7 | **80** | **80** | **80** | 1.3 |
| 4 | **1.11** | 1.12 | 1.14 | 1.17 | **79** | 78 | 77 | 75 | **79** | 78 | 78 | 75 |

Table 6: **IMDB embeddings generated by BERT with an unquantized accuracy and F1 score** 75% **and** 74% **respectively**: Results on LPLR and LPLR-SVD with $B = B' = 8$ bits

| | Frobenius Norm Error | | | | Accuracy (%) | | | | Weighted F1 Score (%) | | | |
|---|---|---|---|---|---|---|---|---|---|---|---|---|
| $B_{nq}$ | **LPLR** | **LSVD** | **DSVD** | **NQ** | **LPLR** | **LSVD** | **DSVD** | **NQ** | **LPLR** | **LSVD** | **DSVD** | **NQ** |
| 1 | 0.313 | **0.241** | 0.229 | 6.63 | 73 | 74 | **75** | 50 | 74 | 74 | **75** | 33 |
| 2 | 0.235 | 0.178 | **0.161** | 1.016 | **74** | **74** | **74** | 50 | **74** | **74** | **74** | 50 |
| 4 | 0.148 | 0.122 | **0.098** | 0.417 | **75** | 74 | **75** | 73 | 74 | 74 | **75** | 73 |

Table 7: **Emotion embeddings generated by BERT with an unquantized accuracy and F1 score** 43% **and** 40% **respectively**: Results on LPLR and LPLR-SVD with $B = B' = 8$ bits

| | Frobenius Norm Error | | | | Accuracy (%) | | | | Weighted F1 Score (%) | | | |
|---|---|---|---|---|---|---|---|---|---|---|---|---|
| $B_{nq}$ | **LPLR** | **LSVD** | **DSVD** | **NQ** | **LPLR** | **LSVD** | **DSVD** | **NQ** | **LPLR** | **LSVD** | **DSVD** | **NQ** |
| 1 | 0.383 | **0.296** | 0.286 | 5.109 | 41 | 43 | **43** | 29 | 38 | 40 | **40** | 13 |
| 2 | 0.291 | 0.215 | **0.202** | 1.561 | **42** | 43 | **43** | 33 | **39** | 40 | **40** | 30 |
| 4 | 0.187 | 0.137 | **0.121** | 0.367 | **42** | 43 | **43** | 41 | 39 | 40 | **40** | 38 |

### 4.3.2 Compressing Weight Matrices of a Large Language Model

In this section, we present results on a major application of matrix compression – compressing the weight matrices of deep neural networks. We choose LlaMa by Touvron et al. [66], a popular foundation Large Language Model (LLM) as the network of our choice. LLMs are a natural candidate for matrix compression, due to their massive stacked transformer layers, rendering them difficult to deploy on several GPUs, let alone a single GPU. Many methods have emerged to quantize and compress these models in order to make them amenable to single GPU deployment and inference, including naive quantization with outlier exclusion (Dettmers et al. [14]), second order methods (Frantar et al. [19]), low rank parameter reduction (Hu et al. [25]), amongst others.

We apply LPLR to the 2-dimensional weight tensors, i.e., matrices in LlaMa, leaving any other tensor, which does not lend itself to a low rank decomposition, unquantized. Figs. 2a and 2b (better

---

[1]LPLR-SVD used in the simulations computes the left low-rank factor $\mathbf{L}$ as $Q(\mathbf{U}_k \mathbf{S})$ where $S_{ij} \sim \mathcal{N}(0, 1/m)$ instead of simply $\mathbf{U}_k$ in order to exploit the equalization property of Gaussian embeddings.

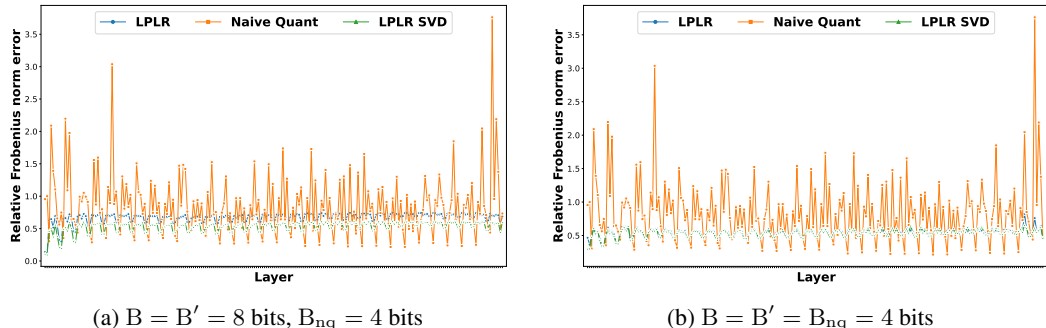

(a) $B = B' = 8$ bits, $B_{nq} = 4$ bits

(b) $B = B' = B_{nq} = 4$ bits

Figure 2: Comparison of LPLR and LPLR SVD on LlaMa weights, ordered by the original sequence of layers

Table 8: Average relative Frobenius norm error on LlaMa weight matrices

| | $B = B' = 8$ bits, $B_{nq} = 4$ bits | | | | $B = B' = B_{nq} = 4$ bits | | |
|---------|------|----------|-------------|---------|------|----------|-------------|
| Metric | LPLR | LPLR-SVD | Naive Quant. | Metric | LPLR | LPLR-SVD | Naive Quant. |
| Mean | 0.672 | **0.537** | 0.836 | Mean | 0.548 | **0.540** | 0.836 |
| Std Dev | 0.080 | **0.079** | 0.470 | Std Dev | **0.053** | 0.055 | 0.470 |

resolution in App. J) showcase our results on applying LPLR and LPLR-SVD with bit budgets of 8 bits and 4 bits respectively, using relative Frobenius norm error as the metric. While it is clear that LPLR and LPLR-SVD perform significantly better across all layers (on average), there are outliers where naïve quant. is the better choice. We can a observe periodic structure in the error profile of naïve quant., implying that the low rank structure is a function of the index of attention layer in transformer blocks. It is important to note that a low Frobenius norm error is not a direct indicator of performance for other task specific metrics. It is possible to construct a holistic compression strategy using error profiles similar to Figs. 3 and 4 to adopt a per-layer quantization strategy, minimizing both task specific metrics as well as relative Frobenius norm error. We discuss this further in Appendix K.

## 5 Conclusions

In this work, we have considered the problem of obtaining a low-precision and low-rank factorization of a matrix. Such a factorization of a matrix into a product of tall and wide matrices has several advantages, including compression of the original matrix. We propose a fast randomized algorithm to obtain this factorization which requires $O(nmd)$ computations – considerably faster than alternative methods. Our algorithm employs a Gaussian sketch to estimate the range space of matrices that are approximately low-rank. By utilizing the properties of subspace approximation and equalization in Gaussian embeddings, we establish an upper bound on the approximation error attained by our algorithm, and show that it can be significantly smaller than its counterparts. Finally, we empirically evaluate our method on several vision and text datasets, where we show significant task performance at highly compressed bit budgets as low as a *single* bit. This provides a novel pragmatic approach to work with large datasets and models in real world settings, making them more accessible to researchers and deployment on regular consumer hardware.

## Acknowledgments and Disclosure of Funding

This work was supported in part by the Air Force Office of Scientific Research (AFOSR) under Award #002484665; in part by National Science Foundation (NSF) CAREER Award under Grant CCF-2236829, Grant DMS-2134248 and Grant ECCS-2037304; in part by the U.S. Army Research Office Early Career Award under Grant W911NF-21-1-0242; in part by the Stanford Precourt Institute; and in part by the ACCESS AI Chip Center for Emerging Smart Systems through InnoHK, Hong Kong, SAR. The authors would also like to thank the anonymous reviewers whose comments and suggestions helped improve the presentation of this work.

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

# Contents

# A  Notations

We first gather some common notations in linear algebra and probability theory that have been used throughout the paper. Boldface upper and lowercase letters, $\mathbf{A}$ and $\mathbf{a}$ denote matrices and vectors respectively. $\mathbf{I}_d$ denotes the $d \times d$ identity matrix. The subscript may be dropped if the dimension is clear from the context. For any matrix $\mathbf{A}$, its $i^{\text{th}}$ row and $j^{\text{th}}$ column are denoted by $\mathbf{a}^{(i)}$ and $\mathbf{a}_j$ respectively. The singular values of $\mathbf{A}$ are denoted by $\sigma_{\max}(\mathbf{A}) = \sigma_1 \geq \sigma_2 \geq \ldots \geq \sigma_r = \sigma_{\min}(\mathbf{A})$, where $r = \text{rank}(\mathbf{A})$. Similarly, the eigenvalues are denoted as $\lambda_1(\mathbf{A}), \ldots, \lambda_r(\mathbf{A})$. The max-norm of $\mathbf{A}$ is defined as $\|A\|_{\max} = \max_{i,j}|A_{ij}|$, the spectral norm of $\mathbf{A}$ is defined as $\|\mathbf{A}\|_2 = \sup_{\|\mathbf{x}\|=1} \|\mathbf{A}\mathbf{x}\| = \sigma_{\max}(\mathbf{A})$, and the Frobenius norm is $\|\mathbf{A}\|_{\text{F}} = \left(\sum_{i,j} A_{ij}^2\right)^{1/2} = \text{Tr}\left[\mathbf{A}^\top \mathbf{A}\right] = \left(\sum_{k\in[r]} \sigma_k^2\right)^{1/2}$. For any vector $\mathbf{x}$, $\|\mathbf{x}\| = \left(x_i^2\right)^{1/2}$ denotes the $\ell_2$-norm, and $\|\mathbf{x}\|_\infty = \max_i |x_i|$ denotes the $\ell_\infty$-norm. We use the notations $\succcurlyeq$ and $\preccurlyeq$ for the positive semi-definite (PSD) cone ordering (or the *Loewner ordering*) of symmetric matrices, i.e., for any symmetric matrices $\mathbf{X}$ and $\mathbf{Y}$, $\mathbf{X} \succcurlyeq \mathbf{Y} \iff \mathbf{X} - \mathbf{Y}$ is PSD. $\mathbf{X}^\dagger$ denotes the Moore-Penrose pseudo-inverse of a matrix $\mathbf{X}$. $\log(\cdot)$ denotes the natural logarithm, i.e., base $e$. $\log_2(\cdot)$, i.e., with base 2 is specified explicitly. A normal distribution with mean $\mu$ and variance $\sigma^2$ is denoted as $\mathcal{N}(0, \sigma^2)$, while a multivariate normal distribution in $\mathbb{R}^d$ is denoted as $\mathcal{N}(\boldsymbol{\mu}, \boldsymbol{\Sigma})$. $\mathbb{E}[\cdot]$ denotes expectation of a random variable. The probability measure over which the expectation is taken is described in text. We also use big-'oh' notation $\text{O}(\cdot)$ that hides constants for asymptotic expressions, while $\widetilde{\text{O}}(\cdot)$ also hides terms that depends logarithmically on dimension.

We now list some notations that are either less commonly known or used specifically in this paper, along with some remarks such as their occurrence in the paper. Several of these notations have also been introduced in-context, but they are additionally collected here for easy reference.

Table 9: Notations used in this paper

| Notation | Description | Remarks |
|---|---|---|
| $\mathbf{A}, \mathbf{L}, \mathbf{R}$ | Input matrix, left and right LPLR factors | $\mathbf{LR}$ is an approximation of $\mathbf{A}$. Entries of $\mathbf{L}$ and $\mathbf{R}$ are represented in low-precision formats |
| $n, d$ | Dimensions of input matrix | $\mathbf{A} \in \mathbb{R}^{n \times d}$. |
| $k$ | Target rank | Often, $k \ll \min\{n, d\}$, but not necessarily. |
| $\mathbf{A} = \mathbf{U}\boldsymbol{\Sigma}\mathbf{V}^\top$ | Full SVD of the matrix $\mathbf{A} \in \mathbb{R}^{n \times d}$ | $\mathbf{U} \in \mathbb{R}^{n \times n}$, $\boldsymbol{\Sigma} \in \mathbb{R}^{n \times d}$, and $\mathbf{V} \in \mathbb{R}^{d \times d}$. |
| $\mathbf{A}_k = \mathbf{U}_k \boldsymbol{\Sigma}_k \mathbf{V}_k^\top$ | Best rank-$k$ approximation of $\mathbf{A}$ | $\mathbf{U} \in \mathbb{R}^{n \times k}$ and $\mathbf{V} \in \mathbb{R}^{n \times k}$ consists of top-$k$ left and right singular vectors respectively. $\boldsymbol{\Sigma} \in \mathbb{R}^{k \times k}$ consists of top-$k$ singular values. |
| $m$ | Sketch size | $m = k + p$, where $k$ is the target rank, and $p$ is the oversampling factor. |
| $\mathbf{S} \in \mathbb{R}^{d \times m}$ | Sketching matrix | In this work, $S_{ij} \sim \mathcal{N}\left(0, \frac{1}{m}\right)$. |
| $\text{Q}, \text{Q}'$ | Quantizers for the first and second low-rank factors with bit-budgets $\text{B}$ and $\text{B}'$, and dynamic ranges $\text{R}_{\text{Q}}$ and $\text{R}_{\text{Q}'}$ respectively. | — |
| $\kappa(\mathbf{A})$ | Condition number of input matrix $\mathbf{A}$ | $\kappa(\mathbf{A}) = \sigma_1, \sigma_r$ |
| $\kappa(\mathbf{A}_k)$ | Condition number of best rank-$k$ approximation $\mathbf{A}_k$ | $\kappa(\mathbf{A}_k) = \sigma_1/\sigma_k$ |
| $\kappa$ | A quantity that depends on the spectrum of $\mathbf{A}$ | $\kappa = \min\left\{\kappa(\mathbf{A}), \frac{\kappa(\mathbf{A}_k)}{(1-\text{O}(1)\sigma_{k+1}/\sigma_k)}\right\}$, defined in Thm. G.2 |
| $\gamma$ | Aspect ratio of the sketch | $\gamma = \frac{d}{m}$, defined in Thm. G.2. |
| $\delta$ | Target rank and sketch-size dependent quantity | $\delta = \frac{k}{m-k-1}$ |
| $\mathcal{W}(\boldsymbol{\Psi}, d)$ | Wishart distribution with covariance matrix $\boldsymbol{\Psi}$ and degree of freedom $d$ | Refer to §B.2.2. |

# B  Preliminaries

## B.1  Linear algebra inequalities

We state (with proofs) some standard inequalities from linear algebra that will be useful in proving our main results.

**Lemma B.1. (Frobenius norm of matrix products)** *For any matrices $\mathbf{A}$ and $\mathbf{B}$, we have*
$$\|\mathbf{AB}\|_{\mathrm{F}} \leq \|\mathbf{A}\|_2 \|\mathbf{B}\|_{\mathrm{F}}.$$

*Proof.* Note that:
$$\|\mathbf{AB}\|_{\mathrm{F}}^2 = \sum_j \|(\mathbf{AB})_j\|_2^2 \leq \sum_j \|\mathbf{AB}_j\|_2^2 \overset{(i)}{\leq} \|\mathbf{A}\|_2^2 \sum_j \|\mathbf{B}_j\|_2^2 = \|\mathbf{A}\|_2^2 \|\mathbf{B}\|_{\mathrm{F}}^2. \tag{5}$$
Here, (i) follows from the definition of spectral norm of a matrix, and this completes the proof. $\square$

**Lemma B.2. (Loewner ordering for matrix products)** *For any matrix $\mathbf{A}$ and $\mathbf{B}$, we have*
$$\sigma_{\min}^2(\mathbf{A}) \, \mathbf{B}^\top \mathbf{B} \preccurlyeq \mathbf{B}^\top \mathbf{A}^\top \mathbf{A} \mathbf{B} \preccurlyeq \sigma_{\max}^2(\mathbf{A}) \, \mathbf{B}^\top \mathbf{B}.$$

*Proof.* For any vector $\mathbf{x} \neq \mathbf{0}$, we have,
$$\mathbf{x}^\top \mathbf{B}^\top \mathbf{A}^\top \mathbf{A} \mathbf{B} \mathbf{x} = \|\mathbf{AB}\mathbf{x}\|_2^2 \geq \sigma_{\min}^2(\mathbf{A}) \|\mathbf{Bx}\|_2^2 = \mathbf{x}^\top \left(\sigma_{\min}^2(\mathbf{A}) \, \mathbf{B}^\top \mathbf{B}\right) \mathbf{x}. \tag{6}$$
The other direction holds similarly. This completes the proof. $\square$

**Lemma B.3. (Max-norm spectral-norm inequality)** *For any matrix $\mathbf{A} \in \mathbb{R}^{n \times d}$, $\|\mathbf{A}\|_{\max} \leq \|\mathbf{A}\|_2$.*

*Proof.* Let $\mathbf{e}_i \in \mathbb{R}^n$ and $\widehat{\mathbf{e}}_j \in \mathbb{R}^d$ denote the $i^{\text{th}}$ and $j^{\text{th}}$ canonical basis vectors in $\mathbb{R}^n$ and $\mathbb{R}^d$ respectively. Then, using Cauchy-Schwarz inequality, we have,
$$\|\mathbf{A}\|_{\max} = \max_{i,j} \left|\mathbf{e}_i^\top \mathbf{A} \widehat{\mathbf{e}}_j\right| \leq \max_{i,j} \|\mathbf{e}_i\| \|\mathbf{A}\widehat{\mathbf{e}}_j\| = \max_j \|\mathbf{A}\widehat{\mathbf{e}}_j\| \leq \sup_{\|\mathbf{x}\|=1} \|\mathbf{Ax}\| = \|\mathbf{A}\|_2, \tag{7}$$
completing the proof. $\square$

**Lemma B.4. (Submultiplicativity of spectral norm)** *For any two matrices $\mathbf{A}$ and $\mathbf{B}$, we have*
$$\|\mathbf{AB}\|_2 \leq \|\mathbf{A}\|_2 \|\mathbf{B}\|_2.$$
*Moreover, an analogous "reverse-submultiplicativity" result for the minimum singular value of a matrix product also holds true, i.e.,*
$$\sigma_{\min}(\mathbf{AB}) \geq \sigma_{\min}(\mathbf{A})\sigma_{\min}(\mathbf{B}).$$

*Proof.* Using Lemma B.2, we have,
$$\|\mathbf{AB}\|_2^2 = \lambda_{\max}\left(\mathbf{B}^\top \mathbf{A}^\top \mathbf{A} \mathbf{B}\right) \overset{(i)}{\leq} \|\mathbf{A}\|_2^2 \cdot \lambda_{\max}\left(\mathbf{B}^\top \mathbf{B}\right) = \|\mathbf{A}\|_2^2 \|\mathbf{B}\|_2^2, \tag{8}$$
where (i) follows from Lemma B.2, completing the proof. The lower bound on $\sigma_{\min}(\mathbf{AB})$ also follows a similar argument. $\square$

**Lemma B.5. (Lower bound on minimum singular value of matrix sums)** *For any matrices $\mathbf{A}$ and $\mathbf{B}$, we have*
$$\sigma_{\min}(\mathbf{A} + \mathbf{B}) \geq \sigma_{\min}(\mathbf{A}) - \|\mathbf{B}\|_2.$$

*Proof.* We have the following chain of inequalities,
$$\sigma_{\min}(\mathbf{A} + \mathbf{B}) = \inf_{\|\mathbf{x}\|=1} \|(\mathbf{A} + \mathbf{B})\mathbf{x}\| \overset{(i)}{\geq} \inf_{\|\mathbf{x}\|=1} (\|\mathbf{Ax}\| - \|\mathbf{Bx}\|)$$
$$\geq \inf_{\|\mathbf{x}\|=1} \|\mathbf{Ax}\| - \sup_{\|\mathbf{x}\|=1} \|\mathbf{Bx}\| = \sigma_{\min}(\mathbf{A}) - \|\mathbf{B}\|_2 .,$$
where (i) is the reverse triangle inequality. This completes the proof. $\square$

**Lemma B.6. (Rotation invariance of singular values)** *For a given matrix $\mathbf{A}$, $\sigma_i(\mathbf{A}) = \sigma_i(\mathbf{UA})$ for all $i = 1, \ldots, \operatorname{rank}(\mathbf{A})$ for any unitary matrix $\mathbf{U}$.*

*Proof.* For $i = 1, \ldots, \mathrm{rank}(\mathbf{A})$, since $\mathbf{U}^\top \mathbf{U} = \mathbf{I}$, we have

$$\sigma_i(\mathbf{UA}) = \sqrt{\lambda_i\left(\mathbf{A}^\top \mathbf{U}^\top \mathbf{UA}\right)} = \sqrt{\lambda_i\left(\mathbf{A}^\top \mathbf{A}\right)} = \sigma_i(\mathbf{A}),$$

completing the proof. $\qquad\square$

**Lemma B.7. (Rotation invariance of Frobenius norm)** *For a given matrix* $\mathbf{A}$*,* $\|\mathbf{UA}\|_{\mathrm{F}} = \|\mathbf{A}\|_{\mathrm{F}}$ *for any unitary matrix* $\mathbf{U}$*.*

*Proof.* We have,

$$\|\mathbf{UA}\|_{\mathrm{F}}^2 = \mathrm{Tr}\left[\mathbf{A}^\top \mathbf{U}^\top \mathbf{UA}\right] = \mathrm{Tr}\left[\mathbf{A}^\top \mathbf{A}\right] = \|\mathbf{A}\|_{\mathrm{F}}^2. \tag{9}$$

$\square$

## B.2 Probability and random matrix theory

We restate some results from probability and random matrix theory which will be useful in deriving the main result of our paper.

### B.2.1 Tail bound for Gaussian distribution

Gaussian distributions have strong concentration properties which we exploit in deriving the results of this paper. The following tail bound on a Gaussian random variable can be found in standard texts such as Wainwright [75, §2.1.2], and is restated here.

**Lemma B.8. (Chernoff bound for centered Gaussian)** *For* $X \sim \mathcal{N}\left(0, \sigma^2\right)$*, we have,*

$$\Pr\left(|X| \geq t\right) \leq 2e^{-\frac{t^2}{2\sigma^2}}. \tag{10}$$

The proof of this follows from a direct application of Chernoff bound.

### B.2.2 Inverse Wishart distribution

Consider a matrix $\mathbf{S} \in \mathbb{R}^{d \times m}$, each row of which is drawn independently from the distribution $\mathcal{N}\left(\mathbf{0}, \boldsymbol{\Psi}\right)$, where $\boldsymbol{\Psi} \in \mathbb{R}^{m \times m}$. Then, the probability distribution of the $m \times m$ random matrix $\mathbf{S}^\top \mathbf{S}$ is called the *Wishart distribution* with $d$ degrees of freedom, denoted as $\mathcal{W}\left(\boldsymbol{\Psi}, d\right)$. Moreover, the distribution of the matrix $\left(\mathbf{S}^\top \mathbf{S}^\top\right)^{-1}$ is called the *inverse Wishart distribution* and is denoted by $\mathcal{W}^{-1}\left(\boldsymbol{\Psi}^{-1}, d\right)$. These distributions have been studied extensively and further details can be found in Mardia et al. [40] or Siskind [59].

**Lemma B.9. (Expected trace of inverse Wishart matrix)** *Suppose* $\mathbf{S} \in \mathbb{R}^{d \times m}$ *matrix, each entry of which is drawn independently from* $\mathcal{N}\left(0, \frac{1}{m}\right)$*. Then, the matrix* $\mathbf{S}^\top \mathbf{S} \in \mathbb{R}^{m \times m}$ *follows the Wishart distribution* $\mathcal{W}\left(\frac{1}{m}\mathbf{I}_m, d\right)$ *that satisfies,*

$$\mathbb{E}\,\mathrm{Tr}\left[\left(\mathbf{S}^\top \mathbf{S}\right)^{-1}\right] = \frac{m^2}{d - m - 1}. \tag{11}$$

*Proof.* From Mardia et al. [40, eq. 3.8.3], if $\mathbf{X} \sim \mathcal{W}^{-1}\left(\boldsymbol{\Psi}^{-1}, d\right)$, then,

$$\mathbb{E}\left[\left(\mathbf{S}^\top \mathbf{S}\right)^{-1}\right] = \frac{\boldsymbol{\Psi}^{-1}}{d - m - 1}. \tag{12}$$

Here, $\boldsymbol{\Psi} = \frac{1}{m}\mathbf{I}_m$, implying $\mathrm{Tr}\left[\boldsymbol{\Psi}^{-1}\right] = m^2$, which completes the proof. $\qquad\square$

### B.2.3 Random Gaussian matrices

The spectral norm of random matrices with Gaussian entries have interesting concentration properties. In this section, we present a lemma from Vershynin [73] that formally states this result.

**Lemma B.10.** *Let the entries of matrix* $\mathbf{S} \in \mathbb{R}^{d \times m}$ *be distributed according to* $S_{ij} \sim \mathcal{N}\left(0, \frac{1}{m}\right)$*. Then for every* $t \geq 0$*, with probability at least* $1 - 2e^{-\frac{mt^2}{2}}$*, we have,*

$$\sqrt{\frac{d}{m}} - 1 - t \leq \sigma_{\min}(\mathbf{S}) \leq \sigma_{\max}(\mathbf{S}) \leq \sqrt{\frac{d}{m}} + 1 + t. \tag{13}$$

The above lemma is a straightforward modification of the result in Vershynin [73, Corr. 5.35], which states the concentration result when the entries are distributed according to $\mathcal{N}(0, 1)$. Note that given $\mathbf{S}$ with entries $S_{ij} \sim \mathcal{N}\left(0, \frac{1}{m}\right)$ as above, the matrix $\widehat{\mathbf{S}} = \sqrt{m}\,\mathbf{S}$ will have entries $\widehat{S}_{ij} \sim \mathcal{N}(0, 1)$. Furthermore, $\sigma_i(\widehat{\mathbf{S}}) = \sqrt{m}\,\sigma_i(\mathbf{S})$ and the conclusion is immediate.

### B.2.4  Subgaussian random variables

Subgaussian random variables refer to a class of distributions that are dominated by the distribution of a centered Gaussian random variable. There are several equivalent ways to characterize subgaussian random variables which can be found in several textbooks (see for example, Vershynin [74, Prop. 2.5.2]). We will focus on the following definition. More formally, the distribution of a random variable $\mathbf{X}$ is subgaussian if the moment generating function of $X^2$ is bounded at some point, i.e.,

$$\mathbb{E}\left[e^{X^2/K^2}\right] \leq 2. \tag{14}$$

**Definition B.11. (Subgaussian norm)** The subgaussian norm of a subgaussian random variable $X$, denoted by $\|X\|_{\psi_2}$ is defined to be the smallest $K$ in (14). In other words,

$$\|X\|_{\psi_2} \triangleq \inf\left\{t \geq 0 \mid \mathbb{E}\left[e^{X^2/t^2}\right] \leq 2\right\}. \tag{15}$$

It can be shown that any bounded random variable $X$ is subgaussian, and satisfies,

$$\|X\|_{\psi_2} \leq \frac{\|X\|_\infty}{\log 2}. \tag{16}$$

We next present a result that upper bounds the spectral norm of a matrix with subgaussian entries.

**Lemma B.12.** *(Vershynin [74, Thm 4.4.5]) Let $\mathbf{X}$ be a $d \times m$ random matrix whose entries $X_{ij}$ are independent, zero-mean, subgaussian random variables. Then, for any $t > 0$, we have,*

$$\|\mathbf{X}\|_2 \leq CK\left(\sqrt{d} + \sqrt{m} + t\right)$$

*with probability exceeding $1 - 2e^{-t^2}$. Here, $K = \max_{i,j}\|A_{ij}\|_{\psi_2}$ and $C$ is an absolute constant.*

## C  Quantization error of uniformly dithered scalar quantizer

For a scalar $x \in [-\mathrm{R}, +\mathrm{R}]$, let us denote the quantization error of uniformly dithered scalar quantizer with a bit-budget of B bits as $\epsilon = \mathrm{Q}_{\mathrm{R,B}}(x) - x$. Clearly, the quantization error is bounded as $|\epsilon| \leq \Delta$. The following result further characterizes its mean and variance of this error.

**Lemma C.1.** *The uniformly dithered scalar quantizer as described in (1) satisfies,*

$$\mathbb{E}\left[\epsilon\right] = 0 \quad \text{and} \quad Var\left(\epsilon\right) \leq \frac{\mathrm{R}^2}{\left(2^{\mathrm{B}} - 1\right)^2},$$

*where the $\mathbb{E}(\cdot)$ is over the randomness due to dithering in the quantizer operation.*

*Proof.* Suppose $x \in [q_k, q_{k+1})$ and $q_{k+1} = q_k + \Delta$, where $\Delta = \frac{2\mathrm{R}}{2^{\mathrm{B}}-1}$. Then,

$$\mathbb{E}\,\mathrm{Q}_{\mathrm{R,B}}(x) = q_{k+1}\,\frac{x - q_k}{\Delta} + q_k\left(1 - \frac{x - q_k}{\Delta}\right) = \frac{(q_k + \Delta)(x - q_k) + q_k(\Delta - x + q_k)}{\Delta} = x.$$

To evaluate the variance,

$$\begin{aligned}
\mathrm{Var}\left(\mathrm{Q}_{\mathrm{R,B}}(x) - x\right)^2 &= (q_{k+1} - x)^2 \frac{(x - q_k)}{\Delta} + (q_k - x)^2\left(1 - \frac{x - q_k}{\Delta}\right) \\
&\leq (q_{k+1} - x)(x - q_k) \\
&\leq \sup_{x \in [q_k, q_{k+1})} (q_{k+1} - x)(x - q_k) \\
&= \left(q_{k+1} - \frac{q_k + q_{k+1}}{2}\right)\left(\frac{q_k + q_{k+1}}{2} - q_k\right) = \frac{\Delta^2}{4} = \frac{\mathrm{R}^2}{\left(2^{\mathrm{B}} - 1\right)^2}.
\end{aligned}$$

This completes the proof. $\qquad\qquad\square$

# D    Gaussian embeddings: Application of equalization to vector quantizers

In this section, we show a result on how Gaussian embeddings help in reducing the $\ell_2$-quantization error of a uniformly dithered vector quantizer. We consider a *clipped version* of the uniform scalar quantizer with bit-budget B described in §2.1 and App. C. In order to quantize a vector $\mathbf{x} \in \mathbb{R}^d$ with $\|\mathbf{x}\|_2 \leq \mathrm{R}$ using a uniform scalar quantizer (as described above), the quantization operation is applied to each coordinate of the vector independently, i.e.,

$$Q_{\mathrm{R}}(\mathbf{x}) = [Q_{\mathrm{R}}(x_1), \ldots, Q_{\mathrm{R}}(x_d)]. \tag{17}$$

Here, the subscript B is dropped because the bit-budget is evident from the context. Furthermore, since $\|\mathbf{x}\|_2 \leq \mathrm{R}$ implies $x_i \in [-\mathrm{R}, +\mathrm{R}]$ for every $i \in [d]$, the quantizer $Q_{\mathrm{R}}$ does not saturate. From Lemma C.1, the expected quantization error is given by,

$$\mathbb{E} \|Q_{\mathrm{R}}(\mathbf{x}) - \mathbf{x}\|_2^2 = \sum_{i \in [d]} \mathbb{E} \left[ (Q_{\mathrm{R}}(x_i) - x_i)^2 \right] \leq \frac{\mathrm{R}^2 d}{\left(2^{\mathrm{B}} - 1\right)^2}. \tag{18}$$

**Quantizing Gaussian embeddings**: Suppose instead of quantizing $\mathbf{x} \in \mathbb{R}^d$ directly, we quantize $\mathbf{u} = \mathbf{S}\mathbf{x} \in \mathbb{R}^m$, where $\mathbf{S} \in \mathbb{R}^{m \times d}$ with $S_{ij} \sim \mathcal{N}\left(0, \frac{1}{m}\right)$. Note that for every $j \in [m]$, we have $u_j \sim \mathcal{N}\left(0, \frac{1}{m}\|\mathbf{x}\|_2^2\right)$. Since $u_j$ can be anything in $(-\infty, +\infty)$, there is a finite probability that the uniform scalar quantizer might get saturated. For this reason, for any scalar $u \in (-\infty, +\infty)$, we define the *clipped uniformly dithered quantizer* with clipping parameter $t$ as follows:

$$Q(u) = \begin{cases} Q_t(u) & \text{if} \quad |u| \leq t \\ t & \text{if} \quad u > t \\ -t & \text{if} \quad u < -t. \end{cases} \tag{19}$$

The dynamic range of the quantizer is parameterized by $t$, which is to be chosen appropriately. Note that it is just for this section for the purposes of illustration, that we choose the clipped variant of the quantizer. In LPLR, we choose the dynamic range to be high enough so in practice it remains unsaturated with a very high probability. The following proposition upper bounds the quantization error of quantizing Gaussian embeddings.

**Proposition D.1.** *For a given vector $\mathbf{x} \in \mathbb{R}^d$ with $\|\mathbf{x}\| \leq \mathrm{R}$, the clipped uniformly dithered quantizer described in (19) with $t = \frac{\mathrm{R}}{\sqrt{m}}$ satisfies*

$$\mathbb{E} \|Q(\mathbf{S}\mathbf{x}) - \mathbf{S}\mathbf{x}\|_2^2 \leq \frac{\mathrm{R}^2}{\left(2^{\mathrm{B}} - 1\right)^2} + \frac{\mathrm{R}^2 \sqrt{2}}{\sqrt{\pi e}},$$

*where the expectation is over the randomness in the construction of $\mathbf{S}$ and the quantization dither.*

*Proof.* Since $\{u_i\}_{i \in [m]}$ are independently and identically distributed, let us denote the distribution as $f(u) = \frac{1}{\sigma\sqrt{2\pi}} e^{-\frac{u^2}{2\sigma^2}}$, where $\sigma = \frac{\|\mathbf{x}\|_2^2}{m}$. The expected quantization error for the clipped quantizer is then given by,

$$\mathbb{E}_{\mathbf{S}, Q_t} \left[ (Q(u) - u)_2^2 \right] = \int_{|u| \leq t} \mathbb{E}_{Q_t} \left[ (Q(u) - u)_2^2 \right] f(u) du + 2 \int_{u > t} (t - u)^2 f(u) du \tag{20}$$

Here, the expectation is over the stochasticity of the quantization dither, as well as the random matrix $\mathbf{S}$. The factor of 2 in the second term appears due to symmetry of clipping and that of Gaussian distribution. Using (18), the first term on the R.H.S. can be upper bounded as,

$$\int_{|u| \leq t} \mathbb{E}_{Q_t} \left[ (Q(u) - u)_2^2 \right] f(u) du \leq \frac{t^2}{\left(2^{\mathrm{B}} - 1\right)^2} \int_{|u| \leq t} f(u) du \leq \frac{t^2}{\left(2^{\mathrm{B}} - 1\right)^2}. \tag{21}$$

To analyze the second term, note that,

$$\frac{1}{\sigma\sqrt{2\pi}} \int_t^\infty (t - u)^2 e^{-\frac{u^2}{2\sigma^2}} du = \frac{1}{2} \left(\sigma^2 + t^2\right) \left(1 - \mathrm{erf}\left(\frac{t}{\sigma\sqrt{2}}\right)\right) - \frac{\sigma t}{\sqrt{2\pi}} e^{-\frac{t^2}{2\sigma^2}} \triangleq \Psi(t, \sigma), \tag{22}$$

where $\mathrm{erf}(z)$ denotes the error function defined as,

$$\mathrm{erf}(z) = \frac{2}{\sqrt{\pi}} \int_0^x e^{-x^2} dx \tag{23}$$

Simple calculation would show,

$$\frac{\partial \Psi(t,\sigma)}{\partial \sigma}$$

$$=\sigma \left(1 - \operatorname{erf}\left(\frac{t}{\sigma\sqrt{2}}\right)\right) - \frac{\sigma t}{\sqrt{2\pi}} e^{-\frac{t^2}{2\sigma^2}} + \frac{t}{\sqrt{2\pi}}\left(1 + \frac{t^2}{\sigma^2}\right) e^{-\frac{t^2}{2\sigma^2}} - \frac{t}{\sqrt{2\pi}} e^{-\frac{t^2}{2\sigma^2}} - \frac{t^3}{\sigma^2\sqrt{2\pi}} e^{-\frac{t^2}{2\sigma^2}}$$

$$=\sigma \left(1 - \operatorname{erf}\left(\frac{t}{\sigma\sqrt{2}}\right)\right) \geq 0. \tag{24}$$

Since $\frac{\partial \Psi(t,\sigma)}{\partial \sigma} \geq 0$, $\Psi(t,\sigma)$ is a non-decreasing function of $\sigma$. Since $\sigma^2 = \frac{\|\mathbf{x}\|_2^2}{m} \leq \frac{\mathrm{R}^2}{m}$, we have the upper bound,

$$2\int_{u>t} (t-u)^2 f(u)du \leq \left(\frac{\mathrm{R}^2}{m} + t^2\right)\left(1 - \operatorname{erf}\left(\frac{t\sqrt{m}}{\mathrm{R}\sqrt{2}}\right)\right) - \frac{\mathrm{R}t\sqrt{2}}{\sqrt{\pi m}} e^{-\frac{mt^2}{2\mathrm{R}^2}}$$

$$= \left(\frac{\mathrm{R}^2}{m} + t^2\right)\operatorname{erfc}\left(\frac{t\sqrt{m}}{\mathrm{R}\sqrt{2}}\right) - \frac{\mathrm{R}t\sqrt{2}}{\sqrt{\pi m}} e^{-\frac{mt^2}{2\mathrm{R}^2}}, \tag{25}$$

where $\operatorname{erfc}(z) = 1 - \operatorname{erf}(z)$ is the complementary error function.

**Quantization error of a scalar Gaussian sketch**: Since $\mathbf{x} \in \mathbb{R}^d$ with $\|\mathbf{x}\|_2 \leq \mathrm{R}$, let $\mathbf{s} \sim \mathcal{N}(\mathbf{0}, \frac{1}{m}\mathbf{I}_d)$. We first consider the quantization of $\mathbf{x}^\top \mathbf{s}$, and upper bound the error $\mathbb{E}\left[\left(Q(\mathbf{x}^\top \mathbf{s}) - \mathbf{x}^\top \mathbf{s}\right)_2^2\right]$. Clearly, $\mathbf{x}^\top \mathbf{s} \sim \mathcal{N}(0, \frac{\mathrm{R}^2}{m})$. Consequently, using (21) and (25), for any $t \geq 0$, we have,

$$\mathbb{E}\left[\left(Q(\mathbf{x}^\top \mathbf{s}) - \mathbf{x}^\top \mathbf{s}\right)_2^2\right] \leq \frac{t^2}{(2^{\mathrm{B}} - 1)^2} + \left(\frac{\mathrm{R}^2}{m} + t^2\right)\operatorname{erfc}\left(\frac{t\sqrt{m}}{\mathrm{R}\sqrt{2}}\right) - \frac{\mathrm{R}t\sqrt{2}}{\sqrt{\pi m}} e^{-\frac{mt^2}{2\mathrm{R}^2}}$$

$$\overset{(\mathrm{i})}{\leq} \frac{t^2}{(2^{\mathrm{B}} - 1)^2} + \left(\frac{\mathrm{R}^2}{m} + t^2\right)\frac{\mathrm{R}\sqrt{2}}{t\sqrt{\pi m}} e^{-\frac{mt^2}{2\mathrm{R}^2}} - \frac{\mathrm{R}t\sqrt{2}}{\sqrt{\pi m}} e^{-\frac{mt^2}{2\mathrm{R}^2}}$$

$$= \frac{t^2}{(2^{\mathrm{B}} - 1)^2} + \frac{\mathrm{R}^3\sqrt{2}}{m^{3/2}t\sqrt{\pi}} e^{-\frac{mt^2}{2\mathrm{R}^2}}. \tag{26}$$

Here, (i) follows from the upper bound $\operatorname{erfc}(z) \leq \frac{e^{-z^2}}{\sqrt{\pi}z}$ from Karagiannidis and Lioumpas [29].

**Quantization error of a vector Gaussian sketch**: We now consider for any $t \geq 0$, the expected quantization error for $\mathbf{x} \in \mathbb{R}^d$ with $\|\mathbf{x}\|_2 \leq \mathrm{R}$ and $\mathbf{S} \in \mathbb{R}^{m \times d}$ with $S_{ij} \sim \mathcal{N}\left(0, \frac{1}{m}\right)$. Each row of $\mathbf{S}$ now independently plays the role of $\mathbf{s}$ in (26). Using (26), the vector quantization error is simply $m$ times the scalar quantization error, and is now given by,

$$\mathbb{E}\|Q(\mathbf{S}\mathbf{x}) - \mathbf{S}\mathbf{x}\|_2^2 \leq \frac{mt^2}{(2^{\mathrm{B}} - 1)^2} + \frac{\mathrm{R}^3\sqrt{2}}{t\sqrt{\pi m}} e^{-\frac{mt^2}{2\mathrm{R}^2}}. \tag{27}$$

**Choice of dynamic range** $t$: Setting $t = \frac{\mathrm{R}}{\sqrt{m}}$ in (27) yields,

$$\mathbb{E}\|Q(\mathbf{S}\mathbf{x}) - \mathbf{S}\mathbf{x}\|_2^2 \leq \frac{\mathrm{R}^2}{(2^{\mathrm{B}} - 1)^2} + \frac{\mathrm{R}^2\sqrt{2}}{\sqrt{\pi e}}. \tag{28}$$

This completes the proof. $\qquad\square$

Note that since $\mathbb{E}[\mathbf{S}^\top \mathbf{S}]$ is an identity matrix, an estimate $\widehat{\mathbf{x}}$ of $\mathbf{x}$ can be recovered from $Q(\mathbf{S}\mathbf{x})$ as $\mathbf{S}^\top Q(\mathbf{S}\mathbf{x})$. We can use the $O(1)$ bound on $\|Q(\mathbf{S}\mathbf{x}) - \mathbf{S}\mathbf{x}\|^2$ to get a bound on $\|\mathbf{S}^\top Q(\mathbf{S}\mathbf{x}) - \mathbf{x}\|^2$ as follows:

$$\|\mathbf{S}^\top Q(\mathbf{S}\mathbf{x}) - \mathbf{x}\|^2 \leq \|Q(\mathbf{S}\mathbf{x}) - \mathbf{S}\mathbf{x}\|^2 + \|\mathbf{S}^\top Q(\mathbf{S}\mathbf{x})\|^2 - \|Q(\mathbf{S}\mathbf{x})\|^2 + \|\mathbf{x}\|^2 - \|\mathbf{S}\mathbf{x}\|^2$$

$$\leq \|Q(\mathbf{S}\mathbf{x}) - \mathbf{S}\mathbf{x}\|^2 + \|\mathbf{S}^\top Q(\mathbf{S}\mathbf{x})\|^2 + \|\mathbf{x}\|^2$$

$$\leq \|Q(\mathbf{S}\mathbf{x}) - \mathbf{S}\mathbf{x}\|^2 + \mathrm{R}^2(\sigma_{\max}^2(\mathbf{S}) + 1) \tag{29}$$

From properties of random Gaussian matrices, we know that $\sigma_{\max}^2(\mathbf{S}) \leq \frac{d}{m}$ with high probability. Hence, the error $\|\mathbf{S}^\top Q(\mathbf{S}\mathbf{x}) - \mathbf{x}\|^2$ only depends on the aspect ratio $d/m$, and not the dimension $d$ directly. Although the reconstruction error $\|\mathbf{S}^\top Q(\mathbf{S}\mathbf{x}) - x\|^2$ scales as $d/m$, it does not necessarily increase with $d$ if we choose the sketch size $m$ to be proportional to $d$, i.e., $m = O(d)$.

Despite this, Gaussian embeddings are not used practically for vector quantization because $\mathbf{S}$ is dense matrix and computing $\mathbf{Sx}$ entails a complexity of $O(d^2)$. The entries of $\mathbf{S}$ themselves are floating point numbers that have to be stored in full precision, hence this defeats the whole purpose of quantizing $\mathbf{x}$ using fewer bits. However, this is not an issue for matrix compression because in LPLR, we do not explicitly compute the $\mathbf{S}^\top Q(\mathbf{SA})$ anywhere. In other words, the effects of $\mathbf{S}$ in the first low-rank factor is nullified by the second low-rank factor. Unlike vector quantization, the corresponding sketch size $m$ for LPLR only needs to be the same order as the inherent rank $k$, which can be much smaller than $\min\{n, d\}$, i.e., the dimensions of the matrix being compressed.

## E  Dynamic range of quantizers

We now derive high probability upper bounds on the maximum magnitude of the input to uniform quantizers $Q$ and $Q'$. Choosing these upper bounds to be the dynamic range of the uniform scalar quantizers will ensure that the quantizer remains unsaturated with a high probability, which is our desired regime of operation.

### E.1  Quantization for the first low-rank factor

We first look at the choice of dynamic range for the quantizer $Q$, which is used to obtain the first low-rank factor. The input to the quantizer is $\mathbf{AS} \in \mathbb{R}^{n \times m}$ where $\mathbf{A} \in \mathbb{R}^{n \times d}$ and $\mathbf{S} \in \mathbb{R}^{d \times m}$, and the entries of $\mathbf{S}$ are i.i.d. as $S_{ij} \sim \mathcal{N}\left(0, \frac{1}{m}\right)$. Lemma E.1 below gives a high probability upper bound on the max norm of $\mathbf{AS}$. This probability is computed over the randomness in the construction of $\mathbf{S}$.

**Lemma E.1.  (Max norm of AS)** *Given matrix* $\mathbf{A} \in \mathbb{R}^{n \times d}$ *with bounded row norms, i.e.,* $\left\|\mathbf{a}^{(i)}\right\| \leq \mathrm{R}$, *and* $\mathbf{S} \in \mathbb{R}^{d \times m}$ *with entries distributed as* $S_{ij} \overset{i.i.d.}{\sim} \mathcal{N}\left(0, \frac{1}{m}\right)$, *with probability exceeding* $1 - \frac{\epsilon}{8n\mathrm{R}^2}$, *the max norm of* $\mathbf{AS}$ *satisfies,*

$$\|\mathbf{AS}\|_{\max} \leq \mathrm{R}\sqrt{\frac{2\log\left(\frac{16\mathrm{R}^2 n^2 m}{\epsilon}\right)}{m}}. \tag{30}$$

*Proof.* Since $(\mathbf{AS})_{ij} = \mathbf{s}_j^\top \mathbf{a}^{(i)}$, where $\mathbf{a}^{(i)} \in \mathbb{R}^d$ is the $i^{\text{th}}$ row of $\mathbf{A}$ and $\mathbf{s}_j \in \mathbb{R}^d$ is the $j^{\text{th}}$ column of $\mathbf{S}$, we have $(\mathbf{AS})_{ij} \sim \mathcal{N}\left(0, \frac{\|\mathbf{a}_i\|_2^2}{m}\right)$. Using Lemma B.8 and an application of union bound gives,

$$\Pr\left(\left|(\mathbf{AS})_{ij}\right| \geq t\right) \leq 2e^{-\frac{mt^2}{2\left\|\mathbf{a}^{(i)}\right\|_2^2}} \leq 2e^{-\frac{mt^2}{2\mathrm{R}^2}}. \tag{31}$$

A subsequent application of union bound over all the entries of $\mathbf{AS}$ yields,

$$\Pr\left(\|\mathbf{AS}\|_{\max} \geq t\right) \leq 2nme^{-\frac{mt^2}{2\mathrm{R}^2}}. \tag{32}$$

Setting $t = \mathrm{R}\sqrt{\frac{2\log\left(\frac{16\mathrm{R}^2 n^2 m}{\epsilon}\right)}{m}}$ in the above completes the proof. $\qquad\square$

### E.2  Quantization for the second low-rank factor

We now obtain a high-probability upper bound on the max norm of $Q(\mathbf{AS})^\dagger \mathbf{A}$, which is the input to the second quantizer $Q'$. Let us define $\mathcal{Q}$ to be the event that the quantizer $Q$ does not saturate. From Lemma E.1, $\mathcal{Q}$ occurs with a sufficiently high probability. An appropriate choice of dynamic range for $Q'$ will ensure that conditioned on the event that $\mathcal{Q}$ occurs, the quantizer $Q'$ also does not saturate with a high probability. The following lemma states this formally.

**Lemma E.2.  (Max norm of** $Q(\mathbf{AS})^\dagger \mathbf{A}$**)** *Let our input matrix* $\mathbf{A} \in \mathbb{R}^{n \times d}$ *have non-zero singular values* $\sigma_1, \ldots, \sigma_k, \sigma_{k+1}, \ldots, \sigma_r$, *where* $r = \text{rank}(\mathbf{A})$, *and bounded row norms* $\left\|\mathbf{a}^{(i)}\right\| \leq \mathrm{R}$. *Let* $\kappa(\mathbf{A}) = \sigma_1/\sigma_r$ *and* $\kappa(\mathbf{A}_k) = \sigma_1/\sigma_k$ *respectively be the condition numbers of* $\mathbf{A}$ *and the best rank-$k$ approximation of* $\mathbf{A}$, *and for some small* $\epsilon > 0$, *let us denote*

$$t = \sqrt{\frac{2\log\left(\frac{32n\mathrm{R}^2}{\epsilon}\right)}{m}}, \quad \kappa = \min\left\{\kappa(\mathbf{A}), \frac{\kappa(\mathbf{A}_k)}{1 - \frac{\sigma_{k+1}}{\sigma_k}\left(\frac{\sqrt{\gamma}+1+t}{\sqrt{\gamma}-1-t}\right)}\right\}, \tag{33}$$

*where $\gamma = d/m$ is the aspect ratio of the sketching matrix $\mathbf{S} \in \mathbb{R}^{d \times m}$ with $S_{ij} \overset{i.i.d.}{\sim} \mathcal{N}\left(0, \frac{1}{m}\right)$. Furthermore, suppose the dynamic range of the quantizer $Q$ is set to $R\sqrt{\frac{2\log\left(\frac{16R^2 n^2 m}{\epsilon}\right)}{m}}$ as dictated by Lemma E.1, and suppose for some absolute constant $C$, the bit-budget $B$ satisfies,*

$$B \geq \log_2 \left( \frac{4CR}{\max\left\{\sigma_r, \sigma_k - \sigma_{k+1}\left(\frac{\sqrt{\gamma}+1+t}{\sqrt{\gamma}-1-t}\right)\right\}\log 2} \left(\frac{\sqrt{\gamma}+1+t/\sqrt{2}}{\sqrt{\gamma}-1-t}\right)\sqrt{2\log\left(\frac{16R^2 n^2 m}{\epsilon}\right)} + 1 \right). \tag{34}$$

*Then, we have,*

$$\left\|Q(\mathbf{AS})^\dagger \mathbf{A}\right\|_{\max} \leq \frac{2\kappa}{\sqrt{\gamma}-1-t} \quad \text{with probability exceeding } 1 - \frac{\epsilon}{4nR^2}. \tag{35}$$

*Proof.* We have the following chain of inequalities:

$$\left\|Q\left(\mathbf{AS}\right)^\dagger \mathbf{A}\right\|_{\max} \overset{(i)}{\leq} \left\|Q\left(\mathbf{AS}\right)^\dagger \mathbf{A}\right\|_2 \overset{(ii)}{\leq} \left\|Q\left(\mathbf{AS}\right)^\dagger\right\|_2 \sigma_1(\mathbf{A}), \tag{36}$$

where (i) and (ii) follow from Lemmas B.3 and B.4 respectively. We now need to upper bound $\left\|Q\left(\mathbf{AS}\right)^\dagger\right\|_2$, which is done as follows:

$$\left\|Q\left(\mathbf{AS}\right)^\dagger\right\|_2 \overset{(i)}{=} \left(\sigma_{\min}\left(Q\left(\mathbf{AS}\right)\right)\right)^{-1} = \left(\sigma_{\min}\left(\mathbf{AS} + \mathbf{E}\right)\right)^{-1} \overset{(ii)}{\leq} \left(\sigma_{\min}(\mathbf{AS}) - \|\mathbf{E}\|_2\right)^{-1} \tag{37}$$

Here, $\mathbf{E} = Q(\mathbf{AS}) - \mathbf{AS} \in \mathbb{R}^{n \times m}$ is the quantization error matrix from $Q$, (i) follows because the singular values of $Q\left(\mathbf{AS}\right)^\dagger$ are inverses of the singular values of $Q\left(\mathbf{AS}\right)$ (assuming $Q\left(\mathbf{AS}\right)$ is invertible), and (ii) follows from Lemma B.5.

**Lower bounding** $\sigma_{\min}(\mathbf{AS})$: It now suffices to derive the a lower bound on $\sigma_{\min}(\mathbf{AS})$, which we do next. We derive two different lower bounds, either of which could be tighter depending on the singular value profile of $\mathbf{A}$. The final lower bound will be the maximum of both.

For the first lower bound, let $\mathbf{A} = \mathbf{U}\boldsymbol{\Sigma}\mathbf{V}^\top$ be the full singular value decomposition of $\mathbf{A}$, where $\mathbf{U} \in \mathbb{R}^{n \times n}$, $\boldsymbol{\Sigma} \in \mathbb{R}^{n \times d}$, and $\mathbf{V} \in \mathbb{R}^{d \times d}$. Then, denoting $\widetilde{\mathbf{S}} = \mathbf{V}^\top \mathbf{S}$, we have,

$$\sigma_{\min}(\mathbf{AS}) = \sigma_{\min}\left(\mathbf{A}_k \mathbf{S} + (\mathbf{A} - \mathbf{A}_k)\mathbf{S}\right) \overset{(i)}{\geq} \sigma_{\min}(\mathbf{A}_k \mathbf{S}) - \left\|(\mathbf{A} - \mathbf{A}_k)\mathbf{S}\right\|_2$$
$$\overset{(ii)}{\geq} \sigma_{\min}(\mathbf{A}_k \mathbf{S}) - \sigma_{k+1}\|\mathbf{S}\|_2, \tag{38}$$

where $\mathbf{A}_k$ is the best rank-$k$ approximation of $\mathbf{A}$. Here, once again, (i) follows from Lemma B.5 and (ii) follows from Lemma B.4. In order to further lower bound $\sigma_{\min}(\mathbf{A}_k \mathbf{S})$, let us denote the SVD of $\mathbf{A}_k$ as $\mathbf{A}_k = \mathbf{U}\boldsymbol{\Sigma}_k \mathbf{V}_k^\top$. Since the best rank-$k$ approximation is obtained by retaining the top-$k$ singular values of $\mathbf{A}$ and zeroing out the rest, we have $\mathbf{U} \in \mathbb{R}^{n \times n}$, $\boldsymbol{\Sigma}_k \in \mathbb{R}^{n \times k}$ and $\mathbf{V}_k \in \mathbb{R}^{d \times k}$. Let us further denote $\widetilde{\mathbf{S}} = \mathbf{V}_k^\top \mathbf{S}$. Then we have

$$\sigma_{\min}\left(\mathbf{A}_k \mathbf{S}\right) = \sigma_{\min}\left(\mathbf{U}\boldsymbol{\Sigma}_k \mathbf{V}_k^\top \mathbf{S}\right) = \sigma_{\min}\left(\mathbf{U}\boldsymbol{\Sigma}_k \widetilde{\mathbf{S}}\right) \overset{(i)}{=} \sigma_{\min}\left(\boldsymbol{\Sigma}_k \widetilde{\mathbf{S}}\right) \overset{(ii)}{\geq} \sigma_k \sigma_{\min}(\widetilde{\mathbf{S}}), \tag{39}$$

where (i) follows from Lemma B.6, and (ii) follows from Lemma B.4. Note that the $(i,j)^{\text{th}}$ entry of $\widetilde{\mathbf{S}}$ is $\widetilde{S}_{ij} = \mathbf{v}_i^\top \mathbf{s}_j$, where $\mathbf{v}_i$ is the $i^{\text{th}}$ column of $\mathbf{V}_k$. Clearly, $\widetilde{S}_{ij}$ is a Gaussian random variable with mean, $\mathbb{E}\left[\widetilde{S}_{ij}\right] = \sum_\ell V_{\ell i} \mathbb{E}[S_{\ell j}] = 0$, and variance, $\text{Var}\left(\widetilde{S}_{ij}\right) = \sum_\ell V_{\ell i}^2 \text{Var}\left(S_{\ell j}\right) = \frac{1}{m}\sum_\ell V_{\ell i}^2 = \frac{1}{m}$, where the last equality follows from the fact that the columns of $\mathbf{V}_k$ are orthonormal. In other words, the entries of $\widetilde{\mathbf{S}} \in \mathbb{R}^{d \times m}$ are distributed according to $\widetilde{S}_{ij} \sim \mathcal{N}\left(0, \frac{1}{m}\right)$. Using (39), (38) can be further lower bounded as

$$\sigma_{\min}(\mathbf{AS}) \geq \sigma_k \sigma_{\min}(\widetilde{\mathbf{S}}) - \sigma_{k+1}\|\mathbf{S}\|_2 \overset{(i)}{=} \sigma_k \sigma_{\min}(\mathbf{S}) - \sigma_{k+1}\|\mathbf{S}\|_2, \tag{40}$$

where (i) once again follows from the rotation invariance of spectrum, i.e., Lemma B.6.

On the other hand, let $r = \text{rank}(\mathbf{A})$ and $\sigma_r$ by the smallest non-zero singular value of $\mathbf{A}$. Then, $\mathbf{A} = \mathbf{U}\boldsymbol{\Sigma}\mathbf{V}^\top = \mathbf{U}\boldsymbol{\Sigma}_r \mathbf{V}_r^\top$ and using the same arguments as in (39), we also have

$$\sigma_{\min}\left(\mathbf{AS}\right) = \sigma_{\min}\left(\mathbf{U}\boldsymbol{\Sigma}_r \mathbf{V}_r^\top \mathbf{S}\right) = \sigma_{\min}\left(\boldsymbol{\Sigma}_r \mathbf{V}_r^\top \mathbf{S}\right) \geq \sigma_r \sigma_{\min}\left(\mathbf{S}\right). \tag{41}$$

Combining (40) and (41), we get
$$\sigma_{\min}(\mathbf{AS}) \geq \max\{\sigma_r \sigma_{\min}(\mathbf{S}), \sigma_k \sigma_{\min}(\mathbf{S}) - \sigma_{k+1} \|\mathbf{S}\|_2\} \tag{42}$$

All we are left with now is to utilize the concentration bounds on the singular values of $\mathbf{S}$. As a consequence of Lemma B.10 and recalling $\gamma = d/m$, with probability exceeding $1 - 2e^{-\frac{mt^2}{2}}$,
$$\sqrt{\gamma} - 1 - t \leq \sigma_{\min}(\mathbf{S}) \leq \|\mathbf{S}\|_2 \leq \sqrt{\gamma} + 1 + t. \tag{43}$$

Substituting this in (42), with probability exceeding $1 - 2e^{-\frac{mt^2}{2}}$,
$$\sigma_{\min}(\mathbf{AS}) \geq \max\left\{\sigma_r\left(\sqrt{\gamma} - 1 - t\right), \sigma_k\left(\sqrt{\gamma} - 1 - t\right) - \sigma_{k+1}\left(\sqrt{\gamma} + 1 + t\right)\right\}. \tag{44}$$

**Upper bounding $\|\mathbf{E}\|_2$**: Finally, conditioned on the event $\mathcal{Q}$, the entries of $\mathbf{E}$ are bounded. Choosing the dynamic range of $\hat{Q}$ as dictated by the upper bound on $\|\mathbf{AS}\|_{\max}$ in Lemma E.1, we have, with probability exceeding $1 - \frac{\epsilon}{8n\mathrm{R}^2}$,
$$|E_{ij}| \leq \Delta = \frac{2\mathrm{R}}{(2^{\mathrm{B}} - 1)}\sqrt{\frac{2\log\left(\frac{16\mathrm{R}^2 n^2 m}{\epsilon}\right)}{m}} \quad \text{for all } i \in [n] \text{ and } j \in [m]. \tag{45}$$

Since $E_{ij}$ is a bounded w.h.p., it is also subgaussian w.h.p. (ref. eq. (16)) with subgaussian norm given by,
$$\|E_{ij}\|_{\psi_2} \leq \frac{\Delta}{\log 2} = \frac{2\mathrm{R}}{(\log 2)(2^{\mathrm{B}} - 1)}\sqrt{\frac{2\log\left(\frac{16\mathrm{R}^2 n^2 m}{\epsilon}\right)}{m}}. \tag{46}$$

From Lemma B.12 we get for some absolute constant $C$,
$$\|\mathbf{E}\|_2 \leq \frac{2C\mathrm{R}\left(\sqrt{d} + \sqrt{m} + \tilde{t}\right)}{(2^{\mathrm{B}} - 1)\log 2}\sqrt{\frac{2\log\left(\frac{16\mathrm{R}^2 n^2 m}{\epsilon}\right)}{m}} \quad \text{w.p. exceeding } 1 - \frac{\epsilon}{8n\mathrm{R}^2} - 2e^{-\tilde{t}^2}$$

$$\text{or, } \|\mathbf{E}\|_2 \leq \frac{2C\mathrm{R}\left(\sqrt{\gamma} + 1 + \tilde{t}\right)}{(2^{\mathrm{B}} - 1)\log 2}\sqrt{2\log\left(\frac{16\mathrm{R}^2 n^2 m}{\epsilon}\right)} \quad \text{w.p. exceeding } 1 - \frac{\epsilon}{8n\mathrm{R}^2} - 2e^{-m\tilde{t}^2}. \tag{47}$$

**Completing the proof**: Finally, combining (36), (37), (44) and (47), we get with probability exceeding $1 - \frac{\epsilon}{8n\mathrm{R}^2} - 2e^{-\frac{mt^2}{2}} - 2e^{-m\tilde{t}^2}$,
$$\left\|Q(\mathbf{AS})^\dagger \mathbf{A}\right\|_{\max} \leq \sigma_1\left[\max\{\sigma_r\left(\sqrt{\gamma} - 1 - t\right), \sigma_k\left(\sqrt{\gamma} - 1 - t\right) - \sigma_{k+1}\left(\sqrt{\gamma} + 1 + t\right)\}\right. \tag{48}$$
$$\left. - \frac{2C\mathrm{R}\left(\sqrt{\gamma} + 1 + \tilde{t}\right)}{(2^{\mathrm{B}} - 1)\log 2}\sqrt{2\log\left(\frac{16\mathrm{R}^2 n^2 m}{\epsilon}\right)}\right]^{-1}$$
$$\leq \frac{\sigma_1}{\sqrt{\gamma} - 1 - t}\left[\max\left\{\sigma_r, \sigma_k - \sigma_{k+1}\left(\frac{\sqrt{\gamma} + 1 + t}{\sqrt{\gamma} - 1 - t}\right)\right\}\right.$$
$$\left. - \frac{2C\mathrm{R}}{(2^{\mathrm{B}} - 1)\log 2}\left(\frac{\sqrt{\gamma} + 1 + \tilde{t}}{\sqrt{\gamma} - 1 - t}\right)\sqrt{2\log\left(\frac{16\mathrm{R}^2 n^2 m}{\epsilon}\right)}\right]^{-1}$$
$$= \frac{\sigma_1}{\sqrt{\gamma} - 1 - t}\left[\mu - \frac{2C\mathrm{R}}{(2^{\mathrm{B}} - 1)\log 2}\left(\frac{\sqrt{\gamma} + 1 + \tilde{t}}{\sqrt{\gamma} - 1 - t}\right)\sqrt{2\log\left(\frac{16\mathrm{R}^2 n^2 m}{\epsilon}\right)}\right]^{-1}, \tag{49}$$

where we denote $\mu = \max\left\{\sigma_r, \sigma_k - \sigma_{k+1}\left(\frac{\sqrt{\gamma} + 1 + t}{\sqrt{\gamma} - 1 - t}\right)\right\}$. Let us choose our bit-budget B of quantizer $\hat{Q}$ to be such that it satisfies $\|\mathbf{E}\|_2 \leq \frac{\mu}{2}$, i.e.,
$$\mathrm{B} \geq \log_2\left(\frac{4C\mathrm{R}}{\mu \log 2}\left(\frac{\sqrt{\gamma} + 1 + \tilde{t}}{\sqrt{\gamma} - 1 - t}\right)\sqrt{2\log\left(\frac{16\mathrm{R}^2 n^2 m}{\epsilon}\right)} + 1\right). \tag{50}$$

Then,
$$\left\|Q(\mathbf{AS})^\dagger \mathbf{A}\right\|_{\max} \leq \frac{2\sigma_1}{\left(\sqrt{\gamma} - 1 - t\right)\mu} \tag{51}$$

Setting

$$t = \sqrt{\frac{2\log\left(\frac{32n\mathrm{R}^2}{\epsilon}\right)}{m}} \quad \text{and,} \quad \widetilde{t} = \sqrt{\frac{\log\left(\frac{32n\mathrm{R}^2}{\epsilon}\right)}{m}} = \frac{t}{\sqrt{2}}, \tag{52}$$

it follows that with probability exceeding $1 - \frac{\epsilon}{4n\mathrm{R}^2}$,

$$\left\|\mathrm{Q}(\mathbf{AS})^\dagger \mathbf{A}\right\|_{\max} \leq \frac{2}{\sqrt{\gamma}-1-t} \min\left\{ \kappa(\mathbf{A}), \frac{\kappa(\mathbf{A}_k)}{1 - \frac{\sigma_{k+1}}{\sigma_k}\left(\frac{\sqrt{\gamma}+1+t}{\sqrt{\gamma}-1-t}\right)} \right\} \tag{53}$$

This completes the proof. $\qquad\square$

# F    Sketched least squares with quantized response

Since the approximation error guarantees of *sketched least squares with quantized response* might be a problem of independent interest, this is a standalone section, and the notations used in this section are independent of the rest of the paper.

Consider the generalized least squares problem

$$\mathbf{X}^* = \arg\min_{\mathbf{x}\in\mathbb{R}^{p\times q}} \|\mathbf{\Phi X} - \mathbf{Y}\|_{\mathrm{F}}^2, \tag{54}$$

where $\mathbf{\Phi} \in \mathbb{R}^{\ell \times p}$ and $\mathbf{Y} \in \mathbb{R}^{\ell \times q}$. The sketched variant of (54) with quantized response is given by,

$$\widetilde{\mathbf{X}} = \arg\min_{\mathbf{X}\in\mathbb{R}^{p\times q}} \|\mathbf{G\Phi X} - \mathrm{Q}(\mathbf{GY})\|_{\mathrm{F}}^2, \tag{55}$$

where $\mathbf{G} \in \mathbb{R}^{m \times \ell}$ is a Gaussian sketch matrix with entries are distributed as $G_{ij} \sim \mathcal{N}\left(0, \frac{1}{m}\right)$, and $\mathrm{Q} \equiv \mathrm{Q}_{\mathrm{R,B}}$ is the uniformly dithered quantizer as described in (1). We assume that the dynamic range $\mathrm{R} \geq \|\mathbf{GY}\|_{\max}$ so that $\mathrm{Q}$ is unsaturated. The solution of (55) can be obtained in closed form as,

$$\widetilde{\mathbf{X}} = (\mathbf{G\Phi})^\dagger \mathrm{Q}(\mathbf{GY}) \tag{56}$$

The following lemma provides a characterization of the accuracy of $\widetilde{\mathbf{X}}$ with respect to the original problem (54).

**Lemma F.1.** *Let* $\mathbf{G} \in \mathbb{R}^{m\times\ell}$ *be a random Gaussian matrix with entries distributed as* $G_{ij} \sim \mathcal{N}\left(0, \frac{1}{m}\right)$, *and* $\mathrm{Q} \equiv \mathrm{Q}_{\mathrm{R,B}}$ *be a uniformly dithered quantizer with dynamic range* $\mathrm{R}$ *and bit-budget* $\mathrm{B}$. *Furthermore, suppose* $\mathbf{\Phi} \in \mathbb{R}^{\ell\times p}$ *and* $\mathbf{Y} \in \mathbb{R}^{\ell\times q}$ *be given, and let us denote*

$$\mathbf{X}^* = \arg\min_{\mathbf{X}\in\mathbb{R}^{p\times q}} \|\mathbf{\Phi X} - \mathbf{Y}\|_{\mathrm{F}}^2 \quad \text{and} \quad \widetilde{\mathbf{X}} = \arg\min_{\mathbf{X}\in\mathbb{R}^{p\times q}} \|\mathbf{G\Phi X} - \mathrm{Q}(\mathbf{GY})\|_{\mathrm{F}}^2.$$

*Let* $\mathbf{E} = \mathrm{Q}(\mathbf{GY}) - \mathbf{GY}$ *be the quantization error matrix. Then, if* $\mathrm{R} \geq \|\mathbf{GY}\|_{\max}$, *we have*

$$\|\mathbf{\Phi X}^* - \mathbf{Y}\|_{\mathrm{F}}^2 \leq \mathbb{E}\left\|\mathbf{\Phi}\widetilde{\mathbf{X}} - \mathbf{Y}\right\|_{\mathrm{F}}^2 \leq \frac{m-1}{m-r-1}\|\mathbf{\Phi X}^* - \mathbf{Y}\|_{\mathrm{F}}^2 + \frac{q\Delta^2}{4}\frac{\sigma_{\max}^2}{\sigma_{\min}^2}\frac{m^2}{(\ell-m-1)},$$

*where* $r = \mathrm{rank}(\mathbf{\Phi})$, *and* $\sigma_{\max}$ *and* $\sigma_{\min}$ *are the maximum and minimum singular values of* $\mathbf{\Phi}$ *respectively.*

*Proof.* The solution of the generalized least squares problem (54) can be written as:

$$\mathbf{X}^* = [\mathbf{x}_1^* \ldots \mathbf{x}_q^*], \quad \text{where,} \quad \mathbf{x}_i^* = \arg\min_{\mathbf{x}\in\mathbb{R}^p} \|\mathbf{\Phi x} - \mathbf{y}_i\|^2, \tag{57}$$

where $\mathbf{y}_i \in \mathbb{R}^p$ denote the $i^{\mathrm{th}}$ column of $\mathbf{Y}$. Consequently, we will first analyze the standard least squares problem

$$\mathbf{x}^* = \arg\min_{\mathbf{x}\in\mathbb{R}^p} \|\mathbf{\Phi x} - \mathbf{y}\|^2, \tag{58}$$

and generalize the results by concatenating $\mathbf{x}_i^*$ to obtain $\mathbf{X}^*$. The sketched variant of (58) with quantized response is given by,

$$\widetilde{\mathbf{x}} = \arg\min_{\mathbf{x}\in\mathbb{R}^p} \|\mathbf{G\Phi x} - \mathrm{Q}(\mathbf{Gy})\|^2. \tag{59}$$

The solution of (59) is available in closed form as $\widetilde{\mathbf{x}} = (\mathbf{G\Phi})^\dagger \mathrm{Q}(\mathbf{Gy})$. Let us denote the quantization error as $\boldsymbol{\epsilon} \triangleq \mathrm{Q}(\mathbf{Gy}) - \mathbf{Gy} \in \mathbb{R}^m$. We then have,

$$\mathbb{E}\|\mathbf{\Phi}\widetilde{\mathbf{x}} - \mathbf{y}\|_2^2 = \mathbb{E}\|\mathbf{\Phi}(\mathbf{G\Phi})^\dagger \mathrm{Q}(\mathbf{Gy}) - \mathbf{y}\|_2^2 = \mathbb{E}\|\mathbf{\Phi}(\mathbf{G\Phi})^\dagger(\mathbf{Gy} + \boldsymbol{\epsilon}) - \mathbf{y}\|_2^2$$

$$\stackrel{(i)}{=} \mathbb{E}\|\mathbf{\Phi}(\mathbf{G}\mathbf{\Phi})^{\dagger}\mathbf{G}\mathbf{y} - \mathbf{y}\|_2^2 + \mathbb{E}\|\mathbf{\Phi}(\mathbf{G}\mathbf{\Phi})^{\dagger}\boldsymbol{\epsilon}\|_2^2. \quad (60)$$

Here, (i) follows as the cross term disappears because,

$$\mathbb{E}\left[(\mathbf{\Phi}(\mathbf{G}\mathbf{\Phi})^{\dagger}\mathbf{G}\mathbf{y} - \mathbf{y})^{\top}\mathbf{\Phi}(\mathbf{G}\mathbf{\Phi})^{\dagger}\boldsymbol{\epsilon}\right] = \mathbb{E}_{\mathbf{G}}\left[(\mathbf{\Phi}(\mathbf{G}\mathbf{\Phi})^{\dagger}\mathbf{G}\mathbf{y} - \mathbf{y})^{\top}\mathbf{\Phi}(\mathbf{G}\mathbf{\Phi})^{\dagger}\mathbb{E}_{\mathrm{Q}}[\boldsymbol{\epsilon}]\right] = \mathbf{0}, \quad (61)$$

where the last equality follows from (2), since $\mathbb{E}_{\mathrm{Q}}[\boldsymbol{\epsilon}] = \mathbf{0}$ when Q is a uniformly dithered quantizer.

Let us denote the quantization error matrix by $\mathbf{E} = \mathrm{Q}(\mathbf{G}\mathbf{Y}) - \mathbf{G}\mathbf{Y} \in \mathbb{R}^{m \times q}$. Generalizing (60) to the generalized least squares problem by treating each column $\mathbf{y}_i$ separately and adding, we have,

$$\mathbb{E}\left\|\mathbf{\Phi}\widetilde{\mathbf{X}} - \mathbf{Y}\right\|_{\mathrm{F}}^2 = \mathbb{E}\left\|\mathbf{\Phi}(\mathbf{G}\mathbf{\Phi})^{\dagger}\mathbf{G}\mathbf{Y} - \mathbf{Y}\right\|_{\mathrm{F}}^2 + \mathbb{E}\left\|\mathbf{\Phi}(\mathbf{G}\mathbf{\Phi})^{\dagger}\mathbf{E}\right\|_{\mathrm{F}}^2$$

$$\stackrel{(i)}{=} \frac{m-1}{m-r-1}\left\|\mathbf{\Phi}\mathbf{X}^* - \mathbf{Y}\right\|_{\mathrm{F}}^2 + \mathbb{E}\|\mathbf{\Phi}(\mathbf{G}\mathbf{\Phi})^{\dagger}\mathbf{E}\|_{\mathrm{F}}^2. \quad (62)$$

Here, (i) follows from known results on the approximation error of sketched generalized least squares Halko et al. [23], Pilanci [48], and $r$ is the rank of $\mathbf{A}$.

**Upper bounding $\mathbb{E}\|\mathbf{\Phi}(\mathbf{G}\mathbf{\Phi})^{\dagger}\mathbf{E}\|_{\mathrm{F}}^2$:** The remainder of the proof concerns upper bounding the final term in (62). Since $\|\mathbf{X}\|_{\mathrm{F}}^2 = \mathrm{Tr}\left[\mathbf{X}^{\top}\mathbf{X}\right]$ for any matrix $\mathbf{X}$, using the cyclic property of trace,

$$\mathbb{E}\left\|\mathbf{\Phi}(\mathbf{G}\mathbf{\Phi})^{\dagger}\mathbf{E}\right\|_{\mathrm{F}}^2 = \mathbb{E}\left[\mathrm{Tr}\left(\mathbf{E}^{\top}\left((\mathbf{G}\mathbf{\Phi})^{\dagger}\right)^{\top}\mathbf{\Phi}^{\top}\mathbf{\Phi}(\mathbf{G}\mathbf{\Phi})^{\dagger}\mathbf{E}\right)\right]$$

$$= \mathbb{E}\left[\mathrm{Tr}\left(\left((\mathbf{G}\mathbf{\Phi})^{\dagger}\right)^{\top}\mathbf{\Phi}^{\top}\mathbf{\Phi}(\mathbf{G}\mathbf{\Phi})^{\dagger}\mathbf{E}\mathbf{E}^{\top}\right)\right]$$

$$= \mathbb{E}_{\mathbf{G}}\left[\mathrm{Tr}\left(\left((\mathbf{G}\mathbf{\Phi})^{\dagger}\right)^{\top}\mathbf{\Phi}^{\top}\mathbf{\Phi}(\mathbf{G}\mathbf{\Phi})^{\dagger}\mathbb{E}_{\mathrm{Q}}\left[\mathbf{E}\mathbf{E}^{\top}\right]\right)\right] \quad (63)$$

Since $\mathrm{R} \geq \|\mathbf{G}\mathbf{Y}\|_{\max}$, from (2), the $(i,j)^{\mathrm{th}}$-entry of the quantization error matrix $\mathbf{E}$ satisfies,

$$\mathbb{E}\left[E_{ij}\right] = 0, \quad \text{and} \quad \mathrm{Var}(E_{ij}) \leq \frac{\Delta^2}{4} = \frac{\mathrm{R}^2}{\left(2^{\mathrm{B}}-1\right)^2}. \quad (64)$$

So,

$$\mathbb{E}\left[\left(\mathbf{E}\mathbf{E}^{\top}\right)_{ij}\right] = \sum_{k=1}^{q}\mathbb{E}\left[E_{ik}E_{jk}\right] = \begin{cases} q\,\mathrm{Var}(E_{ik}) \leq \frac{q\Delta^2}{4} & \text{for } i = j \\ 0 & \text{for } i \neq j. \end{cases} \quad (65)$$

In other words, $\mathbb{E}_{\mathrm{Q}}\left[\mathbf{E}\mathbf{E}^{\top}\right]$ is a diagonal matrix whose diagonal elements are upper bounded by $\frac{q\Delta^2}{4}$. Let us denote $\mathbf{\Lambda} = \left((\mathbf{G}\mathbf{\Phi})^{\dagger}\right)^{\top}\mathbf{\Phi}^{\top}\mathbf{\Phi}(\mathbf{G}\mathbf{\Phi})^{\dagger}$. Then, (63) simplifies to,

$$\mathbb{E}_{\mathbf{G}}\left[\mathrm{Tr}\left(\mathbf{\Lambda} \cdot \mathbb{E}_{\mathrm{Q}}\left[\mathbf{E}^{\top}\mathbf{E}\right]\right)\right] = \mathbb{E}_{\mathbf{G}}\left[\sum_{i=1}^{m}\Lambda_{ii}\left(\mathbb{E}_{\mathrm{Q}}\left[\mathbf{E}^{\top}\mathbf{E}\right]\right)_{ii}\right] \leq \frac{q\Delta^2}{4}\mathbb{E}_{\mathbf{G}}\left[\mathrm{Tr}(\mathbf{\Lambda})\right]. \quad (66)$$

Furthermore,

$$\mathbb{E}_{\mathbf{G}}\left[\mathrm{Tr}(\mathbf{\Lambda})\right] = \mathbb{E}_{\mathbf{G}}\left[\left\|\mathbf{\Phi}(\mathbf{G}\mathbf{\Phi})^{\dagger}\right\|_{\mathrm{F}}^2\right] \stackrel{(i)}{\leq} \mathbb{E}_{\mathbf{G}}\left[\left\|(\mathbf{G}\mathbf{\Phi})^{\dagger}\right\|_{\mathrm{F}}^2\right]\sigma_{\max}^2(\mathbf{\Phi})$$

$$= \mathbb{E}_{\mathbf{G}}\left[\mathrm{Tr}\left[\left(\mathbf{G}\mathbf{\Phi}\mathbf{\Phi}^{\top}\mathbf{G}^{\top}\right)^{-1}\right]\right]\sigma_{\max}^2(\mathbf{\Phi})$$

$$\stackrel{(ii)}{\leq} \frac{\sigma_{\max}^2(\mathbf{\Phi})}{\sigma_{\min}^2(\mathbf{\Phi})}\mathrm{Tr}\left(\mathbb{E}\left[\left(\mathbf{G}\mathbf{G}^{\top}\right)^{-1}\right]\right)$$

$$\stackrel{(iii)}{=} \frac{\sigma_{\max}^2}{\sigma_{\min}^2}\frac{m^2}{(\ell - m - 1)}. \quad (67)$$

Here, (i) follows from Lemma B.1, (ii) is consequence of Lemma B.2, and (iii) follows from Lemma B.9. Combining (63), (66) and (67) yields,

$$\mathbb{E}\left\|\mathbf{\Phi}(\mathbf{G}\mathbf{\Phi})^{\dagger}\mathbf{E}\right\|_{\mathrm{F}}^2 \leq \frac{q\Delta^2}{4}\frac{\sigma_{\max}^2}{\sigma_{\min}^2}\frac{m^2}{(\ell - m - 1)}. \quad (68)$$

This completes the proof. $\qquad\square$

# G  LPLR algorithm: Approximation error analysis

To prove Thm. 3.2, we will first prove the result when both quantizers $Q$ and $Q'$ are unsaturated. We first prove Lemma G.1, which gives us an upper bound on the approximation error when $Q$ and $Q'$ are unsaturated. Since we can ensure that they remain unsaturated with high probability (with appropriate choices of dynamic ranges), the final approximation error upper bound in Thm. 3.2 is slightly worse than Lemma G.1.

**Lemma G.1.** *Let our matrix* $\mathbf{A} \in \mathbb{R}^{n \times d}$ *have non-zero singular values* $\sigma_1, \ldots, \sigma_k, \sigma_{k+1}, \ldots, \sigma_r$, *where* $r = \operatorname{rank}(\mathbf{A})$, *and bounded row norms* $\left\|\mathbf{a}^{(i)}\right\| \leq \mathrm{R}$. *Let* $\kappa(\mathbf{A}) = \sigma_1/\sigma_r$ *and* $\kappa(\mathbf{A}_k) = \sigma_1/\sigma_k$ *respectively be the condition numbers of* $\mathbf{A}$ *and the best rank-$k$ approximation of* $\mathbf{A}$, *and let us denote*

$$t = \sqrt{\frac{2\log\left(\frac{32n\mathrm{R}^2}{\epsilon}\right)}{m}}, \quad \kappa = \min\left\{\kappa(\mathbf{A}), \frac{\kappa(\mathbf{A}_k)}{1 - \frac{\sigma_{k+1}}{\sigma_k}\left(\frac{\sqrt{\gamma}+1+t}{\sqrt{\gamma}-1-t}\right)}\right\},$$

*for some sufficiently small $\epsilon$ that satisfies* $0 < \epsilon \leq \frac{4n\mathrm{R}^2\kappa(\mathbf{A})^2}{\gamma}$. *Here* $\gamma = d/m$ *is the aspect ratio of the sketching matrix* $\mathbf{S} \in \mathbb{R}^{d \times m}$ *with* $S_{ij} \overset{i.i.d.}{\sim} \mathcal{N}\left(0, \frac{1}{m}\right)$. *Suppose the dynamic ranges of the quantizers* $Q$ *and* $Q'$ *are set to* $\mathrm{R}\sqrt{\frac{2\log\left(\frac{16\mathrm{R}^2n^2m}{\epsilon}\right)}{m}}$ *and* $\frac{2\kappa}{\sqrt{\gamma}-1-t}$ *as dictated by Lemmas E.1 and E.2 respectively, and suppose their bit-budgets* $\mathrm{B}$ *and* $\mathrm{B}'$ *satisfy*

$$\mathrm{B} \geq \max\{\mathrm{B}_1, \mathrm{B}_2\} \quad and, \quad \mathrm{B}' \geq \log_2\left(\frac{4\mathrm{R}\kappa}{(\sqrt{\gamma}-1-t)}\sqrt{\frac{nd}{\epsilon}}+1\right),$$

*where,*

$$\mathrm{B}_1 = \log_2\left(\frac{2\mathrm{R}\kappa(\mathbf{A}_k)\sqrt{2n}}{\sqrt{\epsilon\left(\gamma-1-\frac{1}{m}\right)}}\sqrt{\log\left(\frac{16\mathrm{R}^2n^2m}{\epsilon}\right)}+1\right),$$

*and* $\mathrm{B}_2$ *is equal to*

$$\log_2\left(\frac{4C\mathrm{R}}{\max\left\{\sigma_r, \sigma_k - \sigma_{k+1}\left(\frac{\sqrt{\gamma}+1+t}{\sqrt{\gamma}-1-t}\right)\right\}\log 2}\left(\frac{\sqrt{\gamma}+1+t/\sqrt{2}}{\sqrt{\gamma}-1-t}\right)\sqrt{2\log\left(\frac{16\mathrm{R}^2n^2m}{\epsilon}\right)}+1\right).$$

*Furthermore, let* $\mathcal{E}$ *be the event that both quantizers* $Q$ *and* $Q'$ *are unsaturated. Then,*

$$\mathbb{E}\left[\left\|Q(\mathbf{AS})Q'\left(Q(\mathbf{AS})^\dagger\mathbf{A}\right) - \mathbf{A}\right\|_{\mathrm{F}}^2 \mathbb{1}_{\mathcal{E}}\right] \leq \left(1 + \frac{k}{m-k-1}\right)\|\mathbf{A}_k - \mathbf{A}\|_{\mathrm{F}}^2 + \frac{3\epsilon}{4},$$

*where* $\mathbb{1}_{(\cdot)}$ *is the indicator function.*

*Proof.* Let us denote the quantization error matrices from $Q$ and $Q'$ as $\mathbf{E} \in \mathbb{R}^{n \times m}$ and $\mathbf{E}' \in \mathbb{R}^{m \times d}$ respectively, i.e., $\mathbf{E} = Q(\mathbf{AS}) - \mathbf{AS}$ and $\mathbf{E}' = Q'\left(Q(\mathbf{AS})^\dagger\mathbf{A}\right) - Q(\mathbf{AS})^\dagger\mathbf{A}$. Since $\mathbb{1}_{\mathcal{E}} = 1$ implies that $Q$ and $Q'$ are unsaturated, $\mathbb{E}[\mathbf{E}] = \mathbf{0}$ and $\mathbb{E}[\mathbf{E}'] = \mathbf{0}$. Then,

$$\mathbb{E}\left[\left\|Q(\mathbf{AS})Q'\left(Q(\mathbf{AS})^\dagger\mathbf{A}\right) - \mathbf{A}\right\|_{\mathrm{F}}^2 \mathbb{1}_{\mathcal{E}}\right] = \mathbb{E}_{\mathbf{S}}\left[\mathbb{E}_{Q,Q'}\left[\left\|Q(\mathbf{AS})Q'\left(Q(\mathbf{AS})^\dagger\mathbf{A}\right) - \mathbf{A}\right\|_{\mathrm{F}}^2 \mathbb{1}_{\mathcal{E}}\right]\right].$$

Then,

$$\mathbb{E}_{Q,Q'}\left[\left\|Q(\mathbf{AS})Q'\left(Q(\mathbf{AS})^\dagger\mathbf{A}\right) - \mathbf{A}\right\|_{\mathrm{F}}^2\right]\mathbb{1}_{\mathcal{E}}$$

$$= \mathbb{E}_{Q,Q'}\left[\left\|Q(\mathbf{AS})\left(Q(\mathbf{AS})^\dagger\mathbf{A} + \mathbf{E}'\right) - \mathbf{A}\right\|_{\mathrm{F}}^2\right]\mathbb{1}_{\mathcal{E}}$$

$$\overset{(i)}{=} \underbrace{\mathbb{E}_Q\left[\left\|Q(\mathbf{AS})Q(\mathbf{AS})^\dagger\mathbf{A} - \mathbf{A}\right\|_{\mathrm{F}}^2\right]\mathbb{1}_{\mathcal{E}}}_{\mathrm{T}_1} + \underbrace{\mathbb{E}_{Q,Q'}\left[\left\|Q(\mathbf{AS})\mathbf{E}'\right\|_{\mathrm{F}}^2\right]\mathbb{1}_{\mathcal{E}}}_{\mathrm{T}_2} \tag{69}$$

The cross terms disappear in (i) because,

$$\mathbb{E}_{Q,Q'}\left[\operatorname{Tr}\left[\left(Q(\mathbf{AS})Q(\mathbf{AS})^\dagger\mathbf{A} - \mathbf{A}\right)^\top Q(\mathbf{AS})\mathbf{E}'\right]\right]\mathbb{1}_{\mathcal{E}}$$

$$= \operatorname{Tr}\left(\mathbb{E}_Q\left[\left(Q(\mathbf{AS})Q(\mathbf{AS})^\dagger\mathbf{A} - \mathbf{A}\right)^\top Q(\mathbf{AS})\,\mathbb{E}_{Q'}[\mathbf{E}']\right]\right)\mathbb{1}_{\mathcal{E}} \overset{(ii)}{=} 0, \tag{70}$$

where, (ii) follows from $\mathbb{E}_{Q'}[\mathbf{E}'] = \mathbf{0}$ as the quantizer $Q'$ is unbiased when unsaturated.

**Analyzing the term** $T_1$: Recall that $\mathbf{A}_k$ is the best rank-$k$ approximation of $\mathbf{A}$ obtained using computing the full-SVD of $\mathbf{A}$ and subsequently making all singular values of $\mathbf{A}$ less than $\sigma_k$ as 0. We next analyze the first term in (69) as

$$
\|Q(\mathbf{AS})Q(\mathbf{AS})^\dagger \mathbf{A} - \mathbf{A}\|_\mathrm{F}^2 \leq \|Q(\mathbf{AS})(\mathbf{A}_k\mathbf{S})^\dagger \mathbf{A}_k - \mathbf{A}\|_\mathrm{F}^2
$$
$$
= \|\mathbf{A}_k^\top (\mathbf{S}^\top \mathbf{A}_k^\top)^\dagger Q(\mathbf{S}^\top \mathbf{A}^\top) - \mathbf{A}^\top\|_\mathrm{F}^2
$$
$$
= \|\mathbf{A}_k^\top \widetilde{\mathbf{X}} - \mathbf{A}^\top\|_\mathrm{F}^2, \tag{71}
$$

where $\widetilde{\mathbf{X}} \triangleq \arg\min_{\mathbf{X}} \|\mathbf{S}^\top \mathbf{A}_k^\top \mathbf{X} - Q(\mathbf{S}^\top \mathbf{A}^\top)\|_\mathrm{F}^2$. This minimization problem is a Gaussian sketched variant of a generalized least squares problem with the response matrix quantized as seen in App. F. Corresponding to the notations in App. F, here we have $\mathbf{S}^\top$ instead of $\mathbf{G}$, $\mathbf{A}_k^\top$ instead of $\mathbf{\Phi}$, and $\mathbf{A}^\top$ instead of $\mathbf{Y}$. As a consequence of Lemma F.1, we have the upper bound,

$$
\mathbb{E}_Q\left[\|Q(\mathbf{AS})Q(\mathbf{AS})^\dagger \mathbf{A} - \mathbf{A}\|_\mathrm{F}^2\right]\mathbb{1}_\mathcal{E} \leq \frac{m-1}{m-k-1}\|\mathbf{A}_k - \mathbf{A}\|_\mathrm{F}^2 + \frac{n\Delta^2}{4}\frac{\sigma_1^2}{\sigma_k^2}\frac{m^2}{(d-m-1)} \tag{72}
$$

**Analyzing the term** $T_2$: The term $T_2$ can be written as,

$$
\mathbb{E}_{Q,Q'}\left[\|Q(\mathbf{AS})\mathbf{E}'\|_\mathrm{F}^2\right]\mathbb{1}_\mathcal{E}
$$
$$
= \mathbb{E}_{Q,Q'}\left[\|(\mathbf{AS}+\mathbf{E})\mathbf{E}'\|_\mathrm{F}^2\right]\mathbb{1}_\mathcal{E}
$$
$$
= \mathbb{E}_{Q'}\left[\|\mathbf{ASE}'\|_\mathrm{F}^2\right]\mathbb{1}_\mathcal{E} + \mathbb{E}_{Q,Q'}\left[\|\mathbf{EE}'\|_\mathrm{F}^2\right]\mathbb{1}_\mathcal{E} + \mathrm{Tr}\left(\mathbb{E}_{Q'}\left[(\mathbf{ASE}')^\top \mathbb{E}_Q[\mathbf{E}]\,\mathbf{E}'\right]\right)\mathbb{1}_\mathcal{E}
$$
$$
\overset{(i)}{=} \mathbb{E}\left[\|\mathbf{ASE}'\|_\mathrm{F}^2\right]\mathbb{1}_\mathcal{E} + \mathbb{E}\left[\|\mathbf{EE}'\|_\mathrm{F}^2\right]\mathbb{1}_\mathcal{E}, \tag{73}
$$

where, (i) follows once again since $\mathbb{E}_{Q'}[\mathbf{E}']\mathbb{1}_\mathcal{E} = \mathbf{0}$. The first term in (73) can be rewritten as

$$
\mathbb{E}\left[\|\mathbf{ASE}'\|_\mathrm{F}^2\mathbb{1}_\mathcal{E}\right] = \mathbb{E}\left[\mathrm{Tr}\left(\mathbf{E}'^\top \mathbf{S}^\top \mathbf{A}^\top \mathbf{ASE}'\right)\mathbb{1}_\mathcal{E}\right]
$$
$$
= \mathbb{E}\left[\mathrm{Tr}\left(\mathbf{S}^\top \mathbf{A}^\top \mathbf{ASE}'\mathbf{E}'^\top\right)\mathbb{1}_\mathcal{E}\right]
$$
$$
= \mathrm{Tr}\left(\mathbb{E}_\mathbf{S}\left[\mathbf{S}^\top \mathbf{A}^\top \mathbf{AS}\,\mathbb{E}_{Q'}\left[\mathbf{E}'\mathbf{E}'^\top\right]\mathbb{1}_\mathcal{E}\right]\right)
$$
$$
= \mathbb{E}_\mathbf{S}\left[\mathrm{Tr}\left(\mathbf{S}^\top \mathbf{A}^\top \mathbf{AS}\,\mathbb{E}_{Q'}\left[\mathbf{E}'\mathbf{E}'^\top\right]\mathbb{1}_\mathcal{E}\right)\right] \tag{74}
$$

Similar to (65) in the proof of Lemma F.1, $\mathbb{E}_{Q'}\left[\mathbf{E}'\mathbf{E}'^\top\right]\mathbb{1}_\mathcal{E}$ is a diagonal matrix with

$$
\mathbb{E}_{Q'}\left[\left(\mathbf{E}'\mathbf{E}'^\top\right)_{ij}\right]\mathbb{1}_\mathcal{E} = \sum_{k=1}^d \mathbb{E}_{Q'}\left[E'_{ik}E'_{jk}\right]\mathbb{1}_\mathcal{E} = \begin{cases} d\cdot\mathrm{Var}(E_{ik}) \leq \frac{d\Delta'^2}{4} & \text{for } i=j \\ 0 & \text{for } i\neq j. \end{cases} \tag{75}
$$

Let us denote $\mathbf{\Gamma} \triangleq \mathbf{S}^\top \mathbf{A}^\top \mathbf{AS}$. Then using the fact that $\mathbb{1}_\mathcal{E} \leq 1$, we get,

$$
\mathbb{E}_\mathbf{S}\left[\mathrm{Tr}\left(\mathbf{\Gamma}\,\mathbb{E}_{Q'}\left[\mathbf{E}'\mathbf{E}'^\top\right]\mathbb{1}_\mathcal{E}\right)\right] = \mathbb{E}_\mathbf{S}\left[\sum_{i=1}^m \Gamma_{ii}\left(\mathbb{E}_{Q'}\left[\mathbf{E}'\mathbf{E}'^\top\right]\right)_{ii}\right]\mathbb{1}_\mathcal{E}
$$
$$
\leq \frac{d\Delta'^2}{4}\mathrm{Tr}\left(\mathbb{E}_\mathbf{S}\mathbf{\Gamma}\right)
$$
$$
= \frac{d\Delta'^2}{4}\mathrm{Tr}\left(\mathbf{A}^\top \mathbf{A}\,\mathbb{E}_\mathbf{S}\left[\mathbf{SS}^\top\right]\right) \overset{(i)}{=} \frac{d\Delta'^2}{4}\|\mathbf{A}\|_\mathrm{F}^2. \tag{76}
$$

Here, the last equality follows as $\mathbb{E}_\mathbf{S}\left[\mathbf{SS}^\top\right] = \mathbf{I}_d$. This is because the $(i,j)^{\mathrm{th}}$ entry of $\mathbb{E}_\mathbf{S}\left[\mathbf{SS}^\top\right]$ is

$$
\mathbb{E}\left[\left(\mathbf{SS}^\top\right)_{ij}\right] = \sum_{k=1}^m \mathbb{E}\left[S_{ik}S_{jk}\right] = \begin{cases} m\,\mathbb{E}\left[S_{ik}^2\right] = m\,\mathrm{Var}\left(S_{ik}\right) = 1 & \text{for} \quad i=j \\ \sum_{i=1}^m \mathbb{E}[S_{ik}]\mathbb{E}[S_{jk}] = 0 & \text{for} \quad i\neq j. \end{cases}
$$

Furthermore, the second term $\mathbb{E}_{Q,Q'}\left\|\mathbf{EE}'^\top\right\|_\mathrm{F}^2\mathbb{1}_\mathcal{E}$ can be simplified as follows,

$$
\mathbb{E}_{Q,Q'}\|\mathbf{EE}'\|_\mathrm{F}^2\mathbb{1}_\mathcal{E} = \mathbb{E}_{Q,Q'}\left[\mathrm{Tr}\left(\mathbf{E}'^\top \mathbf{E}^\top \mathbf{EE}'\right)\right]\mathbb{1}_\mathcal{E} = \mathbb{E}_Q\left[\mathrm{Tr}\left(\mathbf{E}^\top \mathbf{E}\,\mathbb{E}_{Q'}\left(\mathbf{E}'\mathbf{E}'^\top\right)\right)\right]\mathbb{1}_\mathcal{E}
$$
$$
\overset{(i)}{=} \mathbb{E}_Q\left[\sum_{i=1}^m \left(\mathbf{E}^\top \mathbf{E}\right)_{ii}\left(\mathbb{E}_{Q'}\left[\mathbf{E}'\mathbf{E}'^\top\right]\right)_{ii}\right]\mathbb{1}_\mathcal{E}
$$

$$\overset{(ii)}{\leq} \frac{d\Delta'^2}{4} \sum_{i=1}^{m} \mathbb{E}_Q \left[ \left( \mathbf{E}^\top \mathbf{E} \right)_{ii} \right] \mathbb{1}_{\mathcal{E}}$$

$$\leq \frac{nmd\Delta^2\Delta'^2}{16}. \tag{77}$$

Here, (i) follows since $\mathbb{E}_{Q'} \left[ \mathbf{E}'\mathbf{E}'^\top \right] \mathbb{1}_{\mathcal{E}}$ is a diagonal matrix as seen before, and (ii) follows from the upper bound derived on $\mathbb{E} \left[ \left( \mathbf{E}^\top \mathbf{E} \right)_{ii} \right] \mathbb{1}_{\mathcal{E}}$ in (65).

Combining all of the above, the approximation error of our low-precision low-rank approximation scheme can be upper bounded as,

$$\mathbb{E} \| Q(\mathbf{A}\mathbf{S})Q' \left( Q(\mathbf{A}\mathbf{S})^\dagger \mathbf{A} \right) - \mathbf{A} \|_F^2$$

$$\leq \underbrace{\frac{(m-1)}{(m-k-1)} \| \mathbf{A}_k - \mathbf{A} \|_F^2}_{T_3} + \underbrace{\frac{n\Delta^2}{4} \frac{\sigma_1^2}{\sigma_k^2} \frac{m^2}{(d-m-1)}}_{T_4} + \underbrace{\frac{d\Delta'^2}{4} \| \mathbf{A} \|_F^2}_{T_5} + \underbrace{\frac{nmd\Delta^2\Delta'^2}{16}}_{T_6}. \tag{78}$$

Here, the first term $T_3$ is the low-rank approximation error, whereas the remaining terms, i.e., $T_4$, $T_5$ and $T_6$ appear due to quantization. We now analyze each of them separately.

**Analyzing the term** $T_4$: Since we choose the dynamic range of quantizer Q to be $R\sqrt{\frac{2\log\left(\frac{16R^2n^2m}{\epsilon}\right)}{m}}$ (ref. Lemma E.1), we have

$$\frac{n\Delta^2}{4} \frac{\sigma_1^2}{\sigma_k^2} \frac{m^2}{(d-m-1)} \leq \frac{2nmR^2\kappa(\mathbf{A}_k)^2}{\left(2^B - 1\right)^2 (d-m-1)} \log \left( \frac{16R^2n^2m}{\epsilon} \right) \overset{(i)}{\leq} \frac{\epsilon}{4}. \tag{79}$$

Here, we can ensure (i) holds true, i.e., that this term does not exceed $\epsilon/4$ if we set the bit-budget B to satisfy,

$$B \geq \log_2 \left( \frac{2R\kappa(\mathbf{A}_k)\sqrt{2n}}{\sqrt{\epsilon \left( \gamma - 1 - \frac{1}{m} \right)}} \sqrt{\log \left( \frac{16R^2n^2m}{\epsilon} \right)} + 1 \right). \tag{80}$$

**Analyzing the term** $T_5$: Since we choose the dynamic range of the quantizer $Q'$ to be equal to $\frac{2\kappa}{\sqrt{\gamma}-1-t}$ (ref. to Lemma E.2), where $t$ and $\kappa$ are defined as in (33), we have

$$\frac{d\Delta'^2}{4} \| \mathbf{A} \|_F^2 \leq \frac{ndR^2}{\left(2^{B'} - 1\right)^2} \frac{4\kappa^2}{\left(\sqrt{\gamma} - 1 - t\right)^2} \overset{(i)}{\leq} \frac{\epsilon}{4}, \tag{81}$$

where we have made use of Asm. 3.1. Once again, in order to ensure (i), it suffices to choose the bit-budget $B'$ to be

$$B' \geq \log_2 \left( \frac{4R\kappa}{\left(\sqrt{\gamma} - 1 - t\right)} \sqrt{\frac{nd}{\epsilon}} + 1 \right) \tag{82}$$

Note that in order for the dynamic range of quantizer $Q'$ to hold true in Lemma E.2, we also require B to satisfy (34), i.e.,

$$B \geq \log_2 \left( \frac{4CR}{\max\left\{ \sigma_r, \sigma_k - \sigma_{k+1} \left( \frac{\sqrt{\gamma}+1+t}{\sqrt{\gamma}-1-t} \right) \right\} \log 2} \left( \frac{\sqrt{\gamma} + 1 + t/\sqrt{2}}{\sqrt{\gamma} - 1 - t} \right) \sqrt{2 \log \left( \frac{16R^2n^2m}{\epsilon} \right)} + 1 \right). \tag{83}$$

**Analyzing the term** $T_6$: Using bit-budgets B and $B'$ as in (80) and (82), the final term is

$$nmd\frac{\Delta^2}{4}\frac{\Delta'^2}{4} \leq nmd\frac{\epsilon}{4} \left( \frac{\gamma - 1 - \frac{1}{m}}{nm\kappa(\mathbf{A}_k)} \right) \frac{\epsilon}{4ndR^2} \leq \frac{\epsilon^2\gamma}{16nR^2\kappa(\mathbf{A}_k)^2} \overset{(i)}{\leq} \frac{\epsilon}{4} \tag{84}$$

Here, (i) is ensured by choosing $\epsilon$ to be sufficiently small – specifically, $\epsilon \leq \frac{4nR^2\kappa(\mathbf{A})^2}{\gamma}$.

**Tying it all together**: Combining (79), (81), and (84), with the bit-budgets set appropriately as dictated by (80), (83), and (82), eq. (78) simplifies to

$$\mathbb{E}\|Q(\mathbf{AS})Q'\left(Q(\mathbf{AS})^\dagger \mathbf{A}\right) - \mathbf{A}\|_F^2 \le \left(1 + \frac{k}{m-k-1}\right)\|\mathbf{A}_k - \mathbf{A}\|_F^2 + \frac{3\epsilon}{4}. \tag{85}$$

This completes the proof.

$\square$

## G.1 LPLR approximation error: Proof of Thm. 3.2

We now formally state our main approximation result.

**Theorem G.2. (LPLR approximation error (formal))** *Let our matrix* $\mathbf{A} \in \mathbb{R}^{n \times d}$ *have non-zero singular values* $\sigma_1, \ldots, \sigma_k, \sigma_{k+1}, \ldots, \sigma_r$, *where* $r = \text{rank}(\mathbf{A})$, *and bounded row norms* $\|\mathbf{a}^{(i)}\| \le \text{R}$. *Let* $\kappa(\mathbf{A}) = \sigma_1/\sigma_r$ *and* $\kappa(\mathbf{A}_k) = \sigma_1/\sigma_k$ *respectively be the condition numbers of* $\mathbf{A}$ *and the best rank-$k$ approximation of* $\mathbf{A}$, *and let us denote*

$$t = \sqrt{\frac{2\log\left(\frac{32n\text{R}^2}{\epsilon}\right)}{m}}, \quad \kappa = \min\left\{\kappa(\mathbf{A}), \frac{\kappa(\mathbf{A}_k)}{1 - \frac{\sigma_{k+1}}{\sigma_k}\left(\frac{\sqrt{\gamma}+1+t}{\sqrt{\gamma}-1-t}\right)}\right\},$$

*for some sufficiently small* $\epsilon$ *that satisfies* $0 < \epsilon \le \frac{4n\text{R}^2\kappa(\mathbf{A})^2}{\gamma}$. *Here* $\gamma = d/m$ *is the aspect ratio of the sketching matrix* $\mathbf{S} \in \mathbb{R}^{d \times m}$ *with* $S_{ij} \overset{i.i.d.}{\sim} \mathcal{N}\left(0, \frac{1}{m}\right)$. *Suppose the dynamic ranges of the quantizers* $Q$ *and* $Q'$ *are set to* $\text{R}\sqrt{\frac{2\log\left(\frac{16\text{R}^2 n^2 m}{\epsilon}\right)}{m}}$ *and* $\frac{2\kappa}{\sqrt{\gamma}-1-t}$ *respectively, and suppose their bit-budgets* $\text{B}$ *and* $\text{B}'$ *satisfy*

$$\text{B} \ge \max\{\text{B}_1, \text{B}_2\} \quad and, \quad \text{B}' \ge \log_2\left(\frac{4\text{R}\kappa}{(\sqrt{\gamma}-1-t)}\sqrt{\frac{nd}{\epsilon}}+1\right),$$

*where,*

$$\text{B}_1 = \log_2\left(\frac{2\text{R}\kappa(\mathbf{A}_k)\sqrt{2n}}{\sqrt{\epsilon\left(\gamma - 1 - \frac{1}{m}\right)}}\sqrt{\log\left(\frac{16\text{R}^2 n^2 m}{\epsilon}\right)}+1\right),$$

*and* $\text{B}_2$ *is equal to*

$$\log_2\left(\frac{4C\text{R}}{\max\left\{\sigma_r, \sigma_k - \sigma_{k+1}\left(\frac{\sqrt{\gamma}+1+t}{\sqrt{\gamma}-1-t}\right)\right\}\log 2}\left(\frac{\sqrt{\gamma}+1+t/\sqrt{2}}{\sqrt{\gamma}-1-t}\right)\sqrt{2\log\left(\frac{16\text{R}^2 n^2 m}{\epsilon}\right)}+1\right).$$

*Then, the low-precision and low-rank factorization returned by Alg. 1 satisfies*

$$\mathbb{E}\|\mathbf{LR} - \mathbf{A}\|_F^2 \le \left(1 + \frac{k}{m-k-1}\right)\|\mathbf{A}_k - \mathbf{A}\|_F^2 + \epsilon, \tag{86}$$

*where the expectation is over the random sketching matrix* $\mathbf{S}$, *as well as the inherent stochasticity from quantizers* $Q$ *and* $Q'$.

*Proof.* For the purpose of analysis, we assume that Alg. 1 returns $\mathbf{L} = \mathbf{0}$ and $\mathbf{R} = \mathbf{0}$ if either quantizer $Q$ or $Q'$ gets saturated. In practical implementation, it can easily be checked if either quantizer $Q$ or $Q'$ gets saturated or not, and the algorithm can be repeated again with a new realization of the sketching matrix $\mathbf{S}$ and stochastic quantizer $Q$. Since the choice of dynamic ranges for $Q$ and $Q'$ ensures that they remain unsaturated with a high probability, "reasonably few" realizations of $\mathbf{S}$ would suffice to get at least one good realization in which $Q$ and $Q'$ are unsaturated.

However, in what follows, we assume that that if either quantizer $Q$ or $Q'$ gets saturated, then the algorithm returns $\mathbf{0}$ as an estimate of $\mathbf{A}$, resulting in a Frobenius norm error of $\|\mathbf{A}\|_F$. Since this happen with a very small probability, we show that its contribution to the expected Frobenius norm error of Alg. 1 is small as well. With this in mind, the expected approximation error can be written as

$$\mathbb{E}\|\mathbf{LR} - \mathbf{A}\|_F^2 = \mathbb{E}\left[\left\|Q(\mathbf{AS})Q'\left(Q(\mathbf{AS})^\dagger \mathbf{A}\right) - \mathbf{A}\right\|_F^2 \mathbb{1}_\mathcal{E}\right] + \mathbb{E}\left[\|\mathbf{A}\|_F^2 \mathbb{1}_{\mathcal{E}^C}\right]$$

$$\overset{(i)}{\leq} \mathbb{E}\left[\left\|Q\left(\mathbf{AS}\right)Q'\left(Q\left(\mathbf{AS}\right)^\dagger \mathbf{A}\right) - \mathbf{A}\right\|_F^2 \mathbb{1}_\mathcal{E}\right] + n\mathrm{R}^2 \Pr\left(\mathbb{1}_{\mathcal{E}^C}\right). \quad (87)$$

Inequality (i) follows from Asm. 3.1 and the fact that the expectation of indicator function of an event is the probability of the event. From Lemma G.1, the first term can be upper bounded as:

$$\mathbb{E}\left[\left\|Q(\mathbf{AS})Q'\left(Q(\mathbf{AS})^\dagger\mathbf{A}\right) - \mathbf{A}\right\|_F^2 \mathbb{1}_\mathcal{E}\right] \leq \left(1 + \frac{k}{m-k-1}\right)\|\mathbf{A}_k - \mathbf{A}\|_F^2 + \frac{3\epsilon}{4}. \quad (88)$$

Since Lemmas E.1 and E.2 state the probabilities of quantizers Q and Q' being unsaturated, $\Pr\left(\mathbb{1}_{\mathcal{E}^C}\right)$ can be obtained by an application of union bound as

$$\Pr\left(\mathbb{1}_{\mathcal{E}^C}\right) \leq \frac{\epsilon}{8n\mathrm{R}^2} + \frac{\epsilon}{8n\mathrm{R}^2} = \frac{\epsilon}{4n\mathrm{R}^2}. \quad (89)$$

Then, (87) can be written as:

$$\mathbb{E}\|\mathbf{LR} - \mathbf{A}\|_F^2 \leq \left(1 + \frac{k}{m-k-1}\right)\|\mathbf{A}_k - \mathbf{A}\|_F^2 + \epsilon. \quad (90)$$

This completes the proof. $\qquad \square$

## G.2 Informal version of Thm. G.2

From Thm. G.2, we can get a (simplified) asymptotic dependence of the bit-budgets B and B'. We have $\mathrm{B} \geq \max\{\mathrm{B}_1, \mathrm{B}_2\}$,

$$\mathrm{B}_1 = \log_2\left(\frac{2\mathrm{R}\kappa(\mathbf{A}_k)\sqrt{2n}}{\sqrt{\epsilon\left(\gamma - 1 - \frac{1}{m}\right)}}\sqrt{\log\left(\frac{16\mathrm{R}^2 n^2 m}{\epsilon}\right)} + 1\right)$$

$$\overset{(i)}{=} \frac{1}{2}\log_2\left(\frac{\kappa(\mathbf{A}_k)^2 nm}{\epsilon\left(d - m - 1\right)}\log\left(\frac{mn^2}{\epsilon}\right)\right)$$

$$\overset{(ii)}{=} \frac{1}{2}\log_2\left(\frac{\kappa(\mathbf{A}_k)^2}{\epsilon}\frac{nm}{d}\log\left(\frac{mn^2}{\epsilon}\right)\right), \quad (91)$$

where we have ignored the constant terms inside the $\log_2(\cdot)$ in (i) and considered the regime $m \ll d$ in (ii). Furthermore, $\mathrm{B}_2$ is

$$\mathrm{B}_2 = \log_2\left(\frac{4C\mathrm{R}}{\max\left\{\sigma_r, \sigma_k - \sigma_{k+1}\left(\frac{\sqrt{\gamma}+1+t}{\sqrt{\gamma}-1-t}\right)\right\}\log 2}\left(\frac{\sqrt{\gamma}+1+t/\sqrt{2}}{\sqrt{\gamma}-1-t}\right)\sqrt{2\log\left(\frac{16\mathrm{R}^2 n^2 m}{\epsilon}\right)} + 1\right)$$

$$\overset{(i)}{=} \frac{1}{2}\log_2\left(\frac{1}{(\max\{\sigma_r, \sigma_k - \sigma_{k+1}c'\})^2}\log\left(\frac{mn^2}{\epsilon}\right)\right). \quad (92)$$

where $c'$ is approximately a constant. Here, (i) follows because when $m \ll d$, we have

$$\frac{\sqrt{\gamma}+1+t}{\sqrt{\gamma}-1-t} = \frac{\sqrt{d} + \sqrt{m} + \mathrm{O}\left(\sqrt{\log\left(\frac{n}{\epsilon}\right)}\right)}{\sqrt{d} - \sqrt{m} - \mathrm{O}\left(\sqrt{\log\left(\frac{n}{\epsilon}\right)}\right)} = \mathrm{O}(1). \quad (93)$$

Similarly, $\frac{\sqrt{\gamma}+1+t/\sqrt{2}}{\sqrt{\gamma}-1-t} = \mathrm{O}(1)$. Comparing (91) and (92), we have

$$\mathrm{B} \geq \frac{1}{2}\log_2\left(\frac{\kappa(\mathbf{A}_k)^2}{\epsilon}\frac{nm}{d}\log\left(\frac{mn^2}{\epsilon}\right)\right). \quad (94)$$

Moreover, the bit-budget B' is

$$\mathrm{B}' \geq \log_2\left(\frac{4\mathrm{R}\kappa}{(\sqrt{\gamma}-1-t)}\sqrt{\frac{nd}{\epsilon}} + 1\right)$$

$$\overset{(i)}{=} \log_2\left(\frac{\kappa\sqrt{ndm}}{\sqrt{\epsilon}\left(\sqrt{d}-\sqrt{m}-\sqrt{\frac{n}{\epsilon}}\right)}\right) \overset{(ii)}{=} \frac{1}{2}\log_2\left(\kappa^2\frac{nm}{\epsilon}\right), \quad (95)$$

where once again, to get (i) we have ignored the constants, and (ii) holds true when $m \ll d$. Moreover, suppose $n \approx d$ (if they are not equal, then the total bit-budget should be computed as a

weighted average, since the $\mathbf{L}$ has $nm$ elements and the second low-rank factor has $md$ elements). Then, for some constant $c_3$ that depends on R, the total bit-budget is

$$\mathrm{B} \geq \frac{1}{4} \log_2 \left( \frac{c_3^2 \kappa(\mathbf{A}_k)^2 \kappa^2}{\epsilon^2} \frac{n^2 m^2}{d} \log \left( \frac{mn^2}{\epsilon} \right) \right) = \frac{1}{2} \log_2 \left( \frac{c_3 \kappa(\mathbf{A}_k) \kappa}{\epsilon} \frac{nm}{\sqrt{d}} \sqrt{\log \left( \frac{mn^2}{\epsilon} \right)} \right).$$

Furthermore, the dynamic range of quantizer Q is $c_1 \sqrt{\frac{\log\left(\frac{n}{\epsilon}\right)}{m}}$ and that of quantizer Q$'$ when $m \ll d$ is

$$\frac{2\kappa}{\sqrt{\gamma} - 1 - t} = 2\sqrt{\frac{m}{d}} \min \left\{ \kappa(\mathbf{A}), \frac{\kappa(\mathbf{A}_k)}{1 - c_4 \frac{\sigma_{k+1}}{\sigma_k}} \right\},$$

for constants $c_1$, $c_2$ and $c_4$ that depend on R, where additive logarithmic terms are ignored with respect to $m$. This gives us the result in Thm. 3.2.

# H    Direct-SVD Quantization: Approximation error analysis

In this section, we consider the baseline scheme for obtaining low-precision low-rank factorization by first computing the optimal low-rank factorization using SVD and individually quantizing the low-rank factors with uniform scalar quantizers. For any given matrix, $\mathbf{A} \in \mathbb{R}^{n \times d}$, we compute the full SVD as $\mathbf{A} = \mathbf{U}\boldsymbol{\Sigma}\mathbf{V}^\top$. The best (unquantized) rank-$k$ approximation can be obtained from this by considering the singular vectors corresponding to the top-$k$ singular vectors, and constructing the matrix $\mathbf{A}_k = (\mathbf{U}\boldsymbol{\Sigma})_k \mathbf{V}_k^\top$. Here, $(\mathbf{U}\boldsymbol{\Sigma})_k \in \mathbb{R}^{n \times k}$ is the sub-matrix obtained by selecting the first $k$ columns of $\mathbf{U}\boldsymbol{\Sigma}$, and $\mathbf{V}_k^\top \in \mathbb{R}^{k \times d}$ is the sub-matrix obtained by selecting the first $k$ rows of $\mathbf{V}^\top$. Subsequently, these low-rank factors are quantized with uniform scalar quantizers Q and Q$'$ (having bit-budgets B and B$'$ respectively). The algorithm pseudocode is given in Alg. 2 and we provide an upper bound to the approximation error in Proposition H.1. We use the notation $\widetilde{\mathbf{U}} = \mathbf{U}\boldsymbol{\Sigma} \in \mathbb{R}^{n \times d}$.

Algorithm 2: **Direct-SVD quant.**: Directly quantizing the optimal low-rank factorization

---

**Input**    :Matrix $\mathbf{A} \in \mathbb{R}^{n \times d}$, target rank $k$, Quantizers Q and Q$'$
**Output** :Factorization: $\mathbf{L}\mathbf{R}$ where $\mathbf{L} \in \mathbb{R}^{n \times k}$, $\mathbf{R} \in \mathbb{R}^{k \times d}$

1 Compute SVD and get $\mathbf{A} = \mathbf{U}\boldsymbol{\Sigma}\mathbf{V}^\top$
2 Extract the top-$k$ left (scaled) and right singular vectors and get $\widetilde{\mathbf{U}}_k = (\mathbf{U}\boldsymbol{\Sigma})_k$ and $\mathbf{V}_k^\top$
3 Quantize the factors individually and get $\mathbf{L} \leftarrow Q(\widetilde{\mathbf{U}}_k)$ and $\mathbf{R} \leftarrow Q'(\mathbf{V}_k)$
4 **return $\mathbf{L} \in \mathbb{R}^{n \times k}, \mathbf{R} \in \mathbb{R}^{k \times d}$**

---

**Proposition H.1.  (Direct-SVD quant. approximation error (formal))** *Let our matrix $\mathbf{A} \in \mathbb{R}^{n \times d}$ have maximum singular value $\sigma_1$. Suppose our target rank is $k$, and for some small $\epsilon$ such that $0 < \epsilon \leq 3k\sigma_1^2$, the dynamic ranges of quantizers Q and Q$'$ are set to be $\sigma_1$ and $1$ respectively with bit-budgets satisfying*

$$\mathrm{B} \geq \log_2 \left( \sigma_1 \sqrt{\frac{3nk}{\epsilon}} + 1 \right) \quad and, \quad \mathrm{B}' \geq \log_2 \left( \sigma_1 \sqrt{\frac{3dk}{\epsilon}} + 1 \right).$$

*Then, the factorization returned by direct-SVD quantization in Alg. 2 satisfies*

$$\mathbb{E} \left\| \mathbf{L}\mathbf{R} - \mathbf{A} \right\|_F^2 \leq \left\| \mathbf{A}_k - \mathbf{A} \right\|_F^2 + \epsilon,$$

*where the expectation is over the inherent stochasticity of the quantizers Q and Q$'$.*

*Proof.*  Let us denote the quantization error matrices as $\mathbf{E} = Q(\widetilde{\mathbf{U}}_k) - \widetilde{\mathbf{U}}_k$ and $\mathbf{E}' = Q'(\mathbf{V}_k^\top) - \mathbf{V}_k^\top$. The approximation error is given by

$$\mathbb{E} \left\| Q(\widetilde{\mathbf{U}}_k) Q' \left( \mathbf{V}_k^\top \right) - \mathbf{A} \right\|_F^2 = \mathbb{E} \left\| Q(\widetilde{\mathbf{U}}_k) \left( \mathbf{V}_k^\top + \mathbf{E}' \right) - \mathbf{A} \right\|_F^2$$

$$\overset{(i)}{=} \underbrace{\mathbb{E} \left\| Q(\widetilde{\mathbf{U}}_k) \mathbf{V}_k^\top - \mathbf{A} \right\|_F^2}_{T_1} + \underbrace{\mathbb{E} \left\| Q(\widetilde{\mathbf{U}}_k) \mathbf{E}' \right\|_F^2}_{T_2}, \tag{96}$$

where the cross term disappears in (i) because when $Q'$ is unsaturated, $\mathbb{E}_{Q'}[\mathbf{E}'] = \mathbf{0}$, and

$$\mathbb{E}_{Q'}\left[\mathrm{Tr}\left(\left(Q(\widetilde{\mathbf{U}}_k)\mathbf{V}_k^\top - \mathbf{A}\right)^\top Q(\widetilde{\mathbf{U}}_k)\,\mathbb{E}_{Q'}[\mathbf{E}']\right)\right] = 0.$$

**Analyzing term** $\mathrm{T}_1$: The first term in (96) can be upper bounded as

$$\mathbb{E}\left\|Q(\widetilde{\mathbf{U}}_k)\mathbf{V}_k^\top - \mathbf{A}\right\|_{\mathrm{F}}^2 = \mathbb{E}\left\|\left(\widetilde{\mathbf{U}}_k + \mathbf{E}\right)\mathbf{V}_k^\top - \mathbf{A}\right\|_{\mathrm{F}}^2 \overset{(i)}{=} \|\mathbf{A}_k - \mathbf{A}\|_{\mathrm{F}}^2 + \mathbb{E}\left\|\mathbf{E}\mathbf{V}_k^\top\right\|_{\mathrm{F}}^2. \tag{97}$$

Here, in (i), we utilize the fact that $\mathbf{A}_k = \widetilde{\mathbf{U}}_k\mathbf{V}_k^\top$ and the cross term vanishes once again as quantizer $Q$ is unsaturated, i.e., $\mathbb{E}[\mathbf{E}] = \mathbf{0}$ and,

$$\mathrm{Tr}\left(\left(\widetilde{\mathbf{U}}_k\mathbf{V}_k^\top - \mathbf{A}\right)^\top \mathbb{E}_Q[\mathbf{E}]\mathbf{V}_k^\top\right) = 0.$$

The second term in (97) is

$$\mathbb{E}\left\|\mathbf{E}\mathbf{V}_k^\top\right\|_{\mathrm{F}}^2 = \mathbb{E}\left[\mathrm{Tr}\left(\mathbf{E}\mathbf{V}_k^\top\mathbf{V}_k\mathbf{E}^\top\right)\right] \overset{(i)}{=} \mathrm{Tr}\left(\mathbb{E}[\mathbf{E}^\top\mathbf{E}]\right) = \sum_{i=1}^k \left(\mathbb{E}[\mathbf{E}^\top\mathbf{E}]\right)_{ii} \overset{(ii)}{\leq} nk\frac{\Delta^2}{4}. \tag{98}$$

Here, (i) follows because $\mathbf{V}_k^\top\mathbf{V}_k = \mathbf{I}_k$, and (ii) follows because $\mathbb{E}[\mathbf{E}^\top\mathbf{E}]$ is a diagonal matrix as

$$\left(\mathbb{E}[\mathbf{E}^\top\mathbf{E}]\right)_{ij} = \sum_{\ell=1}^n \mathbb{E}[E_{\ell i}E_{\ell j}] = \begin{cases} n\,\mathrm{Var}\left(E_{\ell i}^2\right) \leq \frac{n\Delta^2}{4} & \text{for } i = j, \\ \sum_{\ell=1}^n \mathbb{E}[E_{\ell i}]\mathbb{E}[E_{\ell j}] = 0 & \text{for } i \neq j. \end{cases} \tag{99}$$

So, (97) can be upper bounded as

$$\mathbb{E}\left\|Q(\widetilde{\mathbf{U}}_k)\mathbf{V}_k^\top - \mathbf{A}\right\|_{\mathrm{F}}^2 \leq \|\mathbf{A}_k - \mathbf{A}\|_{\mathrm{F}}^2 + nk\frac{\Delta^2}{4}. \tag{100}$$

**Analyzing term** $\mathrm{T}_2$: We upper bound the second term in (96) as

$$\mathbb{E}\left\|Q(\widetilde{\mathbf{U}}_k)\mathbf{E}'\right\|_{\mathrm{F}}^2 = \mathbb{E}\left\|\left(\widetilde{\mathbf{U}}_k + \mathbf{E}\right)\mathbf{E}'\right\|_{\mathrm{F}}^2 \overset{(i)}{=} \mathbb{E}\left\|\widetilde{\mathbf{U}}_k\mathbf{E}'\right\|_{\mathrm{F}}^2 + \mathbb{E}\left\|\mathbf{E}\mathbf{E}'\right\|_{\mathrm{F}}^2, \tag{101}$$

where the cross term vanishes in (i) because $\mathbb{E}_{Q'}\left[\mathrm{Tr}\left((\widetilde{\mathbf{U}}_k\mathbf{E}')^\top \mathbb{E}_Q[\mathbf{E}]\,\mathbf{E}'\right)\right] = 0$. The first term in (101) is

$$\begin{aligned}
\mathbb{E}\left\|\widetilde{\mathbf{U}}_k\mathbf{E}'\right\|_{\mathrm{F}}^2 &= \mathbb{E}\left[\mathrm{Tr}\left(\mathbf{E}'^\top\widetilde{\mathbf{U}}_k^\top\widetilde{\mathbf{U}}_k\mathbf{E}'\right)\right] \\
&= \mathbb{E}\left[\mathrm{Tr}\left(\widetilde{\mathbf{U}}_k^\top\widetilde{\mathbf{U}}_k\mathbf{E}'\mathbf{E}'^\top\right)\right] \\
&\overset{(i)}{=} \sum_{i=1}^k \left(\widetilde{\mathbf{U}}_k^\top\widetilde{\mathbf{U}}_k\right)_{ii}\left(\mathbb{E}[\mathbf{E}'\mathbf{E}'^\top]\right)_{ii} \\
&\overset{(ii)}{\leq} d\frac{\Delta'^2}{4}\sum_{i=1}^k \left(\widetilde{\mathbf{U}}_k^\top\widetilde{\mathbf{U}}_k\right)_{ii} \overset{(iii)}{\leq} d\frac{\Delta'^2}{4}\sum_{i=1}^k \sigma_i^2 \leq dk\sigma_1^2\frac{\Delta'^2}{4}.
\end{aligned} \tag{102}$$

Here, (i) and (ii) follow because $\mathbb{E}[\mathbf{E}'\mathbf{E}'^\top]$ is a diagonal matrix following the same argument as (99), i.e.,

$$\left(\mathbb{E}[\mathbf{E}'\mathbf{E}'^\top]\right)_{ij} = \sum_{\ell=1}^n \mathbb{E}[E'_{i\ell}E'_{j\ell}] = \begin{cases} d\,\mathrm{Var}\left(E_{i\ell}'^2\right) \leq \frac{d\Delta'^2}{4} & \text{for } i = j, \\ \sum_{\ell=1}^d \mathbb{E}[E'_{i\ell}]\mathbb{E}[E'_{j\ell}] = 0 & \text{for } i \neq j. \end{cases} \tag{103}$$

Finally, (iii) follows because the $i^{\mathrm{th}}$ column of $\widetilde{\mathbf{U}}_k$ is $\sigma_i\mathbf{u}_i$, so $\left(\widetilde{\mathbf{U}}_k^\top\widetilde{\mathbf{U}}_k\right)_{ii} = (\sigma_i\mathbf{u}_i)^\top(\sigma_i\mathbf{u}_i) = \sigma_i^2\|\mathbf{u}_i\|_2^2 = \sigma_i^2$.

The second term in (101) is

$$\begin{aligned}
\mathbb{E}\|\mathbf{E}\mathbf{E}'\|_{\mathrm{F}}^2 &= \mathbb{E}\left[\mathrm{Tr}\left(\mathbf{E}'^\top\mathbf{E}^\top\mathbf{E}\mathbf{E}'\right)\right] \\
&= \mathbb{E}_Q\left[\mathrm{Tr}\left(\mathbf{E}^\top\mathbf{E}\,\mathbb{E}_{Q'}\left[\mathbf{E}'\mathbf{E}'^\top\right]\right)\right] \\
&\overset{(i)}{=} \mathbb{E}_Q\left[\sum_{i=1}^k \left(\mathbf{E}^\top\mathbf{E}\right)_{ii}\left(\mathbb{E}_{Q'}[\mathbf{E}'\mathbf{E}'^\top]\right)_{ii}\right]
\end{aligned}$$

$$\leq \frac{d\Delta'^2}{4} \sum_{i=1}^{k} \mathbb{E}_Q \left[ \left( \mathbf{E}^\top \mathbf{E} \right)_{ii} \right] \overset{\text{(ii)}}{\leq} k \frac{n\Delta^2}{4} \frac{d\Delta'^2}{4}. \tag{104}$$

Here, (i) and (ii) follow because $\mathbb{E}[\mathbf{E}'\mathbf{E}'^\top]$ and $\mathbb{E}[\mathbf{E}^\top \mathbf{E}]$ are diagonal, which can be seen using similar arguments as in (99). So, (101) is

$$\mathbb{E} \left\| Q(\widetilde{\mathbf{U}}_k) Q' \left( \mathbf{V}_k^\top \right) - \mathbf{A} \right\|_F^2 \leq \| \mathbf{A}_k - \mathbf{A} \|_F^2 + nk \frac{\Delta^2}{4} + dk\sigma_1^2 \frac{\Delta'^2}{4} + ndk \frac{\Delta^2}{4} \frac{\Delta'^2}{4}. \tag{105}$$

**Choice of dynamic range for quantizers** $Q$ **and** $Q'$: Using Lemma B.3, the max-norm of the input to the first quantizer can be upper bounded as $\|\widetilde{\mathbf{U}}_k\|_{\max} \leq \|\widetilde{\mathbf{U}}_k\|_2 = \sigma_1$, and that of the second quantizer is $\|\mathbf{V}_k\|_{\max} \leq \|\mathbf{V}_k\|_2 = 1$. Therefore, choosing the dynamic ranges of $Q$ and $Q'$ to be $\sigma_1$ and 1 respectively, would ensure that they remain unsaturated, and (105) can be rewritten as

$$\mathbb{E} \left\| Q(\widetilde{\mathbf{U}}_k) Q' \left( \mathbf{V}_k^\top \right) - \mathbf{A} \right\|_F^2 \leq \| \mathbf{A}_k - \mathbf{A} \|_F^2 + \frac{nk\sigma_1^2}{(2^B - 1)^2} + \frac{dk\sigma_1^2}{(2^{B'} - 1)^2} + k \frac{n\Delta^2}{4} \frac{d\Delta'^2}{4}. \tag{106}$$

**Choice of bit-budgets** $B$ **and** $B'$: For a given $\epsilon > 0$, suppose we choose $B$ and $B'$ so that

$$B \geq \log_2 \left( \sigma_1 \sqrt{\frac{3nk}{\epsilon}} + 1 \right) \quad \text{and,} \quad B' \geq \log_2 \left( \sigma_1 \sqrt{\frac{3dk}{\epsilon}} + 1 \right).$$

With such a choice, the second and third terms in (106) would not exceed $\epsilon$. Furthermore, the last term of (106) will be

$$k \frac{n\Delta^2}{4} \frac{d\Delta'^2}{4} \leq dnk \frac{\epsilon}{3nk} \frac{\epsilon}{3dk\sigma_1^2} \leq \frac{\epsilon}{3} \quad \text{whenever } \epsilon < 3k\sigma_1^2.$$

This completes the proof. $\qquad\qquad\qquad\qquad\qquad\qquad\qquad\qquad\qquad\qquad\qquad\qquad\square$

Prop. H.1 states that when $n \approx d$, in order to guarantee an $\epsilon$-additive error approximation with respect to the best rank-$k$ approximation, we require (ignoring constant multiplicative factors inside the $\log_2(\cdot)$), a total budget (ref. to Tab. 1) of

$$\frac{n}{n + d} B + \frac{d}{n + d} B' \approx \frac{1}{2} (B + B') = \frac{1}{2} \log_2 \left( \frac{3k\sigma_1^2}{\epsilon} \sqrt{nd} \right) \text{ bits.} \tag{107}$$

# I  LPLR-SVD algorithm: Approximation error analysis

LPLR-SVD is a simpler variant of our LPLR algorithm in which instead of approximating the range space of $\mathbf{A}$ using $\mathbf{AS}$, we compute the full-SVD of $\mathbf{A} \in \mathbb{R}^{n \times d}$ as $\mathbf{A} = \mathbf{U}\mathbf{\Sigma}\mathbf{V}^\top$, where $\mathbf{U} \in \mathbb{R}^{n \times n}$, $\mathbf{\Sigma} \in \mathbb{R}^{n \times d}$, and $\mathbf{V} \in \mathbb{R}^{d \times d}$. The pseudocode of LPLR-SVD is provided in Alg. 3.

We first compute the SVD, $\mathbf{A} = \mathbf{U}\mathbf{\Sigma}\mathbf{V}^\top$, followed by quantizing the singular vectors corresponding to the top-$k$ singular values, scaled by the singular values. We denote $\widetilde{\mathbf{U}} = \mathbf{U}\mathbf{\Sigma}$, and the sub-matrix formed by the first $k$ columns as $\widetilde{\mathbf{U}}_k$. We quantize this basis to get $\mathbf{L} = Q(\widetilde{\mathbf{U}}_k)$ as the first low-rank factor. Subsequently, we project the columns of $\mathbf{A}$ onto the subspace spanned by this quantized basis, i.e., we solve the optimization problem

$$\min_{\mathbf{W} \in \mathbb{R}^{k \times d}} \left\| Q(\widetilde{\mathbf{U}}_k) \mathbf{W} - \mathbf{A} \right\|_F^2 \tag{108}$$

The solution of this is given by $\mathbf{W}^* = Q(\widetilde{\mathbf{U}}_k)^\dagger \mathbf{A}$. The second low rank factor is given by quantizing it again as $\mathbf{R} = Q' \left( Q(\widetilde{\mathbf{U}}_k)^\dagger \mathbf{A} \right)$.

The following result provides an upper bound on the approximation error of LPLR-SVD. The proof of this result is similar to Thm. 3.2 without the complications arising from Gaussian concentration.

**Theorem I.1. (LPLR-SVD approximation error (formal))** *Let our matrix* $\mathbf{A} \in \mathbb{R}^{n \times d}$ *have singular values* $\sigma_1 \geq \ldots \sigma_k \geq \ldots$, *and bounded row norms* $\|\mathbf{a}^{(i)}\| \leq R$. *Let* $\kappa(\mathbf{A}_k) = \sigma_1/\sigma_k$ *be the condition number of the* $\mathbf{A}_k$, *i.e., the best rank-$k$ approximation of* $\mathbf{A}$. *Consider some sufficiently small* $\epsilon$ *that satisfies* $0 < \epsilon < 4k\sigma_1^2$, *and suppose the dynamic ranges of quantizers* $Q$ *and* $Q'$ *are set to be* $\sigma_1$ *and*

**Algorithm 3: LPLR-SVD**: LPLR factorization via Singular Value Decomposition

---

**Input** : Matrix $\mathbf{A} \in \mathbb{R}^{n \times d}$, target rank $k$, Quantizers Q and Q$'$
**Output** : Factorization: **LR** where $\mathbf{L} \in \mathbb{R}^{n \times k}$, $\mathbf{R} \in \mathbb{R}^{k \times d}$

1 Compute SVD and get $\mathbf{A} = \mathbf{U}\mathbf{\Sigma}\mathbf{V}^\top$

2 Extract the top-$k$ left to get an approximate basis for the range space of $\mathbf{A}$ as $\widetilde{\mathbf{U}}_k = (\mathbf{U}\mathbf{\Sigma})_k$

3 Quantize and get an approximate basis $\mathrm{Q}(\widetilde{\mathbf{U}}_k)$

4 Find $\mathbf{W}^* = \arg\min_{\mathbf{W}} \left\| \mathrm{Q}(\widetilde{\mathbf{U}}_k)\mathbf{W} - \mathbf{A} \right\|_{\mathrm{F}}^2$

5 Quantize $\mathbf{W}^*$ using quantizer Q$'$ to get Q$'(\mathbf{W}^*)$

6 **return** *Low-rank and low-precision approximation* **LR** *where* $\mathbf{L} = \mathrm{Q}(\widetilde{\mathbf{U}}_k)$, $\mathbf{R} = \mathrm{Q}'(\mathbf{W}^*)$.

---

$2\kappa(\mathbf{A}_k)$ *respectively. Furthermore, suppose the bit-budgets of the quantizers satisfy*

$$\mathrm{B} \geq \max\{\mathrm{B}_1, \mathrm{B}_2\} \quad and, \quad \mathrm{B}' \geq \log_2\left(4\sigma_1 \kappa(\mathbf{A}_k)\sqrt{\frac{dk}{\epsilon}} + 1\right),$$

*where,*

$$\mathrm{B}_1 = \log_2\left(2\sigma_1\sqrt{\frac{nk}{\epsilon}} + 1\right),$$

*and,*

$$\mathrm{B}_2 = \log_2\left(\frac{4C\kappa(\mathbf{A}_k)}{\log 2}\left(\sqrt{n} + \sqrt{k} + \sqrt{\log\left(\frac{8n\mathrm{R}^2}{\epsilon}\right)}\right) + 1\right).$$

*Then, the low-precision and low-rank factorization returned by Alg. 3 satisfies*

$$\mathbb{E}\|\mathbf{LR} - \mathbf{A}\|_{\mathrm{F}}^2 \leq \|\mathbf{A}_k - \mathbf{A}\|_{\mathrm{F}}^2 + \epsilon.$$

*Proof.* Since the solution of $\mathbf{W}^* = \arg\min_{\mathbf{W}} \left\| \mathrm{Q}(\widetilde{\mathbf{U}}_k)\mathbf{W} - \mathbf{A} \right\|_{\mathrm{F}}^2$ is available in closed form as $\mathbf{W}^* = \mathrm{Q}(\widetilde{\mathbf{U}}_k)^\dagger \mathbf{A}$, we have the low rank factorization as $\mathbf{A} \approx \mathrm{Q}(\widetilde{\mathbf{U}}_k)\mathrm{Q}'\left(\mathrm{Q}(\widetilde{\mathbf{U}}_k)^\dagger \mathbf{A}\right)$. Let us denote the quantization error matrices as $\mathbf{E}' = \mathrm{Q}'\left(\mathrm{Q}(\widetilde{\mathbf{U}}_k)^\dagger \mathbf{A}\right) - \mathrm{Q}(\widetilde{\mathbf{U}}_k)^\dagger \mathbf{A}$, and $\mathbf{E} = \mathrm{Q}(\widetilde{\mathbf{U}}_k) - \widetilde{\mathbf{U}}_k$. Then, the approximation error is

$$\mathbb{E}\left\| \mathrm{Q}(\widetilde{\mathbf{U}}_k)\mathrm{Q}'\left(\mathrm{Q}(\widetilde{\mathbf{U}}_k)^\dagger \mathbf{A}\right) - \mathbf{A} \right\|_{\mathrm{F}}^2 = \mathbb{E}\left\| \mathrm{Q}(\widetilde{\mathbf{U}}_k)\left(\mathrm{Q}(\widetilde{\mathbf{U}}_k)^\dagger \mathbf{A} + \mathbf{E}'\right) - \mathbf{A} \right\|_{\mathrm{F}}^2$$

$$= \underbrace{\mathbb{E}\left\| \mathrm{Q}(\widetilde{\mathbf{U}}_k)\mathrm{Q}(\widetilde{\mathbf{U}}_k)^\dagger \mathbf{A} - \mathbf{A} \right\|_{\mathrm{F}}^2}_{\mathrm{T}_1} + \underbrace{\mathbb{E}\left\| \mathrm{Q}(\widetilde{\mathbf{U}}_k)\mathbf{E}' \right\|_{\mathrm{F}}^2}_{\mathrm{T}_2}, \quad (109)$$

where the last equality follows the because the quantizer Q$'$ is unbiased. Here, the $\mathbb{E}$ is over the stochasticity of the quantizers. This decomposition is similar to (69), and can be treated similarly. The term $\mathrm{T}_1$ consists of the low rank approximation error and the error from the first quantizer, whereas the term $\mathrm{T}_2$ consists of error from the second quantizer. We follow steps similar to the proof of Lemma G.1 which are detailed below.

**Analyzing term** $\mathrm{T}_1$: Using similar reasoning as (71), this term can be written as

$$\mathbb{E}\left\| \mathrm{Q}(\widetilde{\mathbf{U}}_k)\mathrm{Q}(\widetilde{\mathbf{U}}_k)^\dagger \mathbf{A} - \mathbf{A} \right\|_{\mathrm{F}}^2 \leq \mathbb{E}\left\| \mathrm{Q}(\widetilde{\mathbf{U}}_k)\widetilde{\mathbf{U}}_k^\dagger \mathbf{A}_k - \mathbf{A} \right\|_{\mathrm{F}}^2 = \mathbb{E}\left\| \mathrm{Q}(\widetilde{\mathbf{U}}_k)\mathbf{V}_k^\top - \mathbf{A} \right\|_{\mathrm{F}}^2 \quad (110)$$

This is the same as (100) in the analysis of direct-SVD quant., and hence can be upper bounded as

$$\mathbb{E}\left\| \mathrm{Q}(\widetilde{\mathbf{U}}_k)\mathrm{Q}(\widetilde{\mathbf{U}}_k)^\dagger \mathbf{A} - \mathbf{A} \right\|_{\mathrm{F}}^2 \leq \|\mathbf{A}_k - \mathbf{A}\|_{\mathrm{F}}^2 + nk\frac{\Delta^2}{4}. \quad (111)$$

**Analyzing term** $T_2$: This term is again the same as (101) in the analysis of direct-SVD quant., and can be upper bounded as

$$\mathbb{E}\left\|Q(\widetilde{\mathbf{U}}_k)\mathbf{E}'\right\|_F^2 \le dk\sigma_1^2 \frac{\Delta'^2}{4} + k\frac{n\Delta^2}{4}\frac{d\Delta'^2}{4}. \tag{112}$$

Using (111) and (112), we get:

$$\mathbb{E}\left\|Q(\widetilde{\mathbf{U}}_k)Q'\left(Q(\widetilde{\mathbf{U}}_k)^\dagger\mathbf{A}\right) - \mathbf{A}\right\|_F^2 \le \|\mathbf{A}_k - \mathbf{A}\|_F^2 + \underbrace{nk\frac{\Delta^2}{4}}_{T_3} + \underbrace{dk\sigma_1^2\frac{\Delta'^2}{4}}_{T_4} + \underbrace{k\frac{n\Delta^2}{4}\frac{d\Delta'^2}{4}}_{T_5}. \tag{113}$$

**Choice of dynamic range for quantizers** $Q$: Since the input to the quantizer is $\widetilde{\mathbf{U}}_k$, we need an upper bound on $\|\widetilde{\mathbf{U}}_k\|_{\max}$. The $i^{\text{th}}$ column of $\widetilde{\mathbf{U}}_k$ is $(\widetilde{\mathbf{U}}_k)_i = \sigma_i\mathbf{u}_i$, where $1 \le i \le k$. From Lemma B.3,

$$\|\widetilde{\mathbf{U}}_k\|_{\max} \le \|\widetilde{\mathbf{U}}_k\|_2 = \sigma_1. \tag{114}$$

Note that this upper bound is tight in the worst-case sense, i.e., if we consider all matrices with bounded spectral norm, we have $\sup_{\mathbf{X}\,:\,\mathbf{X}=\mathbf{U}\mathbf{\Sigma}\mathbf{V}^\top, \|\mathbf{X}\|_2 \le r}\|\widetilde{\mathbf{U}}_k\|_{\max} = r$. Hence, the dynamic range of quantizer $Q$ is set as $\sigma_1$.

**Choice of dynamic range for quantizers** $Q'$: On the other hand, the input to quantizer $Q'$ is $Q(\widetilde{\mathbf{U}}_k)^\dagger\mathbf{A}$. We adopt an approach similar to the proof of Lemma E.2 to upper bound the max-norm of $Q(\widetilde{\mathbf{U}}_k)^\dagger\mathbf{A}$ as follows:

$$\left\|Q(\widetilde{\mathbf{U}}_k)^\dagger\mathbf{A}\right\|_{\max} \overset{(i)}{\le} \left\|Q(\widetilde{\mathbf{U}}_k)^\dagger\mathbf{A}\right\|_2 \overset{(ii)}{\le} \left\|Q(\widetilde{\mathbf{U}}_k)^\dagger\right\|_2 \sigma_1, \tag{115}$$

where (i) and (ii) follow from Lemmas B.3 and B.4 respectively. We upper bound $\left\|Q(\widetilde{\mathbf{U}}_k)^\dagger\right\|_2$ as:

$$\left\|Q(\widetilde{\mathbf{U}}_k)^\dagger\right\|_2 \overset{(i)}{\le} \left(\sigma_{\min}\left(Q(\widetilde{\mathbf{U}}_k)\right)\right)^{-1} = \left(\sigma_{\min}\left(\widetilde{\mathbf{U}}_k + \mathbf{E}\right)\right)^{-1}$$

$$\overset{(ii)}{\le} \left(\sigma_{\min}(\widetilde{\mathbf{U}}_k) - \|\mathbf{E}\|_2\right)^{-1} = (\sigma_k - \|\mathbf{E}\|_2)^{-1}. \tag{116}$$

Here, (i) follows because the singular values of $Q(\widetilde{\mathbf{U}}_k)^\dagger$ are inverses of the singular values of $Q(\widetilde{\mathbf{U}}_k)$, and (ii) follows from Lemma B.5.

We are now left with upper bounding $\|\mathbf{E}\|_2$. Since a choice of $\sigma_1$ for the dynamic range of $Q$ ensures that quantizer $Q$ remains unsaturated, the entries of $\mathbf{E}$ are bounded as:

$$|E_{ij}| \le \Delta = \frac{2\sigma_1}{2^B - 1} \quad \text{for all } i \in [n] \text{ and } j \in [k]. \tag{117}$$

Since $E_{ij}$ is a bounded w.h.p., it is also subgaussian w.h.p. (ref. eq. (16)) with subgaussian norm given by,

$$\|\mathbf{E}\|_{\psi_2} \le \frac{2\sigma_1}{(2^B - 1)\log 2}. \tag{118}$$

From Lemma B.12 we get for some absolute constant $C$,

$$\|\mathbf{E}\|_2 \le \frac{2C\sigma_1}{(2^B - 1)\log 2}\left(\sqrt{n} + \sqrt{k} + t\right) \text{ with probability exceeding } 1 - 2e^{-t^2}. \tag{119}$$

Setting $t = \sqrt{\log\left(\frac{8nR^2}{\epsilon}\right)}$, we get:

$$\|\mathbf{E}\|_2 \le \frac{2C\sigma_1}{(2^B - 1)\log 2}\left(\sqrt{n} + \sqrt{k} + \sqrt{\log\left(\frac{8nR^2}{\epsilon}\right)}\right) \text{ with probability exceeding } 1 - \frac{\epsilon}{4nR^2}. \tag{120}$$

Let us choose our bit-budget $B$ of quantizer $Q$ to be such that $\|\mathbf{E}\|_2 \le \frac{\sigma_k}{2}$, i.e.,

$$B \ge \log_2\left(\frac{4C\kappa(\mathbf{A}_k)}{\log 2}\left(\sqrt{n} + \sqrt{k} + \sqrt{\log\left(\frac{8nR^2}{\epsilon}\right)}\right) + 1\right), \tag{121}$$

where $\kappa(\mathbf{A}_k) = \sigma_1/\sigma_k$ is the condition number of the best rank-$k$ approximation of $\mathbf{A}$. Then, using (116), we have:

$$\left\|\mathrm{Q}(\widetilde{\mathbf{U}}_k)^\dagger \mathbf{A}\right\|_{\max} \leq 2\kappa(\mathbf{A}_k) \quad \text{with probability exceeding } 1 - \frac{\epsilon}{4n\mathrm{R}^2}. \tag{122}$$

So, if we choose the dynamic range of quantized $\mathrm{Q}'$ to be $2\kappa(\mathbf{A}_k)$, it will remain unsaturated with probability exceeding $1 - \frac{\epsilon}{4n\mathrm{R}^2}$, provided that the bit-budget of first quantizer satisfies B satisfies (121).

**Choice of bit-budgets** B **and** B$'$: Referring to (113), the term $\mathrm{T}_3$ is,

$$nk\frac{\Delta^2}{4} = nk\frac{\sigma_1^2}{(2^\mathrm{B}-1)^2} \leq \frac{\epsilon}{4} \quad \text{if B} \geq \log_2\left(2\sigma_1\sqrt{\frac{nk}{\epsilon}}+1\right). \tag{123}$$

The term $\mathrm{T}_4$ is:

$$dk\sigma_1^2\frac{\Delta'^2}{4} = dk\sigma_1^2\frac{4\kappa(\mathbf{A}_k)^2}{(2^{\mathrm{B}'}-1)^2} \leq \frac{\epsilon}{4} \quad \text{if B}' \geq \log_2\left(4\sigma_1\kappa(\mathbf{A}_k)\sqrt{\frac{dk}{\epsilon}}+1\right). \tag{124}$$

With the above choices of B and B$'$, the term $\mathrm{T}_5$ is:

$$ndk\frac{\Delta^2}{4}\frac{\Delta'^2}{4} \leq ndk\frac{\epsilon}{4nk}\frac{\epsilon}{4dk\sigma_1^2} \leq \frac{\epsilon}{4} \quad \text{if} \quad \epsilon \leq 4k\sigma_1^2. \tag{125}$$

**Tying it all together**: Similar to what is done in the proof of Thm. G.2, for the purpose of analysis, we assume that Alg. 3 returns $\mathbf{L} = \mathbf{0}$ and $\mathbf{R} = \mathbf{0}$ if quantizer $\mathrm{Q}'$ gets saturated. In practical implementation, it can easily be checked if either quantizer $\mathrm{Q}$ or $\mathrm{Q}'$ gets saturated or not, and the algorithm can be repeated again with a new realization of the stochastic quantizer $\mathrm{Q}$. Since the choice of dynamic ranges for $\mathrm{Q}$ ensures that it remains unsaturated with a high probability, "reasonably few" realizations would suffice to get at least one good realization in which $\mathrm{Q}$ is unsaturated.

However, in what follows, we assume that that if quantizer $\mathrm{Q}$ gets saturated, then the algorithm returns $\mathbf{0}$ as an estimate of $\mathbf{A}$, resulting in a Frobenius norm error of $\|\mathbf{A}\|_\mathrm{F}$. Since this happen with a very small probability, we show that its contribution to the expected Frobenius norm error of Alg. 3 is small as well. Let us denote the event that quantizer $\mathrm{Q}$ is unsaturated as $\mathcal{E}$. Then, the expected approximation error can be written as

$$\mathbb{E}\left\|\mathbf{LR} - \mathbf{A}\right\|_\mathrm{F}^2 = \mathbb{E}\left[\left\|\mathrm{Q}(\widetilde{\mathbf{U}}_k)\mathrm{Q}'\left(\mathrm{Q}(\widetilde{\mathbf{U}}_k)^\dagger\mathbf{A}\right) - \mathbf{A}\right\|_\mathrm{F}^2 \mathbb{1}_\mathcal{E}\right] + \mathbb{E}\left[\|\mathbf{A}\|_\mathrm{F}^2 \mathbb{1}_{\mathcal{E}^C}\right]$$

$$\overset{(i)}{\leq} \mathbb{E}\left[\left\|\mathrm{Q}(\widetilde{\mathbf{U}}_k)\mathrm{Q}'\left(\mathrm{Q}(\widetilde{\mathbf{U}}_k)^\dagger\mathbf{A}\right) - \mathbf{A}\right\|_\mathrm{F}^2 \mathbb{1}_\mathcal{E}\right] + n\mathrm{R}^2\Pr\left(\mathbb{1}_{\mathcal{E}^C}\right)$$

$$\overset{(ii)}{\leq} \|\mathbf{A}_k - \mathbf{A}\|_\mathrm{F}^2 + nk\frac{\Delta^2}{4} + dk\sigma_1^2\frac{\Delta'^2}{4} + k\frac{n\Delta^2}{4}\frac{d\Delta'^2}{4} + n\mathrm{R}^2\Pr\left(\mathbb{1}_{\mathcal{E}^C}\right), \tag{126}$$

where inequality (i) follows from Asm. 3.1 and the fact that the expectation of indicator function of an event is the probability of the event, and (ii) follows from (113). Using appropriate choices of B and B$'$ and small enough $\epsilon$, from (122), (123), (124), and (125), we have

$$\mathbb{E}\left\|\mathbf{LR} - \mathbf{A}\right\|_\mathrm{F}^2 \leq \|\mathbf{A}_k - \mathbf{A}\|_\mathrm{F}^2 + \frac{\epsilon}{4} + \frac{\epsilon}{4} + \frac{\epsilon}{4} + n\mathrm{R}^2\frac{\epsilon}{4n\mathrm{R}^2} = \|\mathbf{A}_k - \mathbf{A}\|_\mathrm{F}^2 + \epsilon. \tag{127}$$

Note that from (123) and (121), the bit-budget of the quantizers must satisfy:

$$\mathrm{B} \geq \max\{\mathrm{B}_1, \mathrm{B}_2\}, \quad \text{and B}' \geq \log_2\left(4\sigma_1\kappa(\mathbf{A}_k)\sqrt{\frac{dk}{\epsilon}}+1\right) \tag{128}$$

where,

$$\mathrm{B}_1 = \log_2\left(2\sigma_1\sqrt{\frac{nk}{\epsilon}}+1\right), \tag{129}$$

and,

$$\mathrm{B}_2 = \log_2\left(\frac{4C\kappa(\mathbf{A}_k)}{\log 2}\left(\sqrt{n} + \sqrt{k} + \sqrt{\log\left(\frac{8n\mathrm{R}^2}{\epsilon}\right)}\right) + 1\right). \tag{130}$$

This completes the proof. $\qquad\square$

## I.1   Informal version of Thm. I.1

Consider $n \approx d$. From Thm. I.1, we can get a (simplified) asymptotic dependence of the bit-budgets B and B$'$. In what follows, we ignore the constant multiplicative factors inside $\log_2(\cdot)$. Comparing the expressions for $B_1$ and $B_2$, we have

$$B \geq \log_2\left(\sigma_1\sqrt{\frac{nk}{\epsilon}}\right),\tag{131}$$

and,

$$B' \geq \log_2\left(\sigma_1\kappa(\mathbf{A}_k)\sqrt{\frac{dk}{\epsilon}}\right).\tag{132}$$

This gives a total bit-budget of

$$\frac{1}{2}\log_2\left(\sigma_1^2\kappa(\mathbf{A}_k)\frac{k}{\epsilon}\sqrt{nd}\right)\quad\text{bits.}\tag{133}$$

## J   Additional numerical simulations

### J.1   Image Compression

In addition to Fig. 1 in the main paper, we also provide additional image compression experiments on images of different dimensions in Figs. 6, 5, 7, and 8. The values of B and B$'$ are provided in the respective captions of these figures. For every image, the sketch size/target rank $m$ and the bit-budgets B and B$'$ are chosen so as to ensure parity with a naïve quantizer with bit budget $B_{nq}$. In other words, for an $n \times d$ dimensional image, $B \cdot nm + B' \cdot md = B_{nq} \cdot nd$. The values of $n$ and $d$ are also provided in the corresponding captions.

For image compression, we also perform a normalize and shift operation of our DSVD/LPLR/LSVD output in order to adjust the scale and bias of the output image, as described next.

**Normalize and shift**: Let us denote the factorization obtained from executing DSVD/LPLR/LSVD on the input matrix $\mathbf{X} \in \mathbb{R}^{n \times d}$ by $\mathbf{Y} = \mathbf{LR}$. Then, we can improve the estimate provided by this factorization by considering $\alpha\mathbf{Y} + \beta\mathbf{I}_{n \times d}$. The overhead in storing these two additional scalars in full-precision is negligible when $n$ and $d$ are large. The optimal values of $\alpha$ and $\beta$ can be found as:

$$(\alpha^*, \beta^*) = \arg\min_{\alpha,\beta}\|\alpha\mathbf{Y} + \beta\mathbf{I} - \mathbf{X}\|_F^2.\tag{134}$$

Closed-form expressions for $(\alpha^*, \beta^*)$ can be obtained by equating the derivatives of $\|\alpha\mathbf{Y} + \beta\mathbf{I} - \mathbf{X}\|_F^2$ to zero, and solving the resulting linear equations. Doing this algebra yields,

$$\alpha^* = \left(\|\mathbf{Y}\|_F^2 - \frac{\langle\mathbf{Y},\mathbf{I}\rangle^2}{nd}\right)^{-1}\left(\langle\mathbf{X},\mathbf{Y}\rangle - \frac{\langle\mathbf{X},\mathbf{I}\rangle\langle\mathbf{Y},\mathbf{I}\rangle}{nd}\right),$$

$$\text{and,}\quad \beta^* = \frac{\langle\mathbf{X},\mathbf{I}\rangle}{nd} - \frac{\langle\mathbf{Y},\mathbf{I}\rangle}{nd}\alpha^*.\tag{135}$$

### J.2   Wall-clock time for compressing image embeddings

We now report the wall-clock time for computing the low-rank quantized factors via LPLR, LSVD, and DSVD for embedding compression in Tab. 10. This reinforces the validity of our theoretical assertions concerning the computational advantage of LPLR when compared to its alternative methods.

### J.3   High-resolution figures for Llama weight compression

We now provide higher resolutions figures of Figs. 2a and 2b as Figs. 3 and 4, respectively. The code to reproduce these numerical simulations is available at our github repository https://github.com/pilancilab/matrix-compressor.

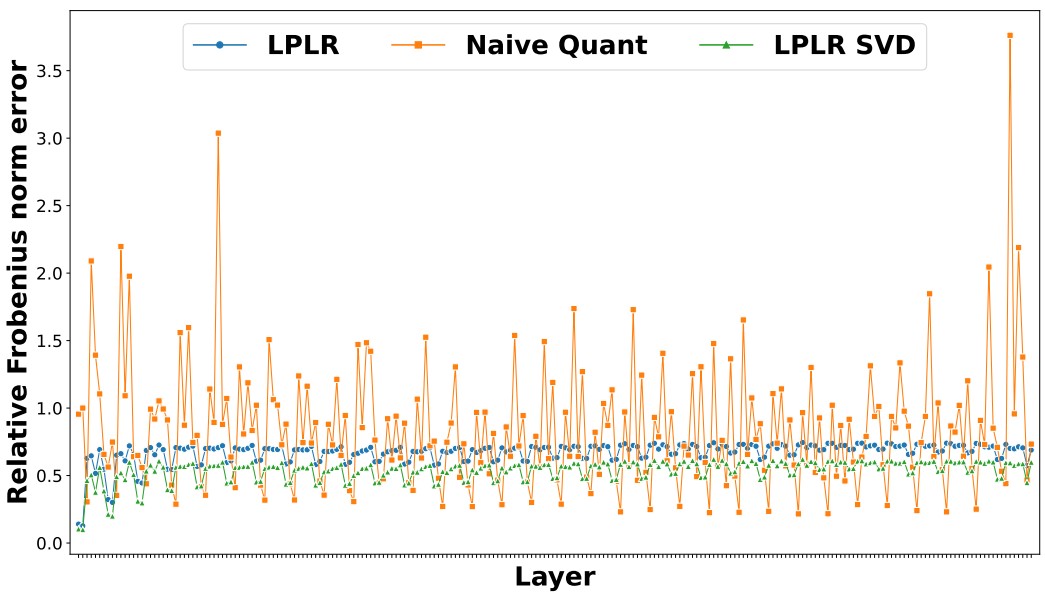

Figure 3: Comparison of LPLR and LPLR-SVD on LlaMa weight matrices with $B = B' = 8$ bits, $B_{nq} = 4$ bits, ordered by the original sequence of layers on the "Layer" - axis. We observe consistently better Frobenius norm error using LPLR and LPLR-SVD, with the exception of specific layers which lend themselves to naïve quantization.

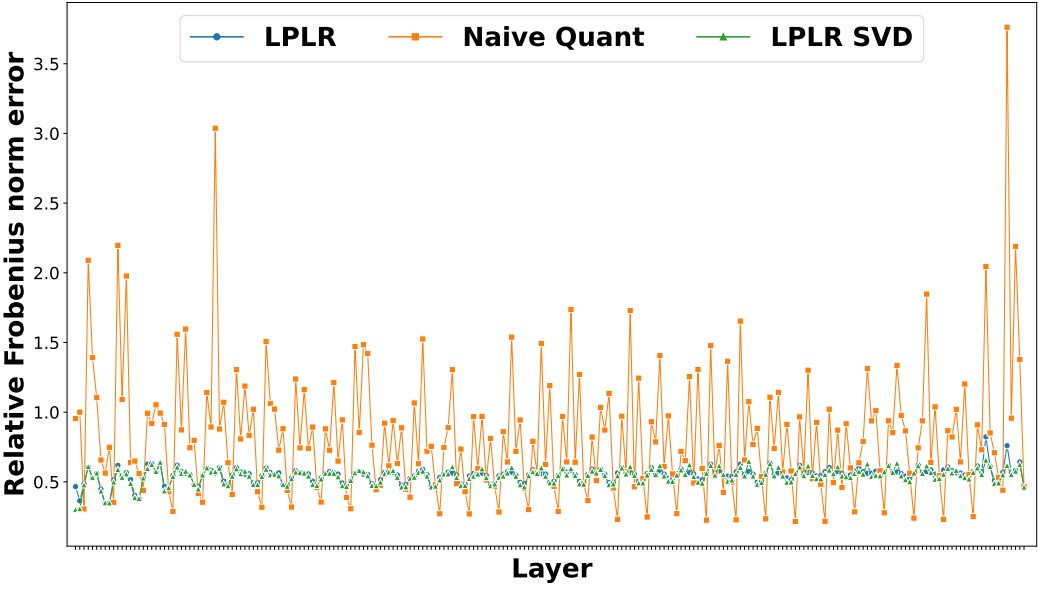

Figure 4: Comparison of LPLR and LPLR-SVD on LlaMa weight matrices with $B = B' = B_{nq} = 4$ bits, ordered by the original sequence of layers on the "Layer" - axis. LPLR and LPLR-SVD demonstrate equivalent performance on a uniform compression budget of 4 bits, while outperforming naïve quant. on almost all layers.

Table 10: Comparison of wall-clock time for computing low rank quantized factors via LPLR, LSVD and DSVD. Each image embedding forms a row of the input matrix, with low rank factors computed for **each** class of the input dataset. Bit-budgets used are $B = B' = 8, B_{nq} = 1$. We report mean and standard deviation.

| Dataset | LPLR | DSVD | LSVD |
|---------|------|------|------|
| CIFAR-10 | $71 \pm 11$ ms | $306 \pm 26$ ms | $312 \pm 8$ ms |
| CIFAR-100 | $14 \pm 3$ ms | $51 \pm 3$ ms | $56 \pm 3$ ms |

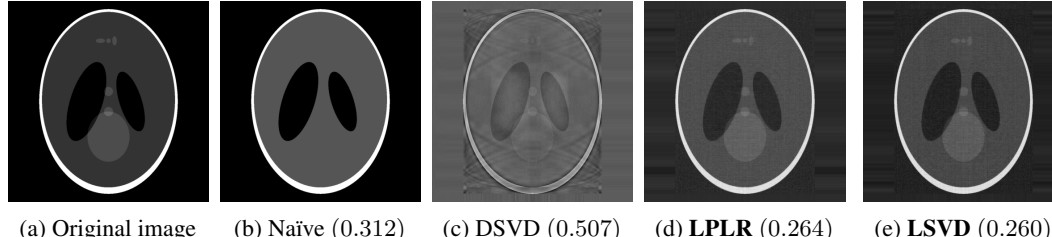

(a) Original image    (b) Naïve (0.312)    (c) DSVD (0.507)    (d) **LPLR** (0.264)    (e) **LSVD** (0.260)

Figure 5: Compression of Shepp-Logan phantom (a standard test image for medical image reconstruction, courtesy: Phantominator, PyPI). $B = 4, B' = 8, B_{nq} = 2, m = 166$. Orig. image dim.: $1000 \times 1000$

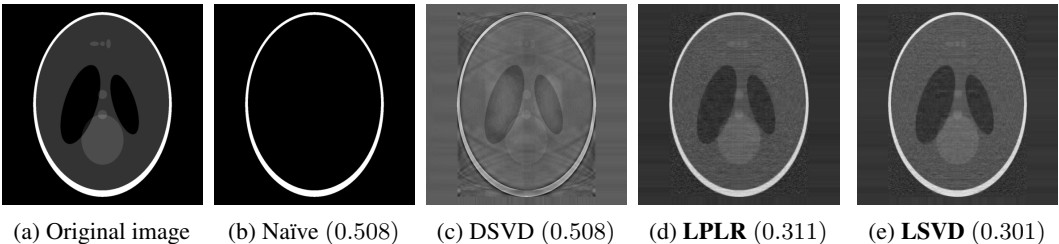

(a) Original image    (b) Naïve (0.508)    (c) DSVD (0.508)    (d) **LPLR** (0.311)    (e) **LSVD** (0.301)

Figure 6: Compression of Shepp-Logan phantom (a standard test image for medical image reconstruction, courtesy: Phantominator, PyPI). $B = 4, B' = 8, B_{nq} = 1, m = 83$. Orig. image dim.: $1000 \times 1000$

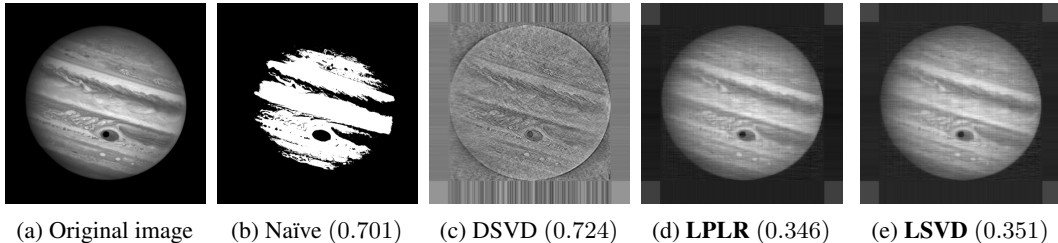

(a) Original image    (b) Naïve (0.701)    (c) DSVD (0.724)    (d) **LPLR** (0.346)    (e) **LSVD** (0.351)

Figure 7: Compression of a Jupiter image showing its Great Red Spot and Ganymede's shadow (courtesy: NASA/ESA Hubble Space Telescope). $B = 2, B' = 8, B_{nq} = 1, m = 110$. Orig. image dim.: $1102 \times 1102$

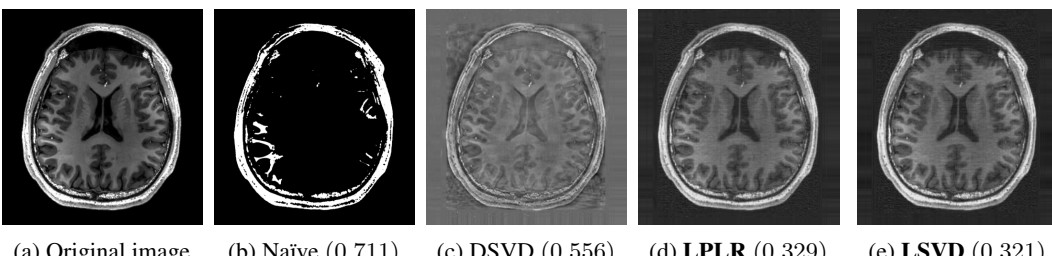

(a) Original image    (b) Naïve (0.711)    (c) DSVD (0.556)    (d) **LPLR** (0.329)    (e) **LSVD** (0.321)

Figure 8: Compression of an MR image of the human brain (courtesy: User Asnaebsa, Wikipedia). $B = 4, B' = 8, B_{nq} = 1, m = 124$ maintains parity. Orig. image dim.: $1534 \times 1433$

# K   Limitations and further discussions

**On uniformly dithered quantizer**: We would like to point out that the approximation error upper bound for our algorithm (derived in Thm. G.2), as well as for the baselines such as direct-SVD quantization (refer to Prop. H.1) holds true only for randomized rounding, or uniformly dithered quantizers. Dithering is preferred due to its ability to produce an unbiased estimate of the input. It offers an advantage by introducing a non-zero probability of quantizing an input to either its ceiling or floor, resulting in reduced variance when averaging multiple independent realizations. Moreover, the unbiasedness of the quantizer output simplifies the analysis. In our experiments, we did not observe any difference while using deterministic rounding instead of dithered rounding.

**Comparison with LPLR with direct-SVD and LPLR-SVD**: As we see from the numerical evaluations (ref. to Tab. 6 and Tab. 7), direct-SVD quant. and LPLR-SVD performs better than our proposed LPLR algorithm (that computes a Gaussian sketch). Intuitively, one would expect the performance of LPLR-SVD and LPLR (with Gaussian sketch) to be similar because the core idea behind both of them is the same – Quantize the first low-rank factor, and then project the columns of A onto the quantized (first) low-rank factor. For LPLR-SVD the basis being quantized (top-k singular vectors) is the exact top-k rangefinder (for the column space of A), whereas in LPLR, we sketch using the Gaussian matrix to get an approximate rangefinder. This sketching error might make LPLR worse over LPLR-SVD. But because sketching also introduces "equalization effect of the Gaussian matrices", we can improve the resolution of the uniform scalar quantizers by choosing a smaller dynamic range. Hence, there is a tradeoff when comparing LPLR and LPLR-SVD – a tradeoff between sketching error and the quantization error.

Let us now focus on LPLR-SVD and direct-SVD quant. We will extrapolate this intuition to a comparison between LPLR and direct-SVD quant. The factorization obtained using direct-SVD quant. is $\mathbf{A} \approx \mathrm{Q}(\widetilde{\mathbf{U}}_k)\mathrm{Q}'(\mathbf{V}_k^\top)$, where the approximation obtained using LPLR-SVD is $\mathbf{A} \approx \mathrm{Q}(\widetilde{\mathbf{U}}_k)\mathrm{Q}'\left(\mathrm{Q}(\widetilde{\mathbf{U}}_k)^\dagger \mathbf{A}\right)$. Here, $\widetilde{\mathbf{U}}_k$ is the $n \times k$ matrix obtained using the first $k$ columns of $\mathbf{U}\boldsymbol{\Sigma}$, where $\mathbf{A} = \mathbf{U}\boldsymbol{\Sigma}\mathbf{V}^\top$ is the full SVD of $\mathbf{A}$. Let us assume $\mathrm{B}' \to \infty$ so that we don't have to worry about the second quantizer. Then, we are left to compare the approximations: $\mathbf{A} \approx \mathrm{Q}(\widetilde{\mathbf{U}}_k)\mathbf{V}_k^\top$ and $\mathbf{A} \approx \mathrm{Q}(\widetilde{\mathbf{U}}_k)\mathrm{Q}(\widetilde{\mathbf{U}}_k)^\dagger \mathbf{A}$. Clearly, the latter has smaller Frobenius norm error because it is the minima of the minimization problem $\arg\min_{\mathbf{W}} \left\|\mathrm{Q}(\widetilde{\mathbf{U}}_k)\mathbf{W} - \mathbf{A}\right\|_\mathrm{F}^2$. So, for $\mathrm{B}' \to \infty$, it is evident that direct-SVD can only perform as good as LPLR-SVD, and not better than that. Furthermore, note that this difference in performance will be small when the bit-budget of the first quantizer, B is large, and it will be large when B is small. When $\mathrm{B} \to \infty$, this difference will be exactly zero. In other words, direct-SVD quant. and LPLR-SVD will be the same factorization when $\mathrm{B} \to \infty$ and $\mathrm{B}' \to \infty$. When both B and B' are finite, direct-SVD can outperform LPLR-SVD depending on which one out of $\mathrm{Q}'\left(\mathrm{Q}(\widetilde{\mathbf{U}}_k)^\dagger \mathbf{A}\right)$ and $\mathrm{Q}'(\mathbf{V}_k^\top)$ is closer to the optimal solution, i.e., $\mathrm{Q}(\widetilde{\mathbf{U}}_k)^\dagger \mathbf{A}$. A limitation of our analysis is that we cannot always definitively predict which one will perform better, because we derive *upper bounds* on the approximation error.

**Comparison of LPLR with naive quantization**: In Tables 1 and 2, we have compared LPLR with other benchmarks. In the column denoting approximation errors, aside from the common quantization error $\epsilon$ present across all rows, DSVD and LPLR introduce an additional term representing the error from low-rank approximation. Consequently, it might appear that naive quantization consistently outperforms these methods. However, it's essential to recognize that compression techniques based on low-rank factorization hold value exclusively when the matrix being compressed is inherently low-rank, meaning that $\|\mathbf{A}_k - \mathbf{A}\|_\mathrm{F}^2$ starts off as small. If $\|\mathbf{A}_k - \mathbf{A}\|_\mathrm{F} = 0$, LPLR can achieve identical error levels as naive quantization, while demanding fewer bits than the latter. In other words, given the same bit-budget, LPLR can achieve a lower level of error compared to naive quantization.

**Matrix compression with and without any computation constraints**: In situations where computational resources are limited, employing the naive strategy emerges as the most cost-effective approach for matrix quantization. Nevertheless, due to its failure to capitalize on a matrix's inherent low-rank arrangement, naive quantization may prove considerably suboptimal. As matrices increase in dimension, accommodating them in memory becomes impractical, rendering approaches like DSVD or LSVD unviable. In such scenarios, our LPLR method stands as a viable alternative,

demanding slightly more computational effort than naive quantization, yet capable of harnessing the low-rank structure for improved approximation accuracy.

If there is no scarcity of computation resources in being able to compute the SVD, and our goal is to just compress an input matrix $\mathbf{A}$, then it is possible to compress the matrix with all the strategies, and choose the best one. However, it must be kept in mind that LPLR provides significant savings with respect to computational complexity, i.e., $O(ndm)$, compared to LPLR-SVD, which has a complexity of $O(nd^2)$. Consequently, if the matrix being compressed is very large, it might not even be feasible to do direct-SVD quant. or LPLR-SVD, given the current memory limitations of available GPUs, and LPLR (with Gaussian sketch) becomes our only option.

Furthermore, a careful examination of Tabs. 6 and 7 shows that even though LPLR-SVD and direct-SVD quant. have a smaller Frobenius norm error than LPLR, the accuracy and weighted F1 scores are still comparable. One aspect of quantization that we have not studied in detail is its inherent regularization effect when testing the performance on test datasets. Whether LPLR leads to a better regularization effect that the other baselines is an open question that will be studied in future works.

**Comparison with existing works on quantized random projections for approximate nearest neighbor search**: Prior works of Li and Li [31] and Li and Li [32] consider the idea of quantized random projections specifically for the the application of approximate nearest neighbor search. Although related, the goals of these works are different from matrix compression considered in our work. These prior works approximate the matrix vector product $\mathbf{A}\mathbf{x}$ by $Q(\mathbf{A}\mathbf{S})Q'(\mathbf{S}^\top\mathbf{x})$. For their setup, in addition to $Q(\mathbf{A}\mathbf{S})$, the random matrix $\mathbf{S}$ needs to be stored in full-precision, in order to process an incoming query $\mathbf{x}$. So, their storage requirement is $nm$ quantized and $dm$ full-precision parameters. In contrast, **LPLR** stores only the quantized entries of $\mathbf{L}$ and $\mathbf{R}$, i.e., $m(n+d)$ quantized entries, and $\mathbf{A} \approx \mathbf{L}\mathbf{R}\mathbf{x}$, implying a smaller storage requirement. Furthermore, since **LPLR** involves computation of $\mathbf{R}\mathbf{x}$, where $\mathbf{R}$ consists of $md$ quantized values, it can leverage modern advancements in hardware primitives for speeding up low-precision computations, such as half and mixed-precision compute.

**Extension to streaming settings**: LPLR as presented in this paper, is used for compression when the entire matrix $\mathbf{A}$ is available. An interesting extension is to consider a variant of LPLR to streaming data settings in order to handle new data points. It is possible to do so using sketching based low rank approximations [68]. The left factor $\mathbf{L}$ is updated with the concatenation of an additional row corresponding to the newly arriving datapoint. The second factor $\mathbf{R}$, which is the solution of a new least squares minimization problem, can be updated using Woodbury matrix inversion lemma.

