# OpenReview forum: "Matrix Compression via Randomized Low Rank and Low Precision Factorization"
_NeurIPS.cc/2023/Conference — NeurIPS 2023 poster_

### Official Review · Reviewer_2AWv · 2023-07-04

**Soundness:** 3 good
**Presentation:** 3 good
**Contribution:** 3 good
**Rating:** 7
**Confidence:** 4

**Summary:**

This work studies the problem of computing a low rank approximation when the low rank factors are under a bit budget constraint, that is, we must output factors L and R with bounded bits such that LR approximates a given input matrix A in the Frobenius norm. The authors show that by incorporating sketching into the quantization procedure, one can get improved bounds, due to the fact that a Gaussian sketch can “flatten” the entries of a vector, which is advantageous when rounding (Appendix D). Empirical results show that this algorithm indeed gives improved results over other naive implementations such as directly rounding SVD factors.

**Strengths:**

The problem of efficiently quantizing low rank approximations is an extremely important problem given that both low rank approximations and low precision is gaining popularity for compressing massive neural networks (https://papers.nips.cc/paper/2020/file/13b919438259814cd5be8cb45877d577-Paper.pdf, https://arxiv.org/abs/2302.03764). This work offers an interesting new method which takes advantage of the “flattening” property of Gaussian sketches in order to obtain improved results for quantization in the context of low rank approximation. This idea is conceptually simple yet interesting. Empirical results are also convincing.

**Weaknesses:**

The contribution is already quite nice, but there are several followup investigations that could strengthen this work much more:
* Are there any lower bounds on the trade-off between the approximation accuracy and the bit budget?
* There are structured sketching transforms such as the Subsampled Randomized Hadamard Transform (see Theorem 2.4 of http://www.cs.cmu.edu/afs/cs/user/dwoodruf/www/wNow3.pdf) which also have the “equalizing” or “flattening” type of behavior, yet can be applied in much faster time, and furthermore also save on the storage of the sketching matrix since it is an integer matrix. Can this be used to get faster implementations in theory/practice?
* Can you report the running time of the experiments?
* Can you comment on whether the bit bounds imply actual savings in the memory usage, or are the practical implementations of bit complexity too crude to capture these improvements?

**Questions:**

* Should put A \approx LR in the abstract
* Some of the references in the Related Works section for Low-rank approximation seem inappropriate for the context. Candes-Tao and Recht et al are not focused on faster algorithms for SVD, but rather a related but different problem of matrix completion. Gradient descent on the low rank factors is indeed one way to solve approximate SVD, which I agree Zhang et al captures, but there may be more appropriate references such as https://arxiv.org/abs/2106.14289 and references within.
* Line 58: you should first establish that n >= d, or else the SVD time should be O(min{nd^2, dn^2})
* Typo (line 77): should be “instead of quantizing x directly”
* Is the randomized rounding in Section 2.1 necessary? Can you get better results with naive rounding?
* Line 177: what do you mean by “saturated” and “unsaturated”? Please define this term, it is unfamiliar to non-experts.

**Limitations:**

Discussion of limitations is limited. This work is mainly theoretical, and there are no potential negative societal impacts.

---

> ### Author Rebuttal · Authors · 2023-08-09
>
> Dear Reviewer,
>
> Thank you for reading our paper and writing the review. We hold in high regard the voluntary nature of the review process, and in what follows, we engage with your concerns.
>
> > Structured sketching
>
> We agree that SRHT sketch also have an equalizing effect, and can indeed lead to faster algorithms to compress a matrix than computing a Gaussian sketch, as Hadamard transform can be computing recursively. We experimented with SRHT but decided not to include the results in our paper for two reasons:
>
> 1. SRHT sketch did not perform better than LPLR with a Gaussian sketch in terms of Frobenius norm error.
> 2. We have done our theoretical analysis with Gaussian sketch.
>
> More importantly, computing the sketch $AS$ is not the most time-consuming step. Solving the minimization problem $argmin_W ||ASW - A||_2^2$, either in closed form or with conjugate gradient descent, is the slowest step. Furthermore, once the low-rank factors $L$ and $R$ are found, only these factors are stored, and not the sketching matrix. We will add a discussion relation to this in App. K as well.
>
> > Lower bounds on the trade-off
>
> This is a very interesting question. It might be possible to derive lower bounds on this tradeoff using covering and packing arguments for the space of all rank-$k$ approximations of a matrix. Such lower bounds are non-trivial and we will mention this in App. K.
>
> > Do bit bounds imply actual savings?
>
> Practical applications of quantization for hardware storage often necessitate working within specific bit budgets, such as 16-bit or 32-bit precision. With this in mind, when implementing our algorithm LPLR in practical contexts, we set values for variables $B$ and $B'$ to align with these hardware precision limitations. Subsequently, the parameter sketch size $m$ becomes an adjustable parameter, allowing us to regulate the compression of the total parameter count, denoted as $(n + d)m$. Given that the total parameters in matrix $A$ amount to $nd$, we show that for sufficiently low-rank matrices, it is feasible to choose $m$ in a manner such that $Bnm + B'md = B_{nq}nd$, where $B_{nq}$ can be as minimal as $1$ bit. This implies that $B_{nq}$ can take on values that are unattainable owing to the inherent hardware limitations.
>
> Moreover, in other scenarios where implementations of bit complexity need not be too crude, the quantization codebook can be customized beyond the confines of non-traditional hardware limitations, we are not restricted solely to the choices of $B$ and $B'$. For instance, when transmitting a matrix over a communication channel, we can employ modulation techniques like amplitude or frequency shift keying. This enables us to use $B = B' = 2$, effectively achieving $4$ quantization levels ($2^2$). Considering that a greater number of amplitude or frequency levels necessitate larger channel resources, the bit bounds derived in our paper reveal the potential for conserving these resources in such situations.
>
> > Randomized vs. naive rounding
>
> Randomized rounding, also known as uniformly dithered quantizer, is preferred due to its ability to produce an unbiased estimate of the input. Dithered quantizers offer an advantage by introducing a non-zero probability of quantizing an input to either its ceiling or floor, resulting in reduced variance when averaging multiple independent realizations. Moreover, the unbiased nature of the quantizer output simplifies the analysis. In our experiments, we did not observe any benefits of using deterministic rounding instead of dithered rounding.
>
> > Saturated and unsaturated quantizer
>
> The dynamic range of a scalar quantizer is defined to the the interval $[-{\rm R}, +{\rm R}]$ as defined in Sec. 2.1. If the input $x$ to the quantizer falls outside this range, i.e., $x > {\rm R}$ or $x < -{\rm R}$, the quantizer is said to be {\it saturated}. We want $\rm R$ to be sufficiently large so that $-{\rm R} \leq x \leq {\rm R}$ is satisfied, so that the quantizer is {\it unsaturated} and the quantized output is an unbiased estimate of the input with bounded variance. We will define this in the main paper.
>
> > Runtime of the experiments
>
> Please refer to the global response (specifically Table 2 and Fig. 3). In Table 2, we compare the wall-clock CPU time for computing the low-rank quantized factors of CIFAR-10 and CIFAR-100 datasets via LPLR, LSVD and DSVD. Each image embedding forms a row of the input matrix, with low rank factors computed for *each* class of the input dataset. It is evident that LPLR significantly outperforms DSVD and LSVD in terms of speed.
>
> Fig. 3 complements the tabulated data by visualizing the runtime involved in two scenarios. The first scenario involves computing the matrix-vector multiplication $\bf Ax$ directly. The second scenario involves approximating this multiplication using a low-rank factorization, specifically $\bf L(Rx)$, where matrices $\bf L$ and $\bf R$ are tall and wide matrices, respectively. This low-rank factorization strategy reduces the overall number of multiplications from $nd$ to $(n+d)m$.
>
> > Misc typos
>
> Thank you for pointing them out. We will proofread and rectify them.

---

> > ### Comment · Reviewer_2AWv · 2023-08-11
> > **Thank you for the responses!**
> >
> > The authors' responses are very helpful in understanding this work. As indicated in my original review, I regard this work highly, and the authors' responses help to increase my confidence in this assessment.
> >
> > I encourage the authors to mention the comment about the most time-consuming step in this procedure in the main text, I find this valuable. I would also appreciate if you mention that naive rounding does not make a difference in practice, I imagine this is much more convenient for practitioners.

---

> > > ### Author Response · Authors · 2023-08-12
> > > **Thank you**
> > >
> > > Dear Reviewer,
> > >
> > > Thank you once again for reading our work, and for providing a thoughtful and constructive feedback. We are happy that our responses have helped clarify your concerns. We will definitely include a mention of the most time-consuming step in the main text, along with emphasizing that naive rounding does not make a difference in practice.
> > >
> > > If you have any further comments or questions, please do not hesitate to reach out.

---

### Official Review · Reviewer_ipPc · 2023-07-06

**Soundness:** 2 fair
**Presentation:** 3 good
**Contribution:** 2 fair
**Rating:** 5
**Confidence:** 2

**Summary:**

This paper introduces a novel low-rank matrix factorization algorithm that is using sketching matrix idea and quantization, such as they do:

1. Use Gaussian RV to generate sketch of the matrix and compute the approximate basis
2. Use Quantization with Q - to get Q(AS)
3. Use Q(AS) and Q' to get  Q'(W)
4. Return Q(AS) and Q'(W)

Authors provide theoretical and numerical analysis of their idea.

**Strengths:**

- introduction section written very well & and very informative and to the point, such as introducing LPLR algorithm briefly, talking about Low-rank approximation and Randomized quantization.

- the introduced algorithm clearly communicated and results are theoretically sound.

**Weaknesses:**

- Abstract seem to be a bit wordy - it would be nice if there were formulations and numbers that grabs readers attention with resutls.

- i like the way motivating the work - with memory constraints - but i am curious is there any application in real life for low rank decomposition that actually saves memory. some examples would be good

- the paper doesn't exactly introduce a new direction - rather seems to be using existing ideas and put them together.

**Questions:**

Table 1 & 2  is a bit puzzling me - it seems that Approximataion Error of Naive Uniform is smaller than LPLR and with cheaper computation cost. Do you think that you can clarify in the table itself  - why LPLR is better ? because it seems to be confusing in the first glance.

**Limitations:**

The are many sketching / matric factorization algorithms, i am puzzled why we only presented Naive Uniform and Direct SVD algorithms as comparison points.

---

> ### Author Rebuttal · Authors · 2023-08-09
>
> Dear Reviewer,
>
> We are grateful for the valuable time you spent in reading our paper and writing the review. We value the fact that the review process is a voluntary endeavor, and in the sections below, we tackle each of your concerns to address and clarify your questions.
>
> > Table 1 & 2 is a bit puzzling me
>
> Tables $1$ and $2$ present the approximation error guarantees for three different compression methods: $\rm (i)$ naive quantization with an error of $\epsilon$, $\rm (ii)$ direct SVD with an error of $\lVert \mathbf{A}_k - \mathbf{A} \rVert_F^2 + \epsilon$, and $\rm (iii)$ LPLR with an error of $(1 + \delta)\lVert \mathbf{A}_k - \mathbf{A} \rVert_F^2 + \epsilon$. At first glance, it might appear that naive quantization is always superior to LPLR since the latter contains additional terms in the approximation error. However, it's crucial to consider that naive quantization also demands more bits than LPLR.
>
> While we do not claim that LPLR will consistently outperform naive quantization, LPLR exhibits better performance when dealing with matrices that are inherently low-rank to begin with. In the case of low-rank matrices $\mathbf{A}$, the first term, $(1 + \delta)\lVert \mathbf{A}_k - \mathbf{A} \rVert_F^2$, representing the error of the best rank-$k$ approximation, is very small (and becomes zero if $\mathbf{A}$ is precisely rank $k$). In such scenarios, both naive quantization and LPLR yield an approximation error of $\epsilon$. However, LPLR achieves this error with fewer bits compared to naive quantization. On the other hand, for matrices that are close to full rank, i.e., with singular values decaying slowly, the term $\lVert \mathbf{A}_k - \mathbf{A} \rVert_F^2$ will not be negligible. In other words, given the same bit-budget, LPLR can achieve a lower level of error compared to naive quantization. We will highlight this in the caption of the tables.
>
> > Why only presented Naive Uniform and Direct SVD algorithms as comparison points.
>
> In this study, we have focused on matrix factorizations that are well-suited for achieving low-rank approximations. To obtain the best rank-$k$ approximation for a given matrix $\mathbf{A}$, we first compute the Singular Value Decomposition (SVD) and then retain the top-$k$ singular values along with their corresponding singular vectors. When considering both low-rank and low-precision requirements, SVD + quantization emerges as a natural benchmark. On the other hand, for matrix quantization, the standard practice is to use naive uniform quantization due to its simplicity. Although other matrix factorizations like $\mathbf{QR}$ or $\mathbf{LU}$ decompositions also provide low-rank approximations, they are suboptimal compared to the SVD-based low-rank approximation method (which is optimal). Hence, they are not expected to outperform direct-SVD when combined with a naive application of uniform quantization. Moreover, concerning matrix compression, it is essential to achieve parity in terms of bit-requirements among various benchmarks for a fair comparison. To accomplish this, direct-SVD, LPLR, and LSVD offer tunable parameters, which can be adjusted. These parameters include the target rank $k$ and the sketch size $m$, which allow us to control the compression ratio and maintain the desired parity.
>
> > Application in real life for low rank decomposition that actually saves memory
>
> Low rank decomposition has found a contemporary use in conserving memory, particularly in compressing neural networks. Besides the sources cited in the initial passage, additional references include:
>
> 1. Y. Idelbayev and M. Á. Carreira-Perpiñán, "Low-Rank Compression of Neural Nets: Learning the Rank of Each Layer," CVPR, 2020.
> 2. T. N. Sainath, B. Kingsbury, V. Sindhwani, E. Arisoy and B. Ramabhadran, "Low-rank matrix factorization for Deep Neural Network training with high-dimensional output targets," ICASSP 2013.
>
> The importance of compressing neural networks stems from several factors, including the need to decrease memory usage, enhance inference speed, and facilitate deployment on devices with limited resources. Low rank decompositions play a pivotal role in achieving these goals by breaking down the initial weight matrices into smaller, organized matrices with fewer parameters.
>
> > The paper doesn't exactly introduce a new direction
>
> We propose an idea which is simple in its execution and pragmatic in its implementation. Our idea is well supported for modern linear algebra libraries, while offering theoretical analysis to effectively identify the regime in which it offers value over conventional, standard methods. Moreover, our method offers a tunable knob to achieve effective bit-rates in software, not currently supported by hardware. For example, in Table $3$, using LPLR with appropriate parameters bits is equivalent to operating at $1$ or $2$-bits per pixel, despite hardware working with a higher bit precision. Moreover, our method is adaptable to future hardware advancements, such as the emergence of $4$-bit GPUs, which can significantly speed up our technique's primitive operations. We believe that the simplicity of our algorithm adds to its appeal and makes it a valuable contribution to the literature, bridging the fields of matrix compression, quantization, and sketching while ensuring compatibility with upcoming hardware developments.
>
> > Abstract seem to be a bit wordy
>
> We appreciate your input on improving the abstract's appeal to readers. We concur with the idea of emphasizing the main outcomes derived from Tables $1$ and $2$, which involve contrasting our novel algorithm with other benchmarks. This contrast could be established through aspects like computational complexity (e.g., highlighting the ${\rm O}(ndm)$ of LPLR in contrast to the ${\rm O}(nd^2)$ of direct SVD quant.). Moreover, we intend to incorporate specific numerical results obtained from our simulation efforts.

---

### Official Review · Reviewer_L5cK · 2023-07-11

**Soundness:** 3 good
**Presentation:** 3 good
**Contribution:** 3 good
**Rating:** 5
**Confidence:** 4

**Summary:**

The authors investigate combining low-rank matrix factorization and (uniform scalar) quantization.
Through theoretical analysis and experiments they demonstrate that this can yield much higher accuracy than directly quantizing the input matrix. One natural choice is to compute the SVD of the matrix and quantize the two factors independently. It's shown that quantizing the left factor first and then computing a new right factor that best approximates the input when multiplied with the quantized left factor yields much better result. The choice of SVD is not critical and could be replaced by randomly mixing the columns of the input matrix, i.e. random sketching. In fact sketching has provable benefits as the entries of sketched matrices are bounded with very high probability. This bounded range improves quantization theoretically. The authors prove several theorems (their proofs are in the extensive appendix) and conduct detailed empirical evaluation.

**Strengths:**

1) All the key ideas of the paper (low-rank approximation, sketching) are sound.
2) Detailed theoretical analysis with rigorous proofs.
3) Extensive experiments.
4) Compression of neural network weights and embeddings via low rank approximation and quantization are popular and impactful topics both to reduce memory usage and to speed up training and inference.


**Weaknesses:**

1) Neither the ideas nor the analysis are particularly inventive in my opinion. It's self evident that optimizing the second factor after quantizing the first (LSVD) is superior to independent quantization (DSVD), in fact the process could be iterated further. While I appreciate the 30+ pages of proofs provided by the authors it seems as if they rely on chaining known results and techniques for sketching and random matrices combining with patient algebra.
2) Despite its seemingly weaker theoretical bounds LSVD is always one of the most accurate method in the experiments (see Tables 4-7). This is also clearly highlighted by the authors in the limitations section.
3) Quantization aware training (not considered in the paper) is highly likely to produce equivalent or better results.


**Questions:**

1) Line 226 and Theorem 3.2: For (1+eps') relative Frobenius error approximation LPLR requires m=O(k/eps') >> k columns, which the text seems to overlook. Could you please discuss?
2) Table 3: Could you also add LSVD to this experiment too? DSVD's Frobenius error is always the same 0.496, and LPLR's error is the same  the bottom of the table, when the bit budget is doubled, as in the top of the table. Could you discuss why? What was the rank (k) of DVSD, was it the same as m of LPLR? I.e. could you make Rank column header precise?
3) Matrix entries are typically rescaled to [0,1] before quantization as it's cheap to store their min/max (or some very low/high quantiles), even for each row (or column) of a matrix. How would such rescaling change your theoretical and empirical results?

4) Line 77: Sx -> x
5) Table 1, row of LPLR: delta is undefined
6) Lines 219-222: missing log2(), 4 times



**Limitations:**

Yes, limitation section is adequate.

---

> ### Author Rebuttal · Authors · 2023-08-09
>
> Dear Reviewer,
>
> Thank you for the time you invested in reading our paper and writing the review. We appreciate that reviewing is a voluntary effort and below, we address each of your concerns to resolve your queries.
>
> > $1 + \epsilon'$ relative Frobenius error approximation LPLR requires $m=O(k/\epsilon') \gg k$
>
> Indeed, increasing the sketch size $m$ leads to a reduction in the relative error concerning the best rank-$k$ approximation, denoted as $||A_k - A||_F^2$. However, it is crucial to consider that low-rank approximations of a matrix are valuable only when the matrix is inherently low-rank approximable, i.e., $||A_k - A||_F^2$ is already sufficiently small to begin with (and precisely zero if $A$ has a rank less than $k$). In such cases, selecting the sketch size $m$ to be slightly larger than $k$, such as $m = k + p$, where $p \geq 2$ is the oversampling factor, ensures that $\delta$ does not become arbitrarily large. Consequently, the additional approximation error introduced by sketching, denoted as $\delta||A_k - A||_F^2$, is of the same order of magnitude as the best rank-$k$ approximation error. This observation aligns well with practical applications where low-rank approximations of matrices are beneficial. The impact of the oversampling factor $p$ has been extensively studied in previous works, for example, see references [16] and [74]. Moreover, from a theoretical perspective, the relative error can be improved to $(1 + \delta)^{1/(2q + 1)}$ with $q$ power iterations, though this comes at the expense of increased computational complexity. In our numerical simulations, we have chosen the value of $m$ to ensure that given the values of $B$ and $B'$, the number of parameters being quantized results in a total bit requirement that achieves parity with the naive quantization benchmark.
>
> > Matrix entries are typically rescaled to [0,1]
>
> We have taken into theoretical account this specific scenario in Table 2, where we make the assumption that each element of the matrix adheres to $A_{ij} = O(1)$. This is achieved by appropriately adjusting the scale of the matrix entries. In our image compression experiments, we have also practically accounted for this rescaling, as we are aware that pixel values within images fall within the range of 0 to 255.
>
> It's crucial to emphasize the contrast between the scenarios depicted in Tables 1 and 2. In Table 1, we make the assumption that the matrix's rows are normalized, denoted as $||A^{(i)}|| = O(1)$. This assumption is particularly relevant when performing nearest neighbor classification, where each row of $A$ represents an embedding vector. In this context, scaling the norm doesn't alter the cosine similarity, which is why this normalization is appropriate.
>
> > Accuracy of LSVD
>
> Indeed, in Tables 4 to 7, LSVD stands out as one of the accurate methods. This should not be interpreted as a drawback; instead, it serves as a valuable insight. In fact, even theoretically, LSVD does not have weaker bounds than LPLR (cf. Thm. I.1 vs. Thm. 3.2) -- there's no $(1 + \delta)$ factor present for LSVD. LSVD precisely calculates the left low-rank factor as $Q(\widetilde{U}_k)$, where $\widetilde{U}_k \in \mathbb{R}^{n \times k}$ is derived from the first $k$ columns of $U\Sigma$. This requires computing the SVD of $A$, which is computationally intensive. On the other hand, LPLR delivers comparable performance to LSVD while demanding significantly less computation. When computing the SVD is practical, LSVD is as a strong contender and deserves consideration as an option. However, in cases where matrices become exceedingly large and SVD computation becomes infeasible, LPLR offers a viable solution to overcome this limitation. We discuss this in lines 1204 to 1210 of App. K.
>
> > Quantization aware training
>
> We are not really sure what is meant by quantization-aware training (QAT) in the context of low-rank approximation for matrix compression. For training a neural network, QAT adds quantize/de-quantize nodes, and treats the quantization loss as part of the training loss. Subsequent fine-tuning of the parameters makes the model more resilient. For QAT in matrix compression, did you mean treating $L$ and $R$ as functions of some trainable hyper-parameters, which are subsequently optimized? If so, yes, it could possibly decrease the approximation error, at the cost of additional computation (which can potentially be prohibitive).
>
> > Neither the ideas nor the analysis are particularly inventive in my opinion.
>
> We believe that the simplicity of our algorithm adds to its appeal and makes it a valuable contribution to the literature, bridging the fields of matrix compression, quantization, and sketching while ensuring compatibility with upcoming hardware developments. Embracing simplicity and clarity can play a crucial role in achieving scalability and facilitating future adaptability. Additionally, such basic principles serve as foundational building blocks that can be further optimized to support more sophisticated developments.
>
> We propose an idea which is simple in its execution and pragmatic in its implementation. Our idea is well supported for modern linear algebra libraries, while offering theoretical analysis to effectively identify the regime in which it offers value over conventional, standard methods. Moreover, our method offers a tunable knob to achieve effective bit-rates in software, not currently supported by hardware. For example, in Table 3, using LPLR with appropriate parameters bits is equivalent to operating at 1 or 2-bits per pixel, despite hardware working with a higher bit precision. Moreover, our method is adaptable to future hardware advancements, such as the emergence of 4-bit GPUs, which can significantly speed up our technique's primitive operations.
>
> > Table 3:
>
> We have rectified this in the last point in global response (and Table 1 of the PDF). Rank column refers to sketch size $(m)$ for LPLR and target rank $(k)$ for DSVD or LSVD.

---

> > ### Comment · Reviewer_L5cK · 2023-08-21
> >
> > Thanks for the detailed reply and explanation. I'll revise my final review upwards accordingly.
> >
> > If you happen to read this comment in time:
> >
> > re: "references [16] and [74]" - these are quantization papers, could you double check?
> >
> > re: QAT: I meant learning factors $L$ and $R$ directly with gradient descent in the quantized bottleneck layer of the form $Q(L)\cdot Q(R)$ replacing $A$, where $Q$ denotes quantization as before.

---

> > > ### Author Response · Authors · 2023-08-21
> > > **Response**
> > >
> > > Dear reviewer,
> > >
> > > Thank you for your response. We are glad that you found our rebuttal convincing and are grateful for your willingness to revise your review upwards.
> > >
> > > To answer your remaining questions:
> > >
> > > 1. Sincere apologies for the confusion -- we meant references [16] and [74] of our supplementary material, which includes references from the main paper as well as the appendix. The specific papers we pointed out to are:
> > >
> > > P. Drineas, R. Kannan, and M. W. Mahoney. Fast monte carlo algorithms for matrices II: Computing a low-rank approximation to a matrix. SIAM Journal on Computing, 36(1):158–183, 2006.
> > >
> > > R. Witten and E. Candès. Randomized algorithms for low-rank matrix factorizations: Sharp performance bounds. Algorithmica, 72(1):264–281, may 2015
> > >
> > > 2. QAT: Thank you for clarifying this. Yes, it is likely that QAT can decrease the approximation error further, but as we mention in our rebuttal, this comes at the cost of additional computation (which can potentially be prohibitive for compressing large data matrices).

---

### Official Review · Reviewer_WupA · 2023-07-11

**Soundness:** 3 good
**Presentation:** 2 fair
**Contribution:** 3 good
**Rating:** 6
**Confidence:** 2

**Summary:**

The paper studies compression of low-rank matrices by simultaneous low-rank factorization and quantization. It proposes a method that first quantizes the randomized rangefinder as the first low-rank factor and then quantize the minimizer of reconstruction error with respect to the remaining factor as the second. Randomized rangefinder uses random Gaussian matrix which possesses the equalization property to maintain low quantization error, compared to naïve quant. Experiments are provided to demonstrate the benefits of the proposed algorithm.

**Strengths:**

1. provide a low rank factorization algorithm that come with quantization for further reducing memory footprint.
2. experiments demonstrate the advantage of the algorithm.

**Weaknesses:**

1. experimental settings should be made clearer. the current description is a bit confusing.


**Questions:**

1. what is the difference between LPLR and LPLR-SVD in experiments?
2. Accuracy seems to decrease with bits for some cases in experiments. Why?

**Limitations:**

No limitations are addressed in the paper.

---

> ### Author Rebuttal · Authors · 2023-08-09
>
> Dear Reviewer,
>
> We are grateful for the time you spent in reading our paper and writing the review. We address your concerns below:
>
> > Difference between LPLR and LPLR-SVD in experiments
>
> LPLR refers to our main algorithm in Alg. $1$, in which the left low-rank factor is $Q(AS)$, where $S$ is the Gaussian sketching matrix. On the other hand, LPLR-SVD computes the left low-rank factor as $Q(U_kS)$, where $U_k$ is the matrix of top-$k$ singular vectors. Obtaining $U_k$ requires computing the SVD of the matrix which can be computationally prohibitive. LPLR-SVD is described in footnote $1$ on page $9$, and also in lines $111$ to $117$. LPLR-SVD is analyzed in detail in Appendix $I$.
>
> > Accuracy seems to decrease with bits for some cases in experiments.
>
> In cases where accuracy increases with decreasing number of bits, we believe that the quantization noise adds an inherent regularization effect. This conjecture is worth exploring in detail and we have acknowledged this in Appendix $K$ (lines $1211$ to $1215$).
>
> > No limitations are addressed in the paper.
>
> We have mentioned some limitations and further discussions in Appendix $K$.
>
> > Experimental settings should be made clearer. the current description is a bit confusing.
>
> We would be extremely grateful if you could kindly indicate any sections that require further clarity. We are more than willing to provide explanations and make any necessary edits. Furthermore, we have also done additional experiments which have been added to our global response, along with corresponding discussions.

---

### Official Review · Reviewer_z4CA · 2023-07-26

**Soundness:** 2 fair
**Presentation:** 3 good
**Contribution:** 2 fair
**Rating:** 6
**Confidence:** 4

**Summary:**

The paper proposed a memory efficient approach to approximate a matrix $A$ by: low-rank approximation $A=LR$ and quantization. The LPLR algorithm first applies a quantized random projection (RP) as the $L$, and then solve a minimization problem for the right loew-rank factor $R$, which is also quantized afterwards. Theoretical approximation error is obtained and compared with an alternative approach that quantized SVD low rank factors instead of RP. Experiments are conducted on image approximation and embedding classification, to show the effectiveness of the proposed method.

**Strengths:**

1. The paper is well-organized and easy to follow. The theoretical analysis seems rigorous.

2. Experiments on multiple ML tasks and datasets are provided which make the results more grounded.

**Weaknesses:**

In my understanding, the main idea of the paper is to waive the need to compute the SVD of A, by using random projection (RP) as a surrogate. I have the following concerns and suggestions:

1. At line 220, the authors wrote that the bits per entry for SVD-quant is $O(nd\sqrt k)$, but in Table 1 and Table 2, it is $O(k\sqrt{nd})$. Please double check and clarify. Also, the authors simply stated that LPLR is better than SVD-quant in terms of bits per query, which is not true with some n, d, m, k (comparing the results in the table). I suggest the authors to carefully compare the results and state the regimes when LPLR is better, and when it is worse.

2. In the main Theorem 3.2, $\kappa$ could be negative, right? Is $1-c_4\sigma_k/\sigma_{k-1}$ bounded? Or do we need to further assume an eigen gap for this result to hold?

3. I understand that the main usage of LPLR is for matrix (data) approximation, so the first experiments (Figure 1 and Table 3) make sense to me. However, for the second set of experiments on classification, why not directly using $Q(AS)$ (i.e., the quantized random projections)? This saves the storage for W (in other words, we may increase the sketch size m when using Q(AS) only). Some recent references on this include

Random projections with asymmetric quantization, Li and Li NeurIPS 2019

Generalization error analysis of quantized compressive learning, Li and Li, NeurIPS 2019

Indeed, the research on QRP is highly related to this submission, since LPLR essentially does an optimization on W to recover the data from QRP. I suggest to add some discussion on this direction in the paper and some empirical comparisons.

4. Also, if LPLR is used for processing or storing the data for classification or search tasks, it might be inconvenient to handle new data points (e.g., in a streaming setting). Thus, it may not be suitable for such tasks. On the other hand, recently people are using low rank approximation in LLM fine-tuning frequently. Experiments related to fine-tuning language models could be a better application scenario for the proposed method.

5. Some references on similar results are missing. A similar result as in Appendix D that $S^TQ(AS)$ with uniform quantization has approximation error independent of $d$ has been established in [EDEN: Communication-Efficient and Robust Distributed
Mean Estimation for Federated Learning, Vargaftik et al., ICML 2022] (or maybe some even earlier paper) for rotation matrix $S$. This related result should be cited. Also, Eq. (2) is a standard result for uniform stochastic rounding. A reference should be added there.

In all, I think the paper proposes a simple but intuitive method for low-rank low-precision matrix approximation from QRPs. The idea is clear, the analysis seems sufficient (despite the above and below questions).The experiments can be improved, but the current results on several tasks and datasets are convincing enough to show the effectiveness of LPLR in matrix approximation. For now, I would recommand borderline accept.

**Questions:**

1. Proposition D.1 only bounds the error between $Q(Sx)$ and $Sx$. How does it imply $||S^TQ(Sx)-x||^2$ is also a constant?

2. What solver is used to solve Algorithm 1 line 4? This should be clarified. Is closed-form feasible?

**Limitations:**

NA.

---

> ### Author Rebuttal · Authors · 2023-08-09
>
> Dear Reviewer,
>
> We are grateful for the time you spent in reading our paper and writing the review. We address your concerns below.
>
> > $Q(AS)$ for classification
>
> The works of Li & Li (2019), although related, are different from matrix compression addressed in our work. They study nearest neighbor search by approximating $Ax$ by $Q(AS)Q'(S^Tx)$, where rows of $A$ are datapoints, $x$ is the query vector, and $S$ is a Gaussian sketching matrix.
>
> 1. Storage: In Li & Li, in addition to $Q(AS)$, $S$ needs to be stored too, to process an incoming query $x$. So, the storage is $nm$ quantized and $dm$ full-precision (FP) entries. $S$ needs to be stored in FP because incoming $x$ needs to be processed. In contrast, LPLR stores only the quantized entries of $L$ and $R$, i.e., $nm + md$ quantized entries, and $Ax \approx LRx$. Hence, storage of LPLR is smaller.
>
> 2. Computation: In Li & Li, a computation of $S^Tx$ is needed for every query $x$. This requires $md$ FP mults, and computing $Q(AS)Q(S^Tx)$, i.e., $nmd$ quantized mults. In contrast, for LPLR, we compute $LRx$ with $m(n + d)$ mults, which is faster.
>
> Nevertheless, we acknowledge that these very interesting works and will definitely cite them.
>
> > Regimes when LPLR is better
>
> Thank you for pointing this out. The bit requirement is indeed $O(k\sqrt{nd})$ for direct-SVD quant., and line 220 is a typo which we will rectify. Tables 1 and 2 represent two distinct regimes. In Table 1, we assume that the $\ell_2$ -- norm of the $i^{th}$ row of matrix $A$ is bounded by a constant, denoted as $||a^{(i)}|| = O(1)$. Conversely, in Table 2, we consider that each entry of matrix $A$ is bounded by a constant, indicated by $A_{ij} = O(1)$.
>
> For the scenario described in Table 1, the bit-requirement for direct-SVD is $0.5\log_2(O(k\sqrt{nd}))$. Meanwhile, for LPLR, the bit-requirement is $0.5\log_2(\tilde{O}(nm/\sqrt{d}))$, disregarding the logarithmic terms. Evidently, LPLR demands fewer bits than direct-SVD because $k$ and $m$ are much smaller than $min(n,d)$, given that $n$ and $d$ can be substantially larger than $k$ and $m$ for inherently low-rank matrices. In the regime presented in Table 2, the bit-requirement for direct-SVD quantification remains $0.5\log_2(O(k\sqrt{nd}))$, unchanged from before. However, LPLR now requires $0.5\log_2(\tilde{O}(nm\sqrt{d}))$, slightly more than direct-SVD due to the additional $\sqrt{n}$ factor inside the logarithm. Thus, it makes sense to expect that direct-SVD can perform better in this regime.
>
> This observation is further supported by our numerical simulations in Tables 4 to 6, where direct-SVD indeed outperforms LPLR in certain scenarios. Nevertheless, it is crucial to emphasize that direct-SVD necessitates computing the SVD, which can be prohibitive for very large matrices due to the current memory limitations of available GPUs, making LPLR the only viable option. As discussed in lines 1204 to 1210 of App. K, if our objective is merely to compress an input matrix $A$ without concerning ourselves with the computational effort needed for compression, we can try all compression techniques and choose the one that yields the minimum Frobenius norm error.
>
> Thank you for raising this concern and we will definitely add these discussions to our main text.
>
> > Thm. 3.2, $\kappa$ could be negative?
>
> No, $\kappa$ is always positive. Yes, $(1 - c_4\sigma_k/\sigma_{k+1})$ can be negative in the statement of Thm. 3.2, but we do not need to assume an eigen gap for this result to hold true. If $1 - c_4\sigma_k/\sigma_{k+1} \leq 0$, we should set $\kappa = \kappa(A)$. This is evident from the proof of Lem. E.2 in lines 868 to 873. Our correct expression for $\kappa$ would be $\kappa = min(\kappa(A), \kappa(A_k)(1 - c_4\sigma_{k+1}/\sigma_k)^{-1})$ if $1 - c_4\sigma_k/\sigma_{k+1} > 0$, and $\kappa = \kappa(A)$, otherwise. Thank you for pointing this out, and we will rectify it in the statement of the theorem.
>
> > Streaming setting & LLM fine-tuning
>
> This is a very interesting point. It is possible to extend LPLR to streaming data settings using sketching based low-rank approximation. In this regard, LPLR is easier to convert to a streaming algorithm than direct SVD based quantization. A recent work on sketching based streaming low rank approximation is *Streaming Low-Rank Matrix Approximation with an Application to Scientific Simulation*, Joel A. Tropp et. al., SIAM Journal on Scientific Computing (2019), url = [https://doi.org/10.1137/18M1201068](https://doi.org/10.1137/18M1201068). Experiments related to fine-tuning large language models is also a potential application scenario we can consider. We will add related discussions on these in App. K.
>
> >  Prop. D.1
>
> Note that: $||S^TQ(Sx) - x||^2 \leq ||Q(Sx) - Sx||^2 + ||S^TQ(Sx)||^2 - ||Q(Sx)||^2 + ||x||^2 - ||Sx||^2 \leq ||Q(Sx) - Sx||^2 + ||S^TQ(Sx)||^2 + ||x||^2 \leq ||Q(Sx) - Sx||^2 + {\rm R}^2(\sigma^2_{max}(S) + 1)$.
>
> We know that $\sigma^2_{max}(S) \leq \frac{d}{m}$ with high probability for Gaussian $S$. The error only depends on the aspect ratio $d/m$, and not $d$ directly. Although we provide the above justification as a response to the question, in LPLR we do not explicitly compute the $S^TQ(AS)$ anywhere. The effect of sketching $AS$ in the first low-rank factor are nullified by the second low-rank factor.
>
> > Solver for $W$
>
> Yes, the solution of this problem is available in closed-form as $W^* = Q(AS)^{\dagger}A$, used in the analysis and also in the numerical simulations. We prefer to keep the general form in line 4 of Alg. 1 because one use an approximation of $W^*$, obtained using conjugate gradient descent, instead of the closed form expression.
>
> > Missing references
>
> Thank you for pointing out this nice work by Vargatik et. al. (2022). We have discussed the literature related to this in Sec. 1.1 (lines 76 to 84), which includes the work DRIVE [67], by the same set of authors. We will add the new reference to the list as well.

---

> > ### Comment · Reviewer_z4CA · 2023-08-11
> > **Thanks for the reply**
> >
> > Thanks for the reply. I still have some follow-up questions:
> >
> > 1. I don't get why LPLR needs less computations. $Q(AS)$ has the same size as $L$, and $Q(S^Tx)$ has the same size as $Rx$, right?
> >
> > 2. Thanks for clarifying. Please add this analytic comparison to the paper and update the theorem statement.
> >
> > 3. Can we extend the method in the mentioned SIAM paper to LPLR to handle the streaming data setting?  If so, I suggest adding some discussion and describing the general ideas. If not, it should be mentioned in the limitation section that LPLR should be used for fix/static data.
> >
> > 4. Prop. D.1: so the reconstruction error $||S^TQ(Sx)-x||^2$ is actually not $O(1)$ but is $d/m$ which increases with $d$?

---

> > > ### Author Response · Authors · 2023-08-12
> > > **Clarifications to follow-up questions**
> > >
> > > Dear Reviewer,
> > >
> > > Thank you once again for your thorough review, which really helped improve the clarity of our paper. Below, we have attended to your follow-up concerns:
> > >
> > > > Why LPLR needs less computations.
> > >
> > > We apologize for the confusion. Computing $Q(AS)Q(S^Tx)$ requires $nm$ multiplications (not $nmd$), which is the same as LPLR. You are correct -- we agree that the computation speedup of LPLR is not due to reduced number of multiplications. Nevertheless, in Li \& Li, the sketching matrix $S$ needs to be stored in full-precision (FP) for computing $S^Tx$ for any incoming $x$. Contrary to this, LPLR requires computation of $Q(W)x$, where $Q(W)$ consists of $md$ quantized values, which can leverage modern advancements in hardware primitives for speeding up low-precision computations (eg., half and mixed-precision compute). Thank you for bringing our attention to this.
> > >
> > > > Please add this analytic comparison to the paper and update the theorem statement.
> > >
> > > Thank you for acknowledging. We will make the necessary edits to the paper.
> > >
> > > > Can we extend the method in the mentioned SIAM paper to LPLR to handle the streaming data setting? If so, I suggest adding some discussion and describing the general ideas.
> > >
> > > An outline of how LPLR can be extended to the streaming setting is as follows:
> > >
> > > Suppose we have a matrix $A_n \in \mathbb{R}^{n \times d}$ with $n$ datapoints, for which we store the sketching matrix $S \in \mathbb{R}^{d \times m}$, the left factor $(L_n)$, and the right factor $(R_n)$. For an incoming datapoint $a_{n+1}$, we can simply update the left factor as: $L_{n+1} \gets [L_n; Q(a_{n+1}S)]$, where $;$ denotes the concatenation of an additional row. The second low-rank factor, which is $R_{n+1} = argmin_W\lVert L_{n+1}W - A_{n+1}\rVert_F$, can be computed from $R_n$ in a fashion similar to online least squares, which uses Woodbury matrix inversion lemma.
> > >
> > > Thank you once again for highlighting this setting. This is a very interesting observation and we will add a discussion regarding this to the main paper.
> > >
> > > > Prop. D.1: so the reconstruction error $\lVert S^\top Q(Sx) - x \rVert^2$ is actually not $O(1)$ but is $d/m$ which increases with $d$?
> > >
> > > Yes, the reconstruction error $\lVert S^TQ(Sx) - x\rVert^2$ scales as $d/m$. But it does not necessarily increase with $d$ if we choose the sketch size $m$ to be proportional to $d$, i.e., $m = O(d)$. In that case, $d/m$ will be $O(1)$. We are grateful for your careful scrutiny. We will add the above discussion to the paper, and mention that the reconstruction error $\lVert S^\top Q(Sx) - x \rVert^2$ does not grow with dimension as long as the sketch size $m$ is proportional to the dimension $d$.

---

> > > > ### Comment · Reviewer_z4CA · 2023-08-12
> > > >
> > > > Thanks for the quick response.
> > > >
> > > > 1. Thanks for clarification. Since QRP is just the left component in LPLR, comparing LPLR and QRP could better demonstrate the benefit brought by the extra optimization problem (3). Anyway, we can say that LPLR targets on a different task of data approximation, so not having such empirical results is OK to me. Please cite the QRP related papers and discuss their connections in the revision.
> > > >
> > > > 2. In the "online" procedure you described, you need the entire data matrix $A_{n+1}$ to do the optimization. However, the original data should have been discarded since the whole motivation about this work is that $A$ is too large to store. I'm not against this, since many of these type of optimization-based algorithms do not handle online data very well. My suggestion is to discuss this setting a bit to at least let the readers know the best application scenario of LPLR, because streaming data is indeed an important case in practice.
> > > >
> > > > 3. Thanks for confirming that the error is in fact not a constant but $d/m$. In the paper you mentioned $m\ll d$ to achieve good compression, so I think this error should be regarded as linear in $d$. I was particularly curious about this claim since prior work showed constant approximation error with $d\times d$ rotation matrix. This matrix is much "stronger" than low-rank random projection matrix. Now the $d/m$ error seems more intuitive and reasonable. There might also be a connection with their result when $m=d$, which could be interesting to the researchers. Anyway, please add the proof and revise the description of this result in the revision.
> > > >
> > > > In summary, thanks for answering my questions. Please update the paper according to our discussions. This revision would improve the quality of the paper. Based on this, I will raise my score to 6.

---

> > > > > ### Author Response · Authors · 2023-08-20
> > > > > **Acknowledgement of discussion**
> > > > >
> > > > > Dear Reviewer z4CA,
> > > > >
> > > > > Thank you for engaging in discussions and increasing your score. We commit to citing the QRP papers to provide a more comprehensive context for our nearest neighbor simulations. We agree with your suggestion regarding the extension of LPLR to an online setting. We believe that it is possible to design an approximate online version for which storing a sketch of the matrix $A$ suffices, i.e., $R_{n+1} = argmin_W\lVert L_{n+1}W - A_{n+1} \rVert_F$ can be obtained in terms of $R_n = argmin_W\lVert L_nW - A_n \rVert_F$, without having to store $A_n$ or $A_{n+1}$. This hinges on appropriately capturing the delta from $A_n$ to $A_{n+1}$. We refrain from delving into details since this is not what we study in our work. Nevertheless, we agree that this is an interesting future research direction and we will add a pertinent discussion in our paper. Regarding point 3, we  will clarify the result to be explicit about the $d/m$ dependence. We agree that there may be a connection with their result, which is worthy of exploration in future work. We are grateful for your incisive review of our work, and will incorporate the outcome of this discourse into our revision.

---

### Official Review · Reviewer_DCJv · 2023-07-27

**Soundness:** 2 fair
**Presentation:** 3 good
**Contribution:** 2 fair
**Rating:** 5
**Confidence:** 3

**Summary:**

The paper studies the low-rank factorization of the matrix in the low-precision setting and proposes a new algorithm which is a combination of randomized low-rank approximation method and quantization.  The paper formally analyzes the guarantee of the proposed algorithms and also give experiments on real world dataset which demonstrate the advantage of the proposed algorithm.

**Strengths:**

1. The presentation of the paper is good. The writing of the paper is clear and easy to follow.

2. To get the formal guarantee, the authors do a careful analysis, which I think is non-trivial.

**Weaknesses:**

1. Technical novelty: the main algorithm (Algorithm 1) seems to just be the standard way in randomized numerical linear algebra then plusing the quantization. Can you authors give more explanations about the technical novelty? (though the analysis I think is not standard)

2. Experiment: I have some questions about the setting and details of the experiments section. See the next question.

**Questions:**

1. Theorem 3.2 shows the guarantee of the proposed algorithms. However, would the sketch size $m$ also have the requirement given the accuracy parameter $\epsilon$ and $k$? Also in experiment it may be a factor that affects the runtime and accuracy a lot, is there some place indicating the choice of $m$?

2. I am a little confused about the naive quantization baseline. Can this way make the matrix be low-rank?

3. As mentioned in the paper, the sketching-based method is popular in randomized low-rank approximation. I think in the experiment the baselines should also include it (with the naive way of quantization).

4. The paper discusses the runtime complexity. Hence, I think it would be better to include the runtime in the experiment section.

**Limitations:**

See the above questions.

---

> ### Author Rebuttal · Authors · 2023-08-09
>
> Dear Reviewer,
>
> Thank you for the time you invested in reading our paper and writing the review. We appreciate that reviewing is a voluntary effort and below, we address each of your concerns to resolve your queries.
>
> > Choice of $m$
>
> Indeed, the sketch size $m$ is an important design parameter. We have explained our choice of $m$ for experiments in lines $269$ -- $276$, where we mention that $m$ is selected so that the bit-budgets are identical between naive quantization and LPLR, i.e., parity is ensured. In Table $3$, the choice of $m$ are the values mentioned in the {\it Rank} column. Note that the values of $m$ satisfy $(nm + dm)\cdot{\rm B} \leq nd\cdot {\rm B_{nq}}$, i.e., $m = \left\lfloor \frac{nd \cdot{\rm B_{nq}}}{(n+d)\cdot{\rm B}} \right\rfloor$. We optimize the choice of $m$ so that the bit requirement of LPLR does not exceed that of naive quantization. The same holds true for embedding classification experiments.
>
> Theoretically, if the matrix $\mathbf{A}$ has approximate rank $k$, i.e., $\lVert \mathbf{A}_k - \mathbf{A} \rVert_F$ is sufficiently small, it suffices to choose $m$ to be slightly larger than $k$, such as $m = k + p$, where $p \geq 2$ is the oversampling factor. The impact of the oversampling factor $p$ has been extensively studied in previous works, for example, see references $[16]$ and $[74]$.
>
> > Can naive quantization make the matrix be low-rank?
>
> Naive quantization *does not* make the matrix low-rank. We have included it as a baseline since the primary goal of the paper is to compress the matrix, and naive quantization, which does not exploit low-rank structure, is a standard practice in existing literature. In our comparisons, we show that there is a better way to compress the matrix, namely, LPLR, which exploits the low-rank structure and attains a smaller error than naive quantization, while maintaining parity w.r.t. the bit-requirement.
>
> > Sketching + Naive quantization as baseline
>
> Utilizing sketching along with naive quantization would sketch the columns of $\mathbf{A}$ as $\mathbf{AS}$, and find the right factor as $argmin_{\mathbf{W}}\lVert \mathbf{ASW - A} \rVert_F^2 = \mathbf{(AS)^{\dagger}A}$. A subsequent naive quantization would give the factorization ${\rm Q}({\bf AS}){\rm Q'}(({\bf AS})^{\dagger}{\bf A})$. This is indeed an alternative benchmark, although it cannot achieve a lower error compared to LPLR because:
> $\lVert{\rm Q} ({\bf AS}){\bf W}^* - {\bf A}\rVert_F^2 \leq \lVert {\rm Q}({\bf AS}){\rm Q'}(({\bf AS})^{\dagger}{\bf A}) - {\bf A}\rVert_F^2$, as ${\bf W}^* = argmin_{\bf W}\lVert {\rm Q}({\bf AS}){\bf W} - {\bf A}\rVert_F^2 = {\rm Q}({\bf AS})^{\dagger}{\bf A}$. If ${\rm B'}$ is sufficient so that ${\rm Q}'({\rm Q}({\bf AS})^{\dagger}{\bf A})$ is closer to ${\rm Q}({\bf AS})^{\dagger}{\bf A}$ than ${\rm Q}'(({\bf AS})^{\dagger}{\bf A})$, then LPLR will have a smaller error.
>
> > Novelty
>
> We propose an idea which is simple in its execution and pragmatic in its implementation. Our idea is well supported for modern linear algebra libraries, while offering theoretical analysis to effectively identify the regime in which it offers value over conventional, standard methods. Moreover, our method offers a tunable knob to achieve effective bit-rates in software, not currently supported by hardware. For example, in Table $3$, using LPLR with appropriate parameters bits is equivalent to operating at $1$ or $2$-bits per pixel, despite hardware working with a higher bit precision. Moreover, our method is adaptable to future hardware advancements, such as the emergence of $4$-bit GPUs, which can significantly speed up our technique's primitive operations. We believe that the simplicity of our algorithm adds to its appeal and makes it a valuable contribution to the literature, bridging the fields of matrix compression, quantization, and sketching while ensuring compatibility with upcoming hardware developments.
>
> > Runtime experiments
>
> Thank you for the valuable suggestion. As a response, we have included some additional experiments on the wall-clock runtime comparison of our method. They can be found in Table 2 and Fig. 3 of the global response PDF.
>
> Table 2 now presents the wall-clock time required for compressing the embeddings of the CIFAR-10 and CIFAR-100 datasets. It is evident that LPLR significantly outperforms DSVD and LSVD in terms of speed. Fig. 3 complements the tabulated data by visualizing the runtime involved in two scenarios. The first scenario involves computing the matrix-vector multiplication $bf Ax$ directly. The second scenario involves approximating this multiplication using a low-rank factorization, specifically $\bf L(Rx)$, where matrices $\bf L$ and $\bf R$ are tall and wide matrices, respectively. This low-rank factorization strategy reduces the overall number of multiplications from $nd$ to $(n + d)m$.

---

> > ### Comment · Reviewer_DCJv · 2023-08-13
> >
> > Thanks for the reply. I still have the follow-up question:
> >
> > 1. As mentioned by the authors, naive quantization does not make the matrix low-rank. However, if the bit-budget is the same, why the LPLR has a better error than naive quantization in Table 3. In that case, would naive quantization get the optimal error in this setting?

---

> > > ### Author Response · Authors · 2023-08-14
> > > **Response to follow-up question**
> > >
> > > Thank you for your question. To clarify your concern, allow us to revisit the relationship of LPLR with its antecedents for compressing models, namely Direct SVD (which reduces the total number of parameters by leveraging low-rankness) and Naive quant (which reduces the number of bits used to represent each parameter). The efficacy of LPLR relies on finding a harmonious equilibrium between these two key factors.
> > >
> > > In Table 3, we do not impose prior expectations on the performance of any particular algorithm. For instance, we don't possess any prior knowledge of the optimal target rank. However, given our analytical demonstration that there exists a regime (a combination of target rank and bit budget *per parameter*) where LPLR can surpass its baselines, we highlight this scenario (extreme model compression) in the *revised* Table 3, which corresponds to what is now Table 1 of the global response pdf. Naive Quant is indeed capable of outperforming LPLR/DSVD/LSVD (and does so) where significant model compression is not a requirement. This can be observed in rows $1$ to $4$ of the second sub-table in the global pdf.
> > >
> > > We hope this clarifies your query. In what follows, we further elaborate this.
> > >
> > > Let's consider compressing $A \in \mathbb{R}^{n \times d}$. If we choose our target rank (essentially, the sketch size) to be $k$, leading to an approximation of $A \approx LR$, where $L \in \mathbb{R}^{n \times k}$ and $R \in \mathbb{R}^{k \times d}$, the total parameter count in $L$ and $R$ becomes $k(n + d)$, instead of the original $nd$ parameters in $A$. If we employ naive quant., which directly quantizes each element of $A$ without capitalizing on any low-rank structure, the total number of bits utilized amounts to $ndB_{nq}$, where $B_{nq}$ denotes the number of bits assigned for quantizing each entry. On the other hand, in the case of LPLR/DSVD/LSVD, where $k(n + d)$ parameters are quantized, the overall bit consumption equals $k(n + d)B$, where $B$ represents, once again, the bit allocation per entry (potentially distinct from $B_{nq}$).
> > >
> > > In Table 3, a fair comparison is upheld by ensuring that each algorithm receives an equal allocation of resources—specifically, they utilize the same total number of bits. This similarity in resource allocation serves as the basis for evaluating their effectiveness in judiciously allocating the bits to maintain performance. In Table 3, the relevant columns, namely $B$ and $B_{\rm nq}$, correspond to LPLR/DSVD/LSVD and naive quantization, respectively. However, it is important to emphasize that we guarantee that the expression $$ \text{Total Bit Budget} = \text{Number of parameters} \times \text{Bit budget per parameter} = k(n+d)B  = ndB_{nq}$$
> > >
> > > remains consistent **across all algorithms**. For matrices that can be effectively approximated by low-rank methods, $\lVert A_k - A \rVert_F$ (where $A_k$ is the best rank-$k$ approximation) is small for some $k \ll {\rm min}(n,d)$. For such matrices, there's room to allow $B$ to exceed $B_{nq}$. This flexibility permits us to employ more bits per parameter while maintaining the same total bit count. The choice of hyperparameter $k$ plays a pivotal role in navigating the trade-off between the error stemming from low-rank approximation and the error resulting from quantization due to precision limitations. A smaller value of $k$ translates to a higher number of bits per parameter, leading to reduced quantization error. Conversely, a larger value of $k$ diminishes low-rank approximation error, but results in fewer bits per parameter, subsequently increasing the quantization error. **Matrices inherently characterized by low-rank traits allow for a substantially smaller $k$ value to be chosen, and within this context, LPLR exhibits superior performance compared to naive quantization.**
> > >
> > > If any concern still remains, please don't hesitate to ask for further clarification.

---

> > > > ### Author Response · Authors · 2023-08-17
> > > > **Further Clarifications**
> > > >
> > > > Dear Reviewer DCJv,
> > > >
> > > > Please let us know if your query about the comparative advantage of LPLR and naive has been addressed satisfactorily. We hope that our response has positively influenced your perception of our work. If you require further clarifications to potentially reconsider your score, we are enthusiastic about engaging in further discussion. We highly value the generous contribution of your time to review our paper.

---

> > > > > ### Comment · Reviewer_DCJv · 2023-08-18
> > > > >
> > > > > Thanks for the detailed response. I think some parts of the response will make the main body of the paper more clear. I will raise my score to 5.

---

### Official Review · Reviewer_D7Xb · 2023-08-03

**Soundness:** 3 good
**Presentation:** 4 excellent
**Contribution:** 3 good
**Rating:** 7
**Confidence:** 4

**Summary:**

The paper introduces a low rank, quantized/low precision matrix factorization which decomposes an n x d matrix A in the form A= LR, where L (of size n x m) and R (of size m x d) are low rank factors. L and R are computed using a random projection matrix S in the form L = Q(AS) and R = Q'(W^*) where W^* is the matrix minimizing the squared Frobenius norm ||Q(AS)W− A||. Q and Q' are two independent quantizers with specified budgets.

The authors contrast their method with an SVD based method for computing the quantized low rank approximation which instead sets L = Q(U_k S_k) and R = Q'(V_k), where U_k/V_k and S_k are the singular vectors/values respectively. The paper has a theorem deriving a bound on the Frobenious norm of the factorization eror. They apply the approximation on image data (for image compression) and embedding matrices (for an embedding classifation task).


**Strengths:**

+ The paper is very well written and very clear in its presentation
+ Clear technical presentation incl. theorems, algorithms etc.
+ Novel idea, providing good review of relevant literature (different matrix sketching approaches, quantization etc.)
+ Thoughtful experiments to demonstrate real world application (using embedding compression)


**Weaknesses:**

I don't see any weaknesses that need to be addressed at this moment.

**Questions:**

* In Table 3, why are the SVD and Naive quant. Frobenious norm errors the same for different rank choice? Is it because the matrices are really low (<15) rank?

* I think Table 3 may be expanded to a set of figures to better convey the message (with potentially more bit budgets).

**Limitations:**

This is an optimization paper and as such the limitations are not immediately obvious, though the eventual use of the model for downstream compression/computation speed up could be quite broad. The societal impact could also be positive given the matrix compression can lead to compute/energy savings. The paper doesn't have a dedicated discussion/section on any limitations.

---

> ### Author Rebuttal · Authors · 2023-08-09
>
> Dear Reviewer,
>
> We are grateful for the valuable time you spent in reading our paper and writing the review. The voluntary nature of the review process is truly valued. In what follows, we address your questions:
>
> > Similar Frobenius norm errors for different rank choice
>
> For naive quantization, the Frobenius norm is not influenced by the rank but is exclusively determined by the allocated bit budget, denoted as $B_{nq}$. As a result, it maintains a consistent value when the bit budget for naive quantization remains unchanged.
>
> Regarding the low-rank methods, namely Direct SVD and LPLR, we identified (and fixed) a software bug in the parameter enumeration code, which led us to mistakenly assess all outcomes under a fixed sketch size equivalent to a rank of 200. This caused a lack of parity with respect to the bit requirement for naive quantization, and the comparison was unfair. Consequently, the Frobenius norm values appeared relatively steady; the slight variance in LPLR stemming from distinct samples of the sketching matrix. We sincerely apologize for this error and present revised outcomes in Table 1 of the global response PDF, now encompassing a significantly broader range of bit budgets.
>
> In Table 1, we assess the performance of LPLR, Direct SVD, and LSVD across a uniform range of bit budgets (including many that are not hardware primitives). This approach enables a more comprehensive examination of the approximation error and its variability when employing different low-rank approximation techniques with varying bit budgets. One can clearly observe that LPLR and LSVD (LPLR-SVD) outperform naive quantization for $1$ -- bit quantization, except in the extreme instance of $\rm B = 32$. This trend persists even as bit budgets increase. For higher bit allocations, the superiority of LPLR (and LSVD) over NQ in terms of approximation error is apparent when the bit allocation $\rm B$ is sufficiently low. This allocation strategy permits a greater allocation of storage to capturing higher-rank components. This delicate balance lies at the core of leveraging the effectiveness of LPLR, which excels in finding the optimal compromise between capturing the optimal number of low-rank factors with precision. We hope that this clarifies the rationale behind the scenario in which LPLR outperforms standard techniques, namely naive quantization.
>
> > Table 3 may be expanded to a set of figures to better convey the message
>
> Thank you for suggesting this. We concur completely, and are forced to prioritize due to the limited number of pages available. We will certainly add a number of figures to the appendix to better convey the efficacy of our method. We have included additional results involving a wider range of bit budgets in the global response PDF (ref. Table 1), which provides a clearer picture of the method's benefits.
>
> > Dedicated discussion/section on limitations.
>
> We have discussed some limitations in Appendix K.

---

> > ### Comment · Reviewer_D7Xb · 2023-08-17
> > **Thanks for your response.**
> >
> > Thank you for the detailed response. It addresses my concerns and questions and it's great that you caught and addressed the bug that caused the steady errors across parameters.

---

### Author Rebuttal · Authors · 2023-08-09

Dear Reviewers,

We greatly appreciate the time you invested in reviewing our paper and sharing your concerns. As part of the global response, we have included deliberations regarding the scenarios in which LPLR demonstrates its practical utility and potential advantages over established baselines. Furthermore, we report additional experimental evaluations pertaining to image compression, along with wall-clock runtimes for LPLR and alternative baselines.

**Comparison between LPLR and baselines**: In Tables 1 and 2, we have compared LPLR with other benchmarks. In the column denoting approximation errors, aside from the common quantization error $\epsilon$ present across all rows, DSVD and LPLR introduce an additional term representing the error from low-rank approximation. Consequently, it might appear that naive quantization consistently outperforms these methods. However, it's essential to recognize that compression techniques based on low-rank factorization hold value exclusively when the matrix being compressed is inherently low-rank, meaning that $\lVert \mathbf{A}_k - \mathbf{A} \rVert_F^2$ starts off as small. If $\lVert \mathbf{A}_k - \mathbf{A} \rVert_F = 0$, LPLR can achieve identical error levels as naive quantization, while demanding fewer bits than the latter. In other words, given the same bit-budget, LPLR can achieve a lower level of error compared to naive quantization.

**Computation-constrained compression**: In situations where computational resources are limited, employing the naive strategy emerges as the most cost-effective approach for matrix quantization. Nevertheless, due to its failure to capitalize on a matrix's inherent low-rank arrangement, naive quantization may prove considerably suboptimal. As matrices increase in dimension, accommodating them in memory becomes impractical, rendering approaches like DSVD or LSVD unviable. In such scenarios, our LPLR method stands as a viable alternative, demanding slightly more computational effort than naive quantization, yet capable of harnessing the low-rank structure for improved approximation accuracy.

**Compression without any computation constraint**: If there is no scarcity of computation resources in being able to compute the SVD, it is possible to compress the matrix with all the strategies, and choose the best one.

**Benefits beyond compression**: A low-rank and low-precision factorization $\mathbf{A \approx LR}$ also enables us to approximate the matrix-vector product $\mathbf{Ax}$ by $\mathbf{L(Rx)}$, which can be computed much faster as $\mathbf{L}$ and $\mathbf{R}$ are tall and thin matrices. This is shown in Fig. 3 of the PDF.

**Clarification to Table 3 of the main paper**:  Regarding the low-rank methods, namely Direct SVD and LPLR, we identified (and fixed) a software bug in the parameter enumeration code, which led us to mistakenly assess all outcomes under a fixed sketch size equivalent to a rank of 200. This caused a lack of parity with respect to the bit requirement for naive quantization, and the comparison was unfair. Consequently, the Frobenius norm values appeared relatively steady; the slight variance in LPLR stemming from distinct samples of the sketching matrix. We sincerely apologize for this error and present revised outcomes in Table 1 of the global response PDF, now encompassing a significantly broader range of bit budgets.

In Table 1, we assess the performance of LPLR, Direct SVD, and LSVD across a uniform range of bit budgets (including many that are not hardware primitives). This approach enables a more comprehensive examination of the approximation error and its variability when employing different low-rank approximation techniques with varying bit budgets. One can clearly observe that LPLR and LSVD (LPLR-SVD) outperform naive quantization for $1$ -- bit quantization, except in the extreme instance of ${\rm B} = 32$. This trend persists even as bit budgets increase. For higher bit allocations, the superiority of LPLR (and LSVD) over NQ in terms of approximation error is apparent when the bit allocation $\rm B$ is sufficiently low. This allocation strategy permits a greater allocation of storage to capturing higher-rank components. This delicate balance lies at the core of leveraging the effectiveness of LPLR, which excels in finding the optimal compromise between capturing the optimal number of low-rank factors with precision. We hope that this clarifies the rationale behind the scenario in which LPLR outperforms standard techniques, namely naive quantization.

---

### Decision · Program_Chairs · 2023-09-21

**Decision:**

Accept (poster)

**Comment:**

The reviewers unanimously recommended acceptance of this paper since it contributes a nice and simple solution to the important problem of matrix compression via low-rank approximation and quantization. The paper is well-executed and well-written.

We encourage the authors to take the reviewers' various suggestions into account when revising the paper. Notably, a reviewer noticed a bug in the experiment results, which requires updating to revised results.

The AC also would like to point out a minor presentation issue: one should never write log_2(O(f(params)). O(f(params)) is a statement about the asymptotic growth rate of f with respect to params. It is not a numerical quantity that you can take the log of. It would be acceptable to write log_2(c*f(params)) for some sufficiently large constant c, or something similar.